# New adjusted missing value imputation in multiple regression with simple random sampling and rank set sampling methods

**Juthaphorn Sinsomboonthong**[1], **Saichon Sinsomboonthong**[2¤☯*]

**1** Department of Statistics, Faculty of Science, Kasetsart University, Bangkok, Thailand, **2** Department of Statistics, School of Science, King Mongkut's Institute of Technology Ladkrabang, Bangkok, Thailand

¤ Current address: Department of Statistics, King Mongkut's Institute of Technology Ladkrabang, Bangkok, Thailand.
☯ These authors contributed equally to this work.
* saichon.si@kmitl.ac.th













**Data availability statement:** The data used to support this study were simulated from a normal distribution using the R studio program.

## Abstract

This research compared the efficiency of several adjusted missing value imputation methods in multiple regression analysis. The four imputation methods were the following: regression-ratio quartile1,3 (R-RQ1,3) imputation of Al-Omari, Jemain and Ibrahim; adjusted regression-chain ratio quartile1,3 (AR-CRQ1,3) imputation of Kadilar and Cinji; adjusted regression-multivariate ratio quatile1,3 (AR-MRQ1,3) imputation of Feng, Ni, and Zou; and adjusted regression-multivariate chain ratio quartile1,3 (AR-MCRQ1,3) imputation of Lu for each simple random sampling (SRS) and rank set sampling (RSS). The performance measures mean square error (MSE) and mean absolute percentage error (MAPE). The study showed that the AR-MRQ1 method with SRS provided the minimum mean square error for small error variance. However, the AR-MCRQ3 provided the minimum mean square error for a large error variance. Considering all error variance in mean absolute percentage error, the AR-MCRQ1 provided the minimum mean absolute percentage error. The AR-MRQ1 method with RSS provided the minimum mean square error for a small error variance. However, the AR-MCRQ3 provided the minimum mean square error for medium and large error variance. Regarding the mean absolute percentage error measure, the AR-MRQ1 provided the minimum mean absolute percentage error for a small error variance. However, the AR-MCRQ1 provided the minimum mean absolute percentage error for medium and large error variance. For both SRS and RSS, AR-MCRQ1 was the best method for missing value imputation in multiple regression analysis, followed by AR-MCRQ3. Moreover, the RSS estimators provided smaller MSE and MAPE than the SRS estimators. Therefore, the RSS estimators were more efficient than the SRS estimators.

## 1. Introduction

Multiple regression analysis is a study of the relationship between many independent variables and a dependent variable to determine which independent variables can estimate or

I have made my raw simulation data available in two attached files in Microsoft Word to this manuscript. My raw simulation data available was included in the simulation code. However, the actual data on gold and platinum losses were taken from an anonymous jewelry company. Moreover, I also have the uploading of the actual data as a supporting information file in Microsoft Word to this manuscript.

**Funding:** The authors would like to thank the committees of the Department of Statistics and other Departments, School of Science, King Mongkut's Institute of Technology Ladkrabang (KMITL) for considering funding the research project [grant number 2567-02-05-009] The funders had no role in study design, data collection and analysis, publication decisions, or manuscript preparation.

**Competing interests:** The authors have declared that no competing interests exist.

explain the variation of the dependent variable. The regression model is a matrix of $Y = X\beta + \varepsilon$. Estimation of the parameters often causes problems. Information or data can be incomplete, and significant data values may be missing, resulting from erratic data storage or transfer tools or limitations of the technology, leading to an inability to utilize the data [1] fully. To use multiple regression analysis effectively, the data must be complete. For this reason, some researchers may not include the data of some variables and make an analysis only on a smaller set of full data, which may not give a meaningful result of a biased estimate or a high standard error. The discarded data can differ significantly from the remaining data [2], making the estimation unreliable. Therefore, a reasonable estimation of missing values is fundamental.

Simple random sampling (SRS) is a sampling of a population of size N, where each possible sample of size n had the same probability of being selected. The sample obtained at such sampling is called a simple random sample [2]. Rank set sampling (RSS) was introduced by McIntyre (1952) for estimating mean pasture and forage yields as a more efficient and cost-effective method than the commonly used simple random sampling in situations where the visual ordering of the sample units can be done quickly, the exact measurement of the units is difficult and expensive [3]. Takahasi and Wakimoto (1968) provided the necessary mathematical theory of RSS [4]. Samawi and Muttlak (1996) studied using RSS to estimate the population ratio and suggested the RSS estimator of the population ratio [5]. The data sampled by SRS and RSS may be quite different, and the various proposed missing value imputation methods investigated in this study may give different results from the data obtained by the two sampling methods.

Three assumptions are discussed in the context of missing imputation, which also forms the basis of later simulations: missing completely at random (MCAR), missing at random (MAR), and missing not at random (MNAR) [6–8].

It is essential to understand why the data are missing. Graham et al. (2003) in [7] described that missing data can informally be thought of as being caused in some combination of three ways: random processes, processes that are measured, and processes that are not measured. Modern missing data methods generally work well for the first two causes, but not for the last. More formally, missing data assumptions are commonly described as falling into one of three categories, defined by [9]: First, data can be "Missing Completely at Random", or MCAR. When data are MCAR, missing cases are no different than non-missing cases in the analysis. Thus, these cases can be thought of as randomly missing from the data, and the only real penalty for failing to account for missing data is loss of power. Second, data can be "Missing at Random", or MAR. In this case, missing data depends on known values and thus is described entirely by variables observed in the data set. Accounting for the values that "cause" the missing data will produce unbiased results in an analysis. Third, data can be missing in an unmeasured fashion, termed "nonignorable" (also called "Missing Not at Random" (MNAR) and "Not Missing at Random" (NMAR)). Since the missing data depends on events or items that the researcher has not measured, it cannot be adquately attributed without measuring some related variables.

Missing data can frequently occur in a longitudinal data analysis. Many imputation methods have been proposed in the literature to handle such an issue. Complete case (CC), mean substitution (MS), last observation carried forward (LOCF), and multiple imputation (MI) are the four most frequently used methods in practice. In a real-world data analysis, the missing data can be MCAR, MAR, or MNAR, depending on why the data was missed. In this paper, simulations under various situations (including missing rates and slope sizes) were conducted to evaluate the performance of the four methods considered using bias, RMSE, and 95% coverage probability as evaluation measures. The results showed that LOCF has the most significant bias and the poorest 95% coverage probability in most cases under both MAR and

MCAR assumptions. Hence, LOCF should not be used in a longitudinal data analysis. Under MCAR assumption, CC and MI methods performed equally well. Under MAR assumption, MI has the minor bias, smallest RMSE, and best 95% coverage probability. Therefore, the CC or MI method is the appropriate method to be used under MCAR. In contrast. the MI method is a more reliable and better-grounded statistical method under MAR MAR [10].

In 1996, rank set sampling (RSS) was studied by [5], which was suggested by [3] and developed by [4] to examine the ratio estimator. It was proved that RSS estimators were more efficient than simple random sampling (SRS). Later in 2003, the chain ratio (CR) estimator with SRS was examined [11]. The efficiency comparison used the mean squared error (MSE). They proved that the CR estimator was more efficient than the traditional ratio estimator under certain conditions.

A 2006 study investigated missing value estimation in multiple regression analysis. Three methods were used to estimate the missing values of a dependent variable: regression, EM algorithm, and pairwise deleted methods. The sample sizes were 20, 100, and 150. The missing values were 10%, 30%, and 50%. The error variances were 25, 50, 75, and 100. The evaluation measure was MSE. The study found that the regression method had the lowest MSE from all 500 iterations, outperforming other methods in many situations [12].

In 2009, modified ratio estimators of the population mean of a variable of interest were suggested, involving the first or third quartiles of an auxiliary variable correlated with the variable of interest. The newly suggested estimators were investigated using SRS and RSS methods. The efficiency was compared in terms of MSE. These estimators were unbiased, and the RSS estimators were more efficient than SRS estimators for the same quartile with the same correlation coefficient and sample size. Also, when the two quartiles were compared, an estimator based on the regression-ratio quartile3 (R-RQ3) imputation of Al-Omari, Jemain, and Ibrahim was more efficient than an estimator based on the regression-ratio quartile1 (R-RQ1) imputation of Al-Omari, Jemain, and Ibrahim [13].

In 2012, imputation methods were proposed when data on the dependent variable was missing in multiple linear regression analysis. The proposed methods of estimation were called ratio-Q1 imputation method (RQ1), ratio-Q3 imputation method (RQ3), regression-ratio-Q1 imputation method (R-RQ1), and regression-ratio-Q3 imputation method (R-RQ3). The efficiency of the proposed methods was compared to the mean and regression imputation method in a variance simulation situation. The evaluation measures were estimated root mean squares error (RMSE) and mean absolute percentage error (MAPE). In each tested situation, linear regression models with two independent variables were considered under the assumption that the error was distributed normally. The missing value is missing at random results showed that the R-RQ1 imputation of Al-Omari, Jemain, and Ibrahim was more efficient in the following situations: (1) a small sample size (n = 20), a large percentage of missing values (20%), a large value of variance ($\sigma^2$ = 1.5, 2); (2) the medium sample size (n = 40, 60), a small percentage of missing values (10%), significant value of variance ($\sigma^2$ = 0.5, 1); (3) the medium sample size (n = 40, 60), the medium percentage of missing values (15%), the medium value of variance ($\sigma^2$ = 1, 1.5); and (4) a large sample size (n = 100), the medium percentage of missing values (15%), the medium value of variance ($\sigma^2$ = 1.5). The R-RQ3 imputation of Al-Omari, Jemain, and Ibrahim was efficient for the situation of large sample size (n = 100), medium percentage of missing values (15%), and medium value of variance ($\sigma^2$ = 1) [1].

In a later year, there was an investigation of the research [15] by [14] of a chain ratio (CR) estimator and a regression estimator with a linear combination of two auxiliary variables that used the supplementary data to increase the accuracy of the estimator. That study proposed a multivariate chain ratio (MCR) estimator and a regression estimator that used a linear combination of two auxiliary variables versus a traditional multivariate ratio (MR) estimator [15]

and a regression estimator that used the information of two auxiliary variables [15], and a chain ratio (CR) estimator that used one auxiliary variable [11]. The evaluation measure was MSE. It was found that the proposed MCR estimator and the proposed regression estimator that used a linear combination of two auxiliary variables were equally highly effective, followed by the traditional MR estimator and the regression estimator that used the data of the two auxiliary variables [15].

Multiple Imputation, Maximum Likelihood, and Fully Bayesian methods are the three most used model-based approaches in missing data problems. Although it is easy to show that when the assumption is missing at random (MAR), the complete case analysis is unbiased and efficient, the methods above are still commonly used in practice for this setting. To examine the performance of relationships between these three methods in this setting, we derive and investigate small samples and asymptotic expressions of the estimates and standard errors. We thoroughly examine how these estimates are related to the three approaches in the linear regression model when the assumption is MAR. They show that when the assumption is MAR in the linear model, the estimates of the regression coefficients using these three methods are asymptotically equivalent to the complete case estimates under general conditions. A simulation of an accurate data set from a liver cancer clinical trial was conducted to compare the properties of these methods when the assumption was MAR [16].

In 2015, a paper proposed imputation estimators when there were missing observations on a dependent variable in multiple linear regression analysis. The proposed estimators were called ratio-Q1 (RQ1), ratio-Q3 (RQ3), regression-ratio-Q1 (R-RQ1) and regression-ratio-Q3 (R-RQ3). In various simulation situations, the efficiencies of the proposed estimations were compared to two existing methods—mean imputation and regression imputation. Mean absolute percentage error (MAPE) was the evaluation measure. For each situation, a multiple linear regression model with two independent variables was considered under the assumption that the error was distributed normally. Variances, sample sizes, and percentages of missing values were varied. Findings revealed that the MAPE from every estimator increased as either the percentage of missing values or the variance of error increased. Moreover, for all situations, R-RQ1, R-RQ3, and regression imputation performed better than the others [17].

In 2017, a study compared the ratio and regression estimators empirically based on bias and coefficient of variation. The author conducted simulation studies based on the sampling rate, population size, heterogeneity of the auxiliary variable X, deviation from linearity, and model misspecification. The study showed that the ratio estimator was better than the regression estimator when the regression line was close to the origin. Ratio and regression estimators worked even if there was a weak linear relationship between X and Y and if there was a minimum model misspecification. The relationship between the target and auxiliary variables was weak, Bootstrap estimated yield lower bias. The regression estimator was generally more efficient than the ratio estimator [18]. In regression analysis, missing covariate data has been a common problem. Many researchers use ad hoc methods to overcome this problem due to the ease of implementation. However, these methods require assumptions about the data that rarely hold in practice. Model-based methods such as Maximum Likelihood (ML) using the expectation maximization (EM) algorithm and Multiple Imputation (MI) are more promising when dealing with difficulties caused by missing data. Then again, inappropriate methods of missing value imputation can lead to serious bias that severely affects the parameter estimates. A simulation study was performed to assess the effects of different missing data techniques on the performance of a regression model. The covariate data were generated using an underlying multivariate normal distribution, and the dependent variable was generated using a combination of explanatory variables. Missing values in covariates were simulated using an assumption called missing at random (MAR). Four levels of missingness (10%, 20%, 30%,

and 40%) were assessed. A linear regression analysis was fitted, and the model performance was measured; MSE, and R-squared were obtained. The study showed that MI was superior in handling missing data with the highest R-squared and the lowest MSE when the percentage of missingness was less than 30%. Both methods could not hold a larger than 30% level of missingness [19].

In 2021, Little reviewed assumptions about missing data mechanisms that underlie methods for the statistical analysis of data with missing values. Little explained Rubin's original definition of missing at random (MAR), its motivation and criticisms, and his sufficient conditions for ignoring the missingness mechanism for likelihood-based, Bayesian, and frequentist inference. Related definitions, including missing completely at random (MCAR), missing at random (MAR), and partially MAR. Little presented a formal argument for weakening Rubin's sufficient conditions for frequentist maximum likelihood and inference with precision based on the observed information. Some simple examples of MAR are described, with an example where the missingness mechanism can be ignored even though MAR does not hold. Alternative approaches to statistical inference based on the likelihood function were reviewed, along with non-likelihood frequentist approaches, including weighted generalized estimating equations. Connections with the causal inference literature were also discussed [20].

Missing data has been a common issue in many domains of study. If this issue was disregarded, an erroneous conclusion might ensue. In 2022, a study developed new imputation methods and compared the efficiency of eight imputation methods: hot deck imputation (HD), k-nearest neighbors imputation (KNN), stochastic regression, imputation (SR), predictive mean matching imputation (PMM), random forest imputation (RF), stochastic regression random forest with equivalent weight imputation (SREW), k-nearest random forest with equivalent weight imputation (KREW), and k-nearest stochastic regression and random forest with equivalent weight imputation (KSREW). Simulations were run using various sample sizes (30, 60, 100, and 150) and missing percentages (10%, 20%, 30%, and 40%). Average mean square error (AMSE) was used to compare the efficiencies. The proposed composite approach outperformed the single ones. On the other hand, increasing the number of components to a four-component method did not affect the imputation performance [21].

In 2022, Isabella Sayers et al. conducted a comparison of five missing value imputation methods in multiple regression model: multiple regression, regression-ratio-Q1 (R-RQ1), regression-ratio-Q3 (R-RQ3), stochastic regression, and k-nearest stochastic regression, with equivalent weighting methods under missing completely at random (MCAR). The sample size, error variance, and missing values percentage varied. The evaluation measure was MSE. The small sample sizes were 20 and 40; the medium sample sizes were 60 and 80, and the large sample sizes were 100 and 120. The percentages of missing values were 5, 10, 15, and 20. The missing values were constructed by a missing-at-random method. The variances of error used in this study were 1, 3, 5, 7, and 9. The research result showed that, for all percentages of missing values, all variances of error, and all sample sizes, most of the R-RQ1 method is the maximum efficiency, followed by the multiple regression method [22].

In 2024, Song and Guo presented a fully informative multiple imputation (fiMI) method. It was based on a linear regression model with a missing response variable, utilizing all observable data to obtain estimates of the regression coefficients and, thereby, the predicted values of the missing response variable. This provided a good explanation of the relationship between the response variable and their respective variables and effectively enhanced the imputation accuracy of the response variable. The stability and sensitivity of the fiMI method were evaluated through a simulation study. Subsequently, the proposed fiMI method was applied to two real data sets: the admission prediction data set and the goalkeeper data set [23].

The studies above inspired the authors to study the use of a ratio estimator that utilizes the 1st and 3rd quartiles [13]. The estimator was used to estimate the missing values in multiple regression analysis with a simple random sampling (SRS). An efficient missing-value imputation method was the regression-ratio quartile1,3 imputation with SRS of Al-Omari, Jemain, and Ibrahim. This method was highly efficient [1,22]. The authors propose three new adjusted methods for comparison: the adjusted regression-chain ratio quartile1,3 imputation with SRS of Kadilar and Cingi from the research [11], the adjusted regression-multivariate ratio quartile1,3 imputation with SRS of Feng, Ni, and Zou from the research [14], which provided highly efficient, and the adjusted regression-multivariate chain ratio quartile1,3 imputation with SRS of Lu from the research [15], which provided the highest efficiency.

In addition, the research compares an estimator of the utilization ratio of the 1st and 3rd quartiles analyzed with a rank set sampling (RSS) (referenced in [18]) versus simple random sampling. It was found that the efficiencies of the utilization ratio estimators of the 1st and 3rd quartiles analyzed with RSS were higher than those analyzed with SRS. Interested in developing further, the authors propose four new adjusted methods for estimating missing values in regression analysis: the adjusted regression-ratio quartile1,3 imputation with RSS of Al-Omari, Gemain, and Ibrahim from the proposed ratio estimator [13]; the adjusted regression-chain ratio quartile1,3 imputation with RSS of Kadilar and Cinji from the proposed chain ratio estimator [11]; the adjusted regression-multivariate ratio quartile1,3 imputation with RSS of Feng, Ni, and Zou from the proposed multivariate ratio estimator [14]; and the adjusted regression-multivariate chain ratio quartile1,3 imputation with RSS of Lu from the proposed multivariate chain ratio estimator [15]. The efficiency measures were mean square error (MSE) and mean absolute percentage error (MAPE). The optimum method was used to select a proper missing value imputation method under different sample sizes, error variances, and percentages of missing values. Moreover, the research also compared the performance of SRS and RSS estimators considering the MSE and MAPE.

Our contributions to the field of statistics are tables suggesting the best missing value imputation method for multiple regression analysis based on sampling method, sample size, missing value percentage, and error variance. In addition, the table suggested a better sampling method for SRS and RSS estimators based on sample sizes and error variances.

## 2. Materials and methods

This research compared new adjusted missing-value imputation methods in multiple regression with simple random sampling and rank-set sampling methods. The measures for comparing missing value imputation were mean square error and mean absolute percentage error. The study comprised many steps, as follows.

2.1 The population data of the independent variables $(X_1, X_2)$ and the error $(\varepsilon)$ were constructed with a size of 100,000 values. This data had a normal distribution with the probability density function,

$$f(x) = \frac{1}{\sigma\sqrt{2\pi}} e^{-\frac{1}{2}\left(\frac{x-\mu}{\sigma}\right)^2}; -\infty < x < \infty, -\infty < \mu < \infty, \sigma^2 > 0. \quad (1)$$

where expected value was $E(X) = \mu$, and variance was $Var(X) = \sigma^2$. The independent variable $X_1$ had parameters $\mu = 3, \sigma^2 = 2.25$. The independent variable $X_2$ had parameters $\mu = 5$, $\sigma^2 = 4$, and the error had parameters $\mu = 0, \sigma^2 = 1, 3, 5, 7, 9$ [1].

2.2 The above independent variables and the error were sampled with simple random sampling (SRS) and rank set sampling (RSS) methods for a small sample size of 20 and 40; a middle sample size of 60 and 80; a large sample size of 100 and 120; a considerable sample size of 200 and 500. For each sampling instance, it was repeated 1,000 times [1].

2.3 The multicollinearity between independent variables was tested whether the independent variables were correlated or not by using the Pearson correlation coefficient. If the correlation coefficient was more significant than or equal to 0.6, return to sampling in Sect 2.2 again. The multicollinearity was tested, and the variables were resampled repeatedly until the correlation coefficient was less than 0.6, indicating that the independent variables were not correlated [24,25]. The construction of the dependent variable ($Y$) is described in Sect 2.5. Multicollinearity can be tested based on VIF and Pearson correlation coefficient. In this research, we used the Pearson correlation coefficient because it was easier to use the values of the independent variables to determine the correlation coefficient without needing to determine the value of the dependent variable's value first.

2.4 The parameter of the y-intercept was set to $\beta_0 = 0.5$, and the regression coefficients were set to $\beta_1 = 1$ and $\beta_2 = -0.3$ to create the dependent variable in Sect 2.5.

2.5 The dependent variable $Y$ was linearly correlated with the independent variables $X_1$ and $X_2$. The y-intercept $\beta_0$, regression coefficient $\beta_1$ and $\beta_1$, and error $\varepsilon$ were constructed using the following linear relationship model,

$$Y_i = \beta_0 + \beta_1 X_{1i} + \beta_2 X_{2i} + \varepsilon_i; i = 1, 2, ..., n, \quad (2)$$

where $Y_i$ was the dependent variable; $X_{1i}$ and $X_{2i}$ were the 1st and 2nd independent variables; $\beta_0$ was the y-intercept or value of $Y$ for $X_{1i} = 0$ and $X_{2i} = 0$, $\beta_1$ and $\beta_2$ were the regression coefficient or slope of the straight line; and $\varepsilon_i$ was the error with $\mu = 0$, and $\sigma^2 = 1, 2, 3, 4$ and 5 [1].

2.6 The independent variable $X_2$ had missing values of 5, 10, 15, 20, 30, and 40 percentages missing completely at random (MCAR) [6,9].

2.7 The number of missing values was calculated, and the positions of missing values were randomized.

2.7.1 The number of missing values of the independent variable $X_2$ was calculated as follows:

Number of missing values = Sample sizes x Missing value percentages

2.7.2 The positions of missing values of the independent variable $X_2$ were randomized and determined as missing completely at random (MCAR). MCAR was assumed in this simulation study. The missing values in $X_2$ were not related to $X_1$. In the simulation, they were simulated independently of each other. The positions of the missing values in $X_2$ were randomized, they were replaced with the series mean of the entire data set in $X_2$. Missing values of 5, 10, 15, 20, 30, and 40% are shown in Table 1.

Sample sizes provided eight values, the proportions of missing values provided six values, and missing value estimation methods provided eight values, which had quite a broad scope in multiple regression analysis with two covariates and MCAR. There were $8x6x8 = 384$ scenarios for each error variance of the sampling method.

2.8 The missing values of the independent variable $X_2$ were calculated using the series mean, which replaced the missing values with the mean of the entire data set in $X_2$ as follows [26]:

$$\hat{X} = \frac{\sum_i^k}{k} X_i, \quad (3)$$

**Table 1. Number of missing values and number of intact data for missing value percentages of 5, 10, 15, 20, 30, and 40 in small, middle, large, and huge sample sizes.**

| | Missing value percentages | | | | | | | | | | | |
|---|---|---|---|---|---|---|---|---|---|---|---|---|
| | 5 | | 10 | | 15 | | 20 | | 30 | | 40 | |
| Sample sizes (n) | No. of missing values | No. of intact data | No. of missing values | No. of intact data | No. of missing values | No. of intact data | No. of missing values | No. of intact data | No. of missing values | No. of intact data | No. of missing values | No. of intact data |
| Small | | | | | | | | | | | | |
| 20 | 1 | 19 | 2 | 18 | 3 | 17 | 4 | 16 | 6 | 14 | 8 | 12 |
| 40 | 2 | 38 | 4 | 36 | 6 | 34 | 8 | 32 | 12 | 28 | 16 | 24 |
| Middle | | | | | | | | | | | | |
| 60 | 3 | 57 | 6 | 54 | 9 | 51 | 12 | 48 | 18 | 42 | 24 | 36 |
| 80 | 4 | 76 | 8 | 72 | 12 | 68 | 16 | 64 | 24 | 56 | 32 | 48 |
| Large | | | | | | | | | | | | |
| 100 | 5 | 95 | 10 | 90 | 15 | 85 | 20 | 80 | 30 | 70 | 40 | 60 |
| 120 | 6 | 114 | 12 | 108 | 18 | 102 | 24 | 96 | 36 | 84 | 48 | 72 |
| Huge | | | | | | | | | | | | |
| 200 | 10 | 190 | 20 | 180 | 30 | 170 | 40 | 160 | 60 | 140 | 80 | 120 |
| 500 | 25 | 475 | 50 | 450 | 75 | 425 | 100 | 400 | 150 | 350 | 200 | 300 |


where $X_i$ was the data value with available values; $k$ was the amount of data with available values; and $\hat{X}$ was the estimate of the missing value predicted from the mean of the available values.

2.9 In Sect 2.8, the estimates of the missing values of the independent variable $X_2$ were placed in the positions of the missing values.

2.10 The regression-ratio imputations were replaced in all eight methods as follows:

2.10.1 Regression-ratio quartile1,3 imputation with SRS of Al-Omari, Jemain, and Ibrahim (R-RQ1,3SRS)

Regression-ratio quartile1 imputation (R-RQ1) was a ratio imputation estimation method applied by replacing $\hat{y}_{reg}$ to $\bar{y}$. This method uses advantage of the 1st quartile of variables $X_1$ and $X_2$ [1]. Regression-ratio quartile3 imputation (R-RQ3) is described in the same way.

2.10.1.1 Multiple regression imputation mechanisms of MCAR

Regression-ratio quartile1,3 imputation with SRS of Al-Omari, Jemain, and Ibrahim relies on the imputation of the multiple regression $\hat{y}_{reg}$ into $\bar{y}$ for the missing value estimator presented in Sect 2.10.1.4, as follows:

1) The data $X_{2,r+1}, \ldots, X_{2n}$, the missing values of the independent variables, were replaced with $X_{2,r+1,new}, \ldots, X_{2n,new}$. The estimate of the dependent variable $\hat{Y}_i$ was calculated using the missing value estimation with multiple regression analysis. 2) The independent variable $X_{1i}$ with complete information and the independent variable $X_{2i,new}$ with replaced missing values provided the estimation of the dependent variable $\hat{Y}_i$ as follows: $\hat{Y}_i = \hat{\beta}_0 + \hat{\beta}_1 X_{1i} + \hat{\beta}_2 X_{2i}$.

2.10.1.2 Ratio estimator with simple random sampling

The estimators, Eqs (10) and (11), that we used in this study, were derived ultimately from the ratio estimator with simple random sampling (R-SRS). Here, we show the derivation steps from the equation of R-SRS to Eqs (10) and (11) in the subsections below. Simple random sampling is a sampling of a population of size N, where each possible sample of size n had the same probability of being selected. The sample obtained at such sampling is called a simple random sample [2].

The ratio estimator ($\hat{\mu}_Y$) is a method for estimating $\mu_Y$ with $\hat{\mu}_Y$ based on the relationship between the auxiliary variable $X$ (e.g. sample mean of $\bar{x}$) and the dependent variable $Y$ (e.g. sample mean of $\bar{y}$) under the condition that the population mean $\mu_X$ is a known parameter. We defined it under simple random sampling as follows [13]:

$$\hat{\mu}_{YSRS} = \mu_X \left( \frac{\bar{y}_{SRS}}{\bar{x}_{SRS}} \right), \tag{4}$$

where $\hat{\mu}_{YSRS}$ was a ratio estimate, $\bar{x}_{SRS} = \frac{\sum_i^n}{n} x_i$ and $\bar{y}_{SRS} = \frac{\sum_i^n}{n} y_i$ were the sample mean of the independent variable $X_2$ and the dependent variable $Y$, respectively.

2.10.1.3 Ratio estimator using quartile1,3 (RQ1,3)

AI-Omari et al. (2009) proposed the following ratio estimator [13],

$$\hat{\mu}_{YR} = \bar{y}_{SRS} \left( \frac{\mu_X}{\bar{x}_{SRS}} \right). \tag{5}$$

In addition, they also proposed a ratio estimator for estimating the population mean of the dependent variable by taking advantage of the 1st and 3rd quartiles of the independent variable $X_1$ and $X_2$. The calculation formula is the following,

$$\hat{\mu}_{YRQ1} = \bar{y}_{SRS} \left( \frac{\mu_X + q_1}{\bar{x}_{SRS} + q_1} \right), \tag{6}$$

and

$$\hat{\mu}_{YRQ3} = \bar{y}_{SRS}\left(\frac{\mu_X + q_3}{\bar{x}_{SRS} + q_3}\right), \tag{7}$$

where $\hat{\mu}_{YRQ1}$ was an estimate of the mean using quartile1, $\hat{\mu}_{YRQ3}$ was an estimate of the mean using quartile3; $\bar{x}_{SRS}$ and $\bar{y}_{SRS}$ were the sample mean of the independent variable $X_2$ and the dependent variable $Y$, respectively; $q_1$ and $q_3$ were quartile1 and 3 of the population of the independent variable $X_2$, respectively.

2.10.1.4 Missing value estimator

In Eqs (6) and (7), Jomprapan (2012) [1] substitute $\bar{x}_{in}$ for $\mu_X$ since it is difficult to obtain the population mean. Therefore, the sample mean $\bar{x}_{in}$ is instead of the population mean $\mu_X$ and the sample mean $\bar{x}_{ir}$ was instead of the sample mean $\bar{x}_{SRS}$. Two estimators of missing value were obtained as Eqs (8) and ((9), which are called ratio quartile1 imputation with simple random sampling (RQ1SRS) and ratio quartile3 imputation with simple random sampling (RQ3SRS),

$$\hat{y}_{RQ1SRS} = \bar{y}\prod_{i=1}^{k}\left(\frac{\bar{x}_{in} + q_1}{\bar{x}_{ir} + q1}\right), \tag{8}$$

and

$$\hat{y}_{RQ3SRS} = \bar{y}\prod_{i=1}^{k}\left(\frac{\bar{x}_{in} + q_3}{\bar{x}_{ir} + q3}\right), \tag{9}$$

where $\hat{y}_{RQ1SRS}$ was a ratio quartile1 estimate, $\hat{y}_{RQ3SRS}$ was a ratio quartile3 estimate, $\bar{x}_{in}$ was a sample mean of the independent variable $X_2$ when the data was complete, $\bar{x}_{ir}$ was a sample mean of the independent variable $X_2$ without accounting for the missing values, $\bar{y}$ was a mean of the complete dependent variable $Y$ and $q_1$, $q_3$ were quartile1, 3 of the samples of the random variable $X_2$, respectively [1,17].

However, using only one estimator ($\bar{y}$) for all missing values may not be appropriate because using a constant for all missing values causes the estimator's variance to be underestimated. Therefore, she took the estimate from the multiple regression analysis, $\hat{y}_{reg}$ substituted it in $\bar{y}$ in Eqs (8) and (9), resulting in Eqs (10) and (11), which are called regression-ratio quartile1 imputation with a simple random sampling of Al-Omari, Jemain, and Ibrahim (R-RQ1SRS) and regression-ratio quartile3 imputation with a simple random sampling of Al-Omari, Jemain, and Ibrahim (R-RQ3SRS),

$$\hat{y}_{R-RQ1SRS} = \hat{y}_{reg}\prod_{i=1}^{k}\left(\frac{\bar{x}_{in} + q_1}{\bar{x}_{ir} + q1}\right), \tag{10}$$

and

$$\hat{y}_{R-RQ3SRS} = \hat{y}_{reg}\prod_{i=1}^{k}\left(\frac{\bar{x}_{in} + q_3}{\bar{x}_{ir} + q3}\right), \tag{11}$$

where $\hat{y}_{R-RQ1SRS}$ was the regression-ratio quartile1 imputation estimate with a simple random sampling; $\hat{y}_{R-RQ3SRS}$ was the regression-ratio quartile3 imputation estimate with a simple

random sampling; and $\hat{y}_{reg}$ was the value of the dependent variable $Y$ for multiple regression analysis.

The $\hat{y}_{R-RQ1SRS}$ and $\hat{y}_{R-RQ3SRS}$ were calculated from Eqs (10) and (11), respectively, which were substituted in the same position as the missing value of the independent variable $X_2$ for the estimable dependent variable $\hat{y}_i$ in Sect 2.10.1.1.

The authors propose three adaptation methods, based on ideas from [11,14,15], taking advantage of the 1st and 3rd quartiles and $\hat{y}_{reg}$ was obtained by imputation with multiple regression analysis to replace $\bar{y}$ using simple random sampling as follows in Sect 2.10.2–2.10.4.

2.10.2 Adjusted regression-chain ratio quartile1,3 imputation with SRS of Kadilar and Cingi (AR-CRQ1,3SRS)

Kadilar and Cingi (2003) proposed the following chain ratio estimator [11],

$$\hat{\mu}_{YCR} = \bar{y}_{SRS}\left(\frac{\mu_X}{\bar{x}_{SRS}}\right)^{\alpha}. \tag{12}$$

In addition to the two methods proposed in Sect 2.10.1.4 by [1], the authors propose another technique, a new adaptation of the chain ratio method [11], for estimating missing values, that took advantage of the 1st and 3rd quartiles of variables $X_2$. The technique is the following Equations,

$$\hat{\mu}_{YCRQ1} = \bar{y}_{SRS}\left(\frac{\mu_X + q_1}{\bar{x}_{SRS} + q_1}\right)^{\alpha}, \tag{13}$$

and

$$\hat{\mu}_{YCRQ3} = \bar{y}_{SRS}\left(\frac{\mu_X + q_3}{\bar{x}_{SRS} + q_3}\right)^{\alpha}, \tag{14}$$

The authors also propose using $\hat{y}_{reg}$ to replace $\bar{y}$ in Eqs (13) and (14). Hence, $\mu_X$ was substituted by $\bar{x}_{in}$ since obtaining the population mean was difficult. Also, $\bar{x}_{ir}$ was used instead of $\bar{x}_{SRS}$. Two estimators of missing values were obtained as Eqs (15) and (16). These are called the adjusted regression-chain ratio quartile1 imputation with a simple random sampling of Kadilar and Cingi (AR-CRQ1SRS) and the adjusted regression-chain ratio quartile3 imputation with a simple random sampling of Kadilar and Cingi (AR-CRQ3SRS),

$$\hat{y}_{AR-CRQ1SRS} = \hat{y}_{reg}\prod_{i=1}^{k}\left(\frac{\bar{x}_{in} + q_1}{\bar{x}_{ir} + q_1}\right)^{\alpha}, \tag{15}$$

and

$$\hat{y}_{AR-CRQ3SRS} = \hat{y}_{reg}\prod_{i=1}^{k}\left(\frac{\bar{x}_{in} + q_3}{\bar{x}_{ir} + q_3}\right)^{\alpha}, \tag{16}$$

where $\hat{y}_{AR-CRQ1SRS}$ was the adjusted regression-chain ratio quartile1 imputation estimate with a simple random sampling; $\hat{y}_{AR-CRQ3SRS}$ was the adjusted regression-chain ratio quartile3 imputation estimate with a simple random sampling; and $\alpha$ was any constant value. In this case, we set $\alpha = 0.10, 0.30, 0.50, 0.70, 0.90$. The experiment showed that $\alpha = 0.90$ provided the best estimation performance.

2.10.3 Adjusted regression-multivariate ratio quartile1,3 imputation with SRS of Feng, Ni, and Zou (AR-MRQ1,3SRS)

Feng et al. (1998) proposed a multivariate ratio estimator [14],

$$\hat{\mu}_{YMR} = \varepsilon \bar{y}_{SRS} \left( \frac{\mu_X}{\bar{x}_{SRS}} \right). \tag{17}$$

The authors propose a new adaptation of the multivariate ratio method of [14]. The adapted methods estimated missing values by taking advantage of the 1st and 3rd quartiles of variables $X_1$ and $X_2$,

$$\hat{\mu}_{YMRQ1} = \varepsilon \bar{y}_{SRS} \left( \frac{\mu_X + q_1}{\bar{x}_{SRS} + q_1} \right), \tag{18}$$

and

$$\hat{\mu}_{YMRQ3} = \varepsilon \bar{y}_{SRS} \left( \frac{\mu_X + q_3}{\bar{x}_{SRS} + q_3} \right). \tag{19}$$

In addition, the authors also propose using $\hat{y}_{reg}$ to replace $\bar{y}$ in Eqs (18) and (19) and replacing $\mu_X$ with $\bar{x}_{in}$. Also, $\bar{x}_{ir}$ replaced $\bar{x}_{SRS}$, resulting in two missing value estimators defined by Eqs (20) and (21). These are called the adjusted regression-multivariate ratio quartile1 imputation with a simple random sampling of Feng, Ni, and Zou (AR-MRQ1SRS) and the adjusted regression-multivariate ratio quartile3 imputation with a simple random sampling of Feng, Ni, and Zou (AR-MRQ3SRS),

$$\hat{y}_{AR-MRQ1SRS} = \hat{y}_{reg} [\varepsilon_1 \left( \frac{\bar{x}_{1n} + q_1}{\bar{x}_{1r} + q_1} \right) + \varepsilon_2 \left( \frac{\bar{x}_{2n} + q_1}{\bar{x}_{2r} + q_1} \right)], \tag{20}$$

and

$$\hat{y}_{AR-MRQ3SRS} = \hat{y}_{reg} [\varepsilon_1 \left( \frac{\bar{x}_{1n} + q_3}{\bar{x}_{1r} + q_3} \right) + \varepsilon_2 \left( \frac{\bar{x}_{2n} + q_3}{\bar{x}_{2r} + q_3} \right)], \tag{21}$$

where $\hat{y}_{AR-MRQ1SRS}$ was the adjusted regression-multivariate ratio quartile1 imputation estimate with a simple random sampling; $\hat{y}_{AR-MRQ3SRS}$ was the adjusted regression-multivariate ratio quartile3 imputation estimate with a simple random sampling; and $\varepsilon_1, \varepsilon_2$ were the weighted values, and $\varepsilon_1 + \varepsilon_2 = 1$. In this case, we set $\varepsilon_1 = 0.10, 0.30, 0.50, 0.70, 0.90$ and $\varepsilon_2 = 0.10, 0.30, 0.50, 0.70, 0.90$. By trial and error, $\varepsilon_1 = 0.9$ and $\varepsilon_2 = 0.10$ were found to give the best estimation performance.

2.10.4 Adjusted regression-multivariate chain ratio quartile1,3 imputation with SRS of Lu (AR-MCRQ1,3SRS)

Lu (2013) proposed a multivariate chain ratio estimator [15],

$$\hat{\mu}_{YMCR} = \bar{y}_{SRS} \left( \frac{\omega_1 \bar{x}_{1n} + \omega_2 \bar{x}_{2n}}{\omega_1 \bar{x}_{1r} + \omega_2 \bar{x}_{2r}} \right)^{\alpha}. \tag{22}$$

The authors propose a new adaptation of the multivariate chain ratio method [15]. This adaptation estimated the missing values by taking advantage of the 1st and 3rd quartiles of

variables $X_1$ and $X_2$,

$$\hat{\mu}_{YMCRQ1} = \bar{y}_{SRS}\left(\frac{\omega_1\left(\bar{x}_{1n}+q_1\right)+\omega_2\left(\bar{x}_{2n}+q_1\right)}{\omega_1\left(\bar{x}_{1r}+q_1\right)+\omega_2\left(\bar{x}_{2r}+q_1\right)}\right)^{\alpha}, \tag{23}$$

and

$$\hat{\mu}_{YMCRQ3} = \bar{y}_{SRS}\left(\frac{\omega_1\left(\bar{x}_{1n}+q_3\right)+\omega_2\left(\bar{x}_{2n}+q_3\right)}{\omega_1\left(\bar{x}_{1r}+q_3\right)+\omega_2\left(\bar{x}_{2r}+q_3\right)}\right)^{\alpha}. \tag{24}$$

In addition, the authors also propose using $\hat{y}_{reg}$ to replace $\bar{y}$ in Eqs (23) and (24), resulting in two missing value estimators defined by Eqs (25) and (26). These are called the adjusted regression-multivariate chain ratio quartile1 imputation with a simple random sampling of Lu (AR-MCRQ1SRS) and the adjusted regression-multivariate chain ratio quartile3 imputation with a simple random sampling of Lu (AR-MCRQ3SRS),

$$\hat{y}_{AR-MCRQ1SRS} = \hat{y}_{reg}\left(\frac{\omega_1\left(\bar{x}_{1n}+q_1\right)+\omega_2\left(\bar{x}_{2n}+q_1\right)}{\omega_1\left(\bar{x}_{1r}+q_1\right)+\omega_2\left(\bar{x}_{2r}+q_1\right)}\right)^{\alpha}, \tag{25}$$

and

$$\hat{y}_{AR-MCRQ3SRS} = \hat{y}_{reg}\left(\frac{\omega_1\left(\bar{x}_{1n}+q_3\right)+\omega_2\left(\bar{x}_{2n}+q_3\right)}{\omega_1\left(\bar{x}_{1r}+q_3\right)+\omega_2\left(\bar{x}_{2r}+q_3\right)}\right)^{\alpha}, \tag{26}$$

where $\hat{y}_{AR-MCRQ1SRS}$ was the adjusted regression-multivariate chain ratio quartile1 imputation estimate with a simple random sampling; $\hat{y}_{AR-MCRQ3SRS}$ was the adjusted regression-multivariate chain ratio quartile3 imputation estimate with a simple random sampling; $\alpha$ was any constant value; and $\omega_1, \omega_2$ were the weighted values, and $\omega_1 + \omega_2 = 1$. We set $\alpha = 0.10, 0.30, 0.50, 0.70, 0.90$; $\omega_1 = 0.10, 0.30, 0.50, 0.70, 0.90$, and $\omega_2 = 0.10, 0.30, 0.50, 0.70, 0.90$. The experiment showed that $\alpha = 0.10$, $\omega_1 = 0.10$, and $\omega_2 = 0.90$ gave the best estimation performance.

According to the research of [13], a ratio estimator was proposed by taking advantage of the 1st and 3rd quartiles with a rank set sampling (RSS) (referenced in [18]) compared with a simple random sampling (SRS). It was found that the ratio estimator of the 1st and 3rd quartiles with RSS had higher efficiency than SRS. Therefore, the authors adopted this concept and proposed four additional adjustment methods for estimating missing value in the regression analysis: [11,13–15] by utilization of the 1st and 3rd quartiles and using $\hat{y}_{reg}$ which was $\hat{y}$ obtained by imputation of the regression method to replace $\bar{y}$ by RSS as follows:

2.10.5 Adjusted regression-ratio quartile1,3 imputation with RSS of Al-Omari, Jemain, and Ibrahim (AR-RQ1,3RSS)

2.10.5.1 Multiple regression imputation

This imputation was similar to Sect 2.10.1.1, regression-ratio quartile1,3 imputation with RSS of Al-Omari, Jemain, and Ibrahim relies on the imputation the multiple regression, $\hat{y}_{reg}$ into $\bar{y}$ in the missing value estimator presented in Sect 2.10.5.3.

2.10.5.2 Ratio estimator with a rank set sampling (R-RSS)

A rank set sampling method followed the steps below [13,18].

1) The first data set of size n units was randomized. Then, the smallest value of the independent variable $X$, which was correlated to the value of the dependent variable $Y$, was selected.

2) The second data set of size n units was randomized. Then, the second smallest value of the independent variable, which was correlated to the value of the dependent variable, was selected, and so on.

3) The nth data set of size n units was randomized. Then, the nth most significant value of the independent variable, which was correlated to the value of the dependent variable, was selected.

The selected data sets were used to estimate the missing values. A rank set sampling estimator $\mu_Y$ was based on the relationship between the auxiliary variable $X$ and the dependent variable $Y$, assuming that the population mean $\mu_X$ was a known parameter. We could define it under rank set sampling as follows [13],

$$\hat{\mu}_{YRSS} = \mu_X \left( \frac{\bar{y}_{RSS}}{\bar{x}_{RSS}} \right), \tag{27}$$

where $\hat{\mu}_{YRSS}$ was a ratio estimate; $\bar{x}_{RSS} = \frac{\sum_i^n}{n} x_i$ and $\bar{y}_{RSS} = \frac{\sum_i^n}{n} y_i$ were the sample mean of the independent variable $X_2$ and the dependent variable $Y$, respectively.

2.10.5.3 Proposed missing value estimator

AI-Omari et al. (2009) proposed the following ratio estimator [13],

$$\hat{\mu}_{YR} = \bar{y}_{RSS} \left( \frac{\mu_X}{\bar{x}_{RSS}} \right). \tag{28}$$

In addition, they also proposed a ratio estimator to estimate the population mean of the dependent variable by taking advantage of the 1st and 3rd quartiles of the variables $X_1$ and $X_2$. The calculation formula is the following,

$$\hat{\mu}_{YRQ1} = \bar{y}_{RSS} \left( \frac{\mu_X + q_1}{\bar{x}_{RSS} + q_1} \right), \tag{29}$$

and

$$\hat{\mu}_{YRQ3} = \bar{y}_{RSS} \left( \frac{\mu_X + q_3}{\bar{x}_{RSS} + q_3} \right), \tag{30}$$

where $\hat{\mu}_{YRQ1}$ was an estimate of the mean using quartile1, $\hat{\mu}_{YRQ3}$ was an estimate of the mean using quartile3; $\bar{x}_{RSS}$ and $\bar{y}_{RSS}$ were sample mean of the independent variable $X_2$ and the dependent variable $Y$, respectively; $q_1$ and $q_3$ were quartile1 and 3 of the population of the independent variable $X_2$, respectively.

The authors propose a new adjusted ratio method of [13] using $\hat{y}_{reg}$, which was $\hat{y}$ obtained by imputation with the multiple regression method, to replace $\bar{y}$ in Eqs (29) and (30). The population mean ($\mu_X$) was substituted by the sample mean ($\bar{x}_{in}$) since obtaining the population mean was difficult. The sample mean ($\bar{x}_{ir}$) was substituted into the sample mean ($\bar{x}_{RSS}$), resulting in two missing value estimators defined in Eqs (31) and (32) below. We called these estimators an adjusted regression-ratio quartile1 imputation with a rank set sampling of Al-Omari, Jemain, and Ibrahim (AR-RQ1RSS) and an adjusted regression-ratio quartile3 imputation with a rank set sampling of Al-Omari, Jemain, and Ibrahim (AR-RQ3RSS),

$$\hat{y}_{AR-RQ1RSS} = \hat{y}_{reg} \prod_{i=1}^{k} \left( \frac{\bar{x}_{in} + q_1}{\bar{x}_{ir} + q_1} \right), \tag{31}$$

and

$$\hat{y}_{AR-RQ3RSS} = \hat{y}_{reg} \prod_{i=1}^{k} \left( \frac{\bar{x}_{in} + q_3}{\bar{x}_{ir} + q_3} \right), \tag{32}$$

where $\hat{y}_{AR-RQ1RSS}$ was the adjusted regression-ratio quartile1 imputation estimate with a rank set random sampling; $\hat{y}_{AR-RQ3RSS}$ was the adjusted regression-ratio quartile3 imputation estimate with a rank set sampling; and $\hat{y}_{reg}$ was the value of the dependent variable $Y$.

2.10.6 Adjusted regression-chain ratio quartile1,3 imputation with RSS of Kadilar and Cingi (AR-CRQ1,3RSS)

Kadilar and Cingi (2003) proposed the following chain ratio estimator [11],

$$\hat{\mu}_{YCR} = \bar{y}_{RSS} \left( \frac{\mu_X}{\bar{x}_{RSS}} \right)^{\alpha}. \tag{33}$$

The authors propose a new adaptation of the chain ratio method of [11], for estimating values that take advantage of the 1st and 3rd quartiles of variables $X_1$ and $X_2$. The technique is the following Equatios,

$$\hat{\mu}_{YCRQ1} = \bar{y}_{RSS} \left( \frac{\mu_X + q_1}{\bar{x}_{RSS} + q_1} \right)^{\alpha}, \tag{34}$$

and

$$\hat{\mu}_{YCRQ3} = \bar{y}_{RSS} \left( \frac{\mu_X + q_3}{\bar{x}_{RSS} + q_3} \right)^{\alpha}. \tag{35}$$

In addition, the authors also propose using $\hat{y}_{reg}$ to replace $\bar{y}$ in Eqs (34) and (35). Hence, $\bar{x}_{in}$ substituted $\mu_X$ since it was difficult to obtain the population mean. Also, $\bar{x}_{ir}$ was used instead of $\bar{x}_{RSS}$. Two estimators of missing values were obtained as Eqs (36) and (37). These are called the adjusted regression-chain ratio quartile1 imputation with a rank set sampling of Kadilar and Cingi (AR-CRQ1RSS) and the adjusted regression-chain ratio quartile3 imputation with a simple random sampling of Kadilar and Cingi (AR-CRQ3RSS),

$$\hat{y}_{AR-CRQ1RSS} = \hat{y}_{reg} \prod_{i=1}^{k} \left( \frac{\bar{x}_{in} + q_1}{\bar{x}_{ir} + q_1} \right)^{\alpha}, \tag{36}$$

and

$$\hat{y}_{AR-CRQ1RSS} = \hat{y}_{reg} \prod_{i=1}^{k} \left( \frac{\bar{x}_{in} + q_3}{\bar{x}_{ir} + q_3} \right)^{\alpha}, \tag{37}$$

where $\hat{y}_{AR-CRQ1RRS}$ was the adjusted regression-chain ratio quartile1 imputation estimate with a rank set sampling; $\hat{y}_{AR-CRQ3RSS}$ was the adjusted regression-chain ratio quartile3 imputation estimate with a rank set sampling; and $\alpha$ was any constant value. In this case, we set $\alpha$ = 0.10, 0.30, 0.50, 0.70, 0.90. The experiment showed that $\alpha = 0.90$ provided the best estimation performance.

2.10.7 Adjusted regression-multivariate ratio quartile1,3 imputation with RSS of Feng, Ni, and Zou (AR-MRQ1,3RSS)

Feng et al. (1998) proposed a multivariate ratio estimator [14],

$$\hat{\mu}_{YMR} = \varepsilon \bar{y}_{RSS} \left( \frac{\mu_X}{\bar{x}_{RSS}} \right). \tag{38}$$

The authors propose a new adaptation of the multivariate ratio method [14]. The adapted methods estimated missing values by taking advantage of the 1st and 3rd quartiles of variables $X_1$ and $X_2$,

$$\hat{\mu}_{YMRQ1} = \varepsilon \bar{y}_{RSS} \left( \frac{\mu_X + q_1}{\bar{x}_{RSS} + q_1} \right), \tag{39}$$

and

$$\hat{\mu}_{YMRQ3} = \varepsilon \bar{y}_{RSS} \left( \frac{\mu_X + q_3}{\bar{x}_{RSS} + q_3} \right). \tag{40}$$

In addition, the authors also propose using $\hat{y}_{reg}$ to replace $\bar{y}$ in Eqs (39) and (40) and replacing $\mu_X$ with $\bar{x}_{in}$. Also, $\bar{x}_{ir}$ replaced $\bar{x}_{RSS}$, resulting in two missing value estimators defined by Eqs (41) and (42). These are called the adjusted regression-multivariate ratio quartile1 imputation with a rank set sampling of Feng, Ni, and Zou (AR-MRQ1RSS) and the adjusted regression-multivariate ratio quartile3 imputation with a rank set sampling of Feng, Ni, and Zou (AR-MRQ3RSS),

$$\hat{y}_{AR-MRQ1RSS} = \hat{y}_{reg}[\varepsilon_1 \left( \frac{\bar{x}_{1n} + q_1}{\bar{x}_{1r} + q_1} \right) + \varepsilon_2 \left( \frac{\bar{x}_{2n} + q_1}{\bar{x}_{2r} + q_1} \right)], \tag{41}$$

and

$$\hat{y}_{AR-MRQ3RSS} = \hat{y}_{reg}[\varepsilon_1 \left( \frac{\bar{x}_{1n} + q_3}{\bar{x}_{1r} + q_3} \right) + \varepsilon_2 \left( \frac{\bar{x}_{2n} + q_3}{\bar{x}_{2r} + q_3} \right)], \tag{42}$$

where $\hat{y}_{AR-MRQ1RSS}$ was the adjusted regression-multivariate ratio quartile1 imputation estimate with a rank set sampling; $\hat{y}_{AR-MRQ3RRS}$ was the adjusted regression-multivariate ratio quartile3 imputation estimate with a rank set sampling; and $\varepsilon_1, \varepsilon_2$ were the weighted values, and $\varepsilon_1 + \varepsilon_2 = 1$. In this case, we set $\varepsilon_1 = 0.10, 0.30, 0.50, 0.70, 0.90$ and $\varepsilon_2 = 0.10, 0.30, 0.50, 0.70, 0.90$. By trial and error, $\varepsilon_1 = 0.9$ and $\varepsilon_2 = 0.10$ were found to give the best estimation performance.

2.10.8 Adjusted regression-multivariate chain ratio quartile1,3 imputation with RSS of Lu (AR-MCRQ1,3RSS)

Lu (2013) proposed a multivariate chain ratio estimator [15],

$$\hat{\mu}_{YMCR} = \bar{y}_{RSS} \left( \frac{\omega_1 \bar{x}_{1n} + \omega_2 \bar{x}_{2n}}{\omega_1 \bar{x}_{1r} + \omega_2 \bar{x}_{2r}} \right)^{\alpha}. \tag{43}$$

The authors propose a new adaptation of the multivariate chain ratio method [15]. This adaptation estimated the missing values by taking advantage of the 1st and 3rd quartiles of variables $X_1$ and $X_2$,

$$\hat{\mu}_{YMCRQ1} = \bar{y}_{RSS} \left( \frac{\omega_1 \left( \bar{x}_{1n} + q_1 \right) + \omega_2 \left( \bar{x}_{2n} + q_1 \right)}{\omega_1 \left( \bar{x}_{1r} + q_1 \right) + \omega_2 \left( \bar{x}_{2r} + q_1 \right)} \right)^{\alpha}, \tag{44}$$

and

$$\hat{\mu}_{YMCRQ3} = \bar{y}_{RSS}\left(\frac{\omega_1\left(\bar{x}_{1n}+q_3\right)+\omega_2\left(\bar{x}_{2n}+q_3\right)}{\omega_1\left(\bar{x}_{1r}+q_3\right)+\omega_2\left(\bar{x}_{2r}+q_3\right)}\right)^{\alpha}. \tag{45}$$

In addition, the authors also propose using $\hat{y}_{reg}$ to replace $\bar{y}$ in Eqs (44) and (45), resulting in two missing value estimators defined by Eqs (46) and (47). These are called the adjusted regression-multivariate chain ratio quartile1 imputation with a rank set sampling of Lu (AR-MCRQ1RSS) and the adjusted regression-multivariate chain ratio quartile3 imputation with a rank set sampling of Lu (AR-MCRQ3RSS),

$$\hat{y}_{AR-MCRQ1RSS} = \hat{y}_{reg}\left(\frac{\omega_1\left(\bar{x}_{1n}+q_1\right)+\omega_2\left(\bar{x}_{2n}+q_1\right)}{\omega_1\left(\bar{x}_{1r}+q_1\right)+\omega_2\left(\bar{x}_{2r}+q_1\right)}\right)^{\alpha}, \tag{46}$$

and

$$\hat{y}_{AR-MCRQ3RSS} = \hat{y}_{reg}\left(\frac{\omega_1\left(\bar{x}_{1n}+q_3\right)+\omega_2\left(\bar{x}_{2n}+q_3\right)}{\omega_1\left(\bar{x}_{1r}+q_3\right)+\omega_2\left(\bar{x}_{2r}+q_3\right)}\right)^{\alpha}, \tag{47}$$

where $\hat{y}_{AR-MCRQ1RSS}$ was the adjusted regression-multivariate chain ratio quartile1 imputation estimate with a rank set sampling; $\hat{y}_{AR-MCRQ3SRS}$ was the adjusted regression-multivariate chain ratio quartile3 imputation estimate with a rank set sampling; $\alpha$ was any constant value; and $\omega_1, \omega_2$ were the weighted values, and $\omega_1 + \omega_2 = 1$. We set $\alpha = 0.10, 0.30, 0.50, 0.70, 0.90$; $\omega_1 = 0.10, 0.30, 0.50, 0.70, 0.90$, and $\omega_2 = 0.10, 0.30, 0.50, 0.70, 0.90$. The experiment showed that $\alpha = 0.10$, $\omega_1 = 0.10$, and $\omega_2 = 0.90$ gave the best estimation performance.

2.11 The true value of the dependent variable ($y$) generated from Sect 2.5 and the estimated value (or predicted value) of the dependent variable ($\hat{y}_i$) obtained from Sect 2.10 were used to calculate the mean squared error and the mean absolute percentage error.

2.12 Comparison of the eight regression-ratio imputation methods was based on mean squared error and mean absolute percentage error.

2.12.1 Mean Square Error (MSE) [1,27–29].

$$MSE = \frac{1}{n}\sum_{i=1}^{n}\left(y_i - \hat{y}_i\right)^2, \tag{48}$$

where $y_i$ was the actual value of the i th dependent variable; $\hat{y}_i$ was the i th predicted value of the dependent variable; and MSE was the mean square error of the i th predicted value of the dependent variable.

2.12.2 Mean Absolute Percentage Error (MAPE) [1,17].

$$MAPE = \frac{1}{n}\sum_{i=1}^{n}\frac{|y_i - \hat{y}_i|}{|y_i|}X100, \tag{49}$$

where MAPE was the mean absolute percentage error of the predicted value of the dependent variable.

2.13 R Studio Version 4.2.1 program was used to simulate each experimental scenario. For each sampling method, the simulation was repeated for 1,000 cycles [1,17].

2.14 The actual data collected for multiple regression analysis are the following.

Two precious metal loss data set, gold and platinum were collected. The loss was in the production process of a jewelry company. The three independent variables were the total weight of the specimen ($X_1$), the total output weight of the specimen ($X_2$), and the recovery value ($X_3$). The dependent variable was the loss of precious metals ($Y$). The sample size used was 12 sample units.

## 3. Results

### 3.1. Results of a study on simulated data

Mean square error for various simulation scenarios are the following: error variances of 1, 3, 5, 7, and 9; sample sizes of 20, 40, 60, 80, 100, 120, 200, and 500; and missing value percentages of 5, 10, 15, 20, 30 and 40%, with SRS in missing value estimation methods as shown in Tables 2, 3, 4, 5 and 6.

As shown in Tables 2 and 3 for error variance of 1 and 3, most of the sample size and missing value percentage, the AR-MRQ1 method achieved the minimum mean square error, followed by AR-MRQ3. On the other hand, as can be observed in Tables 4, 5, and 6, for error variances of 5, 7, and 9, for every sample size and missing value percentage, the AR-MCRQ3 method achieved the minimum mean square error, followed by AR-MCRQ1.

Mean absolute percentage error for various simulation scenarios are the following: error variances of 1, 3, 5, 7, and 9; sample sizes of 20, 40, 60, 80, 100, 120, 200 and 500; and missing value percentages of 5, 10, 15, 20, 30, and 40%, with SRS in missing value estimation methods as shown in Tables 7, 8, 9, 10, and 11. As shown in Tables 7, 8, 9, 10, and 11, for error variance of 1, 3, 5, 7, and 9, every sample size and missing value percentage, AR-MCRQ1 method achieved the minimum mean absolute percentage error, followed by AR-MCRQ3.

Mean square error for various simulation scenarios are the following: error variances of 1, 3, 5, 7, and 9; sample sizes of 20, 40, 60, 80, 100, 120, 200, and 500; and missing value percentages of 5, 10, 15, 20, 30, and 40%, with RSS in missing value estimation methods as shown in Tables 12, 13, 14, 15, and 16.

As shown in Table 12, for an error variance of 1, for every sample size and missing value percentage, the AR-MRQ1 method achieved the minimum mean square error, followed by AR-MRQ3. On the other hand, as shown in Tables 13, 14, and 15, for error variances of 3, 5, and 7, every sample size and missing value percentage, the AR-MCRQ3 method achieved the minimum mean square error, followed by AR-MCRQ1. As shown in Table 16, for an error variance of 9 and most of the sample sizes and missing value percentage, the AR-MCRQ1 method achieved the minimum mean square error, followed by AR-MCRQ3.

Mean absolute percentage error for various simulation scenarios are the following: error variances of 1, 3, 5, 7, and 9; sample sizes of 20, 40, 60, 80, 100, 120, 200, and 500; and missing value percentages of 5, 10, 15, 20, 30, and 40% with RSS in missing value estimation methods as shown in Tables 17, 18, 19, 20, and 21.

As shown in Table 17, for error variance of 1, for most of the sample size and missing value percentages, the AR-MRQ1 method achieved the minimum mean square error, followed by AR-MRQ3. On the other hand, as shown in Tables 19, 20, and 21, for error variances of 5, 7, and 9, most of the sample size and missing value percentage, the AR-MCRQ1 method achieved the minimum mean absolute percentage error, followed by AR-MCRQ3. As shown in Table 18, for an error variance of 3 and most of the sample sizes and missing value percentage, the AR-MCRQ3 method achieved the minimum mean square error, followed by AR-MCRQ1.

**Table 2. Mean square errors for the scenario where error variance was 1; sample sizes of 20, 40, 60, 80, 100, 120, 200, and 500; and missing value percentages of 5, 10, 15, 20, 30, and 40% with SRS in missing value estimation methods.**

| Sample sizes | Missing values | Listwise delition | Missing value estimation methods | | | | | | | |
|---|---|---|---|---|---|---|---|---|---|---|
| | | | R-RQ1 | R-RQ3 | AR-CRQ1 | AR-CRQ3 | AR-MRQ1 | AR-MRQ3 | AR-MCRQ1 | AR-MCRQ3 |
| 20 | 5% | 1.0668 | 1.0045 | 1.0044 | 1.0045 | 1.0044 | **0.9952** | 0.9952 | 1.0715 | **1.0693** |
| | 10% | 1.0768 | 1.0015 | 1.0019 | 1.0018 | 1.0021 | **0.9951** | 0.9951 | **1.1698** | **1.1655** |
| | 15% | 1.0960 | 1.0130 | 1.0141 | 1.0136 | 1.0147 | **1.0076** | 1.0076 | **1.2620** | **1.2555** |
| | 20% | 1.1148 | 1.0265 | 1.0269 | 1.0267 | 1.0274 | **1.0240** | **1.0239** | **1.3676** | **1.3593** |
| | 30% | 1.1380 | 1.0741 | 1.0703 | 1.0716 | 1.0692 | 1.0669 | 1.0669 | 1.5727 | **1.5604** |
| | 40% | 1.1416 | 1.1249 | 1.1069 | 1.1131 | 1.0994 | 1.0848 | 1.0848 | 1.7653 | **1.7482** |
| 40 | 5% | 1.0807 | 1.0135 | 1.0137 | 1.0136 | 1.0137 | **1.0077** | **1.0077** | 1.0898 | **1.0877** |
| | 10% | 1.1067 | 1.0174 | 1.0181 | 1.0178 | 1.0185 | **1.0129** | 1.0129 | **1.1867** | **1.1823** |
| | 15% | 1.1101 | 1.0113 | 1.0122 | 1.0118 | 1.0127 | **1.0093** | 1.0093 | **1.2767** | **1.2701** |
| | 20% | 1.1482 | 1.0498 | 1.0501 | 1.0499 | 1.0505 | **1.0488** | **1.0488** | **1.3978** | **1.3894** |
| | 30% | 1.1532 | 1.0974 | 1.0918 | 1.0937 | 1.0897 | 1.0866 | 1.0866 | **1.6051** | **1.5926** |
| | 40% | 1.1510 | 1.1431 | 1.1233 | 1.1299 | 1.1147 | 1.0989 | 1.0989 | **1.7815** | **1.7646** |
| 60 | 5% | 1.0785 | 1.0122 | 1.0125 | 1.0124 | 1.0126 | **1.0073** | 1.0073 | **1.0899** | **1.0877** |
| | 10% | 1.1018 | 1.0140 | 1.0148 | 1.0145 | 1.0152 | **1.0105** | 1.0105 | **1.1816** | **1.1773** |
| | 15% | 1.1249 | 1.0322 | 1.0329 | 1.0326 | 1.0333 | **1.0309** | 1.0309 | **1.2940** | **1.2876** |
| | 20% | 1.1425 | 1.0510 | 1.0508 | 1.0508 | 1.0510 | **1.0501** | **1.0501** | **1.3978** | **1.3894** |
| | 30% | 1.1565 | 1.1040 | 1.0977 | 1.0998 | 1.0953 | 1.0917 | 1.0917 | **1.6159** | **1.6034** |
| | 40% | 1.1531 | 1.1556 | 1.1342 | 1.1412 | 1.1247 | 1.1055 | 1.1055 | **1.7997** | **1.7826** |
| 80 | 5% | 1.0666 | 1.0031 | 1.0033 | 1.0032 | 1.0034 | **0.9990** | 0.9990 | 1.0866 | **1.0844** |
| | 10% | 1.1029 | 1.0149 | 1.0157 | 1.0154 | 1.0161 | **1.0115** | 1.0115 | **1.1839** | **1.1795** |
| | 15% | 1.1171 | 1.0163 | 1.0171 | 1.0167 | 1.0176 | **1.0153** | 1.0153 | **1.2799** | **1.2735** |
| | 20% | 1.1349 | 1.0391 | 1.0389 | 1.0389 | 1.0390 | **1.0384** | **1.0384** | **1.3888** | **1.3804** |
| | 30% | 1.1531 | 1.1056 | 1.0989 | 1.1011 | 1.0963 | 1.0924 | 1.0924 | **1.6175** | **1.6051** |
| | 40% | 1.1564 | 1.1598 | 1.1382 | 1.1453 | 1.1286 | 1.1090 | 1.1090 | **1.8004** | **1.7834** |
| 100 | 5% | 1.0750 | 1.0085 | 1.0088 | 1.0087 | 1.0089 | **1.0045** | 1.0045 | **1.0894** | **1.0872** |
| | 10% | 1.1002 | 1.0124 | 1.0131 | 1.0128 | 1.0135 | **1.0096** | 1.0096 | **1.1834** | **1.1791** |
| | 15% | 1.1226 | 1.0202 | 1.0210 | 1.0207 | 1.0216 | **1.0192** | 1.0192 | **1.2774** | **1.2710** |
| | 20% | 1.1357 | 1.0391 | 1.0391 | 1.0390 | 1.0393 | **1.0386** | 1.0386 | **1.3874** | **1.3791** |
| | 30% | 1.1512 | 1.1002 | 1.0936 | 1.0958 | 1.0911 | 1.0874 | 1.0874 | **1.6094** | **1.5970** |
| | 40% | 1.1621 | 1.1665 | 1.1452 | 1.1521 | 1.1357 | 1.1168 | 1.1168 | **1.8004** | **1.7837** |
| 120 | 5% | 1.0713 | 1.0042 | 1.0045 | 1.0043 | 1.0046 | **1.0002** | 1.0002 | **1.0845** | **1.0823** |
| | 10% | 1.1019 | 1.0149 | 1.0155 | 1.0152 | 1.0159 | **1.0126** | 1.0126 | **1.1897** | **1.1854** |
| | 15% | 1.1217 | 1.0238 | 1.0245 | 1.0242 | 1.0249 | **1.0231** | 1.0231 | **1.2841** | **1.2778** |
| | 20% | 1.1364 | 1.0410 | 1.0408 | 1.0408 | 1.0409 | **1.0404** | **1.0404** | **1.3885** | **1.3802** |
| | 30% | 1.1554 | 1.1072 | 1.1003 | 1.1026 | 1.0976 | 1.0936 | 1.0936 | **1.6220** | **1.6095** |
| | 40% | 1.1496 | 1.1621 | 1.1400 | 1.1472 | 1.1300 | 1.1087 | 1.1087 | **1.8017** | **1.7849** |
| 200 | 5% | 1.0782 | 1.0147 | 1.0150 | 1.0148 | 1.0151 | 1.0111 | 1.0112 | **1.0973** | **1.0951** |
| | 10% | 1.1011 | 1.0073 | 1.0081 | 1.0078 | 1.0086 | 1.0047 | 1.0047 | **1.1768** | **1.1725** |
| | 15% | 1.1199 | 1.0203 | 1.0210 | 1.0207 | 1.0215 | 1.0197 | 1.0197 | **1.2777** | **1.2714** |
| | 20% | 1.1436 | 1.0495 | 1.0490 | 1.0491 | 1.0491 | 1.0488 | 1.0488 | **1.4027** | **1.3943** |
| | 30% | 1.1558 | 1.1041 | 1.0972 | 1.0995 | 1.0945 | 1.0907 | 1.0907 | **1.6154** | **1.6029** |
| | 40% | 1.1604 | 1.1693 | 1.1470 | 1.1542 | 1.1369 | 1.1157 | 1.1157 | **1.8083** | **1.7915** |
| 500 | 5% | 1.0736 | 1.0066 | 1.0070 | 1.0069 | 1.0072 | 1.0033 | 1.0033 | **1.0883** | **1.0861** |
| | 10% | 1.1072 | 1.0125 | 1.0133 | 1.0130 | 1.0138 | 1.0102 | 1.0102 | **1.1847** | **1.1804** |
| | 15% | 1.1264 | 1.0247 | 1.0254 | 1.0251 | 1.0260 | 1.0242 | 1.0242 | **1.2844** | **1.2781** |
| | 20% | 1.1410 | 1.0465 | 1.0461 | 1.0462 | 1.0462 | 1.0460 | 1.0460 | **1.3967** | **1.3883** |
| | 30% | 1.1493 | 1.1008 | 1.0935 | 1.0960 | 1.0906 | 1.0863 | 1.0863 | **1.6152** | **1.6028** |
| | 40% | 1.1625 | 1.1697 | 1.1472 | 1.1544 | 1.1370 | 1.1160 | 1.1161 | **1.8098** | **1.7929** |

Bold values indicate the best method from comparing the four methods with SRS. The listwise deletion method is a control condition.


**Table 3. Mean square errors for the scenario where error variance was 3; sample sizes of 20, 40, 60, 80, 100, 120, 200, and 500; and missing value percentages of 5, 10, 15, 20, 30, and 40% with SRS in missing value estimation methods.**

| Sample sizes | Missing values | Listwise delition | Missing value estimation methods | | | | | | | |
|---|---|---|---|---|---|---|---|---|---|---|
| | | | R-RQ1 | R-RQ3 | AR-CRQ1 | AR-CRQ3 | AR-MRQ1 | AR-MRQ3 | AR-MCRQ1 | AR-MCRQ3 |
| 20 | 5% | 3.0359 | 2.9873 | 2.9871 | 2.9872 | 2.9871 | **2.9596** | 2.9596 | 2.9648 | **2.9631** |
| | 10% | 3.0744 | 2.9892 | 2.9918 | 2.9907 | 2.9931 | **2.955** | 2.955 | **2.9832** | **2.9798** |
| | 15% | 3.1144 | 2.9918 | 2.9986 | 2.9959 | 3.0021 | **2.9645** | 2.9645 | **3.0037** | **2.9987** |
| | 20% | 3.0696 | 2.9091 | 2.9205 | 2.9159 | 2.9266 | **2.9025** | **2.9025** | **2.972** | **2.9653** |
| | 30% | 3.1096 | 2.8693 | 2.8901 | 2.882 | 2.9024 | 2.9294 | 2.9294 | 3.0255 | **3.0159** |
| | 40% | 3.1651 | 2.8222 | 2.8448 | 2.836 | 2.8608 | 2.9695 | 2.9696 | 3.1041 | **3.0909** |
| 40 | 5% | 3.0582 | 2.9893 | 2.9899 | 2.9897 | 2.9902 | **2.9692** | **2.9692** | 2.9888 | **2.9871** |
| | 10% | 3.092 | 2.9843 | 2.9878 | 2.9863 | 2.9896 | **2.9596** | 2.9596 | **2.9965** | **2.993** |
| | 15% | 3.1522 | 3.0213 | 3.0286 | 3.0256 | 3.0323 | **3.0071** | 3.0071 | **3.0781** | **3.0729** |
| | 20% | 3.1167 | 2.9457 | 2.9577 | 2.9529 | 2.9642 | **2.9502** | **2.9502** | **3.0289** | **3.0223** |
| | 30% | 3.1449 | 2.8948 | 2.9154 | 2.9073 | 2.9278 | 2.9655 | 2.9655 | **3.0691** | **3.0594** |
| | 40% | 3.1574 | 2.7963 | 2.8191 | 2.8102 | 2.8356 | 2.9602 | 2.9603 | **3.0835** | **3.0703** |
| 60 | 5% | 3.0767 | 3.0084 | 3.0094 | 3.009 | 3.0098 | **2.9881** | 2.9881 | **2.9996** | **2.9979** |
| | 10% | 3.1042 | 2.9956 | 2.9995 | 2.9979 | 3.0015 | **2.9729** | 2.9729 | **3.0033** | **3** |
| | 15% | 3.1057 | 2.9591 | 2.9671 | 2.9638 | 2.9712 | **2.9471** | 2.9471 | **3.0027** | **2.9977** |
| | 20% | 3.1323 | 2.9514 | 2.9645 | 2.9591 | 2.9715 | **2.9602** | **2.9602** | **3.0222** | **3.0157** |
| | 30% | 3.1348 | 2.8869 | 2.9071 | 2.8991 | 2.9194 | 2.9595 | 2.9596 | **3.0661** | **3.0564** |
| | 40% | 3.1296 | 2.7752 | 2.7963 | 2.7878 | 2.8117 | 2.9345 | 2.9345 | **3.078** | **3.0646** |
| 80 | 5% | 3.0844 | 3.0168 | 3.0177 | 3.0173 | 3.0182 | **2.9977** | 2.9977 | 3.0139 | **3.0121** |
| | 10% | 3.109 | 3.0007 | 3.0047 | 3.0031 | 3.0067 | **2.9795** | 2.9795 | **3.0133** | **3.0099** |
| | 15% | 3.1026 | 2.9578 | 2.9658 | 2.9625 | 2.9699 | **2.9473** | 2.9473 | **3.0073** | **3.0022** |
| | 20% | 3.1388 | 2.9565 | 2.9692 | 2.964 | 2.9761 | **2.9665** | **2.9665** | **3.0416** | **3.0351** |
| | 30% | 3.1396 | 2.8808 | 2.9011 | 2.893 | 2.9134 | 2.9556 | 2.9557 | **3.0676** | **3.0578** |
| | 40% | 3.1638 | 2.7933 | 2.8151 | 2.8065 | 2.8312 | 2.96 | 2.9601 | **3.0886** | **3.0754** |
| 100 | 5% | 3.0747 | 3.0037 | 3.0047 | 3.0043 | 3.0052 | **2.9851** | 2.9851 | **2.9994** | **2.9977** |
| | 10% | 3.097 | 2.9872 | 2.9913 | 2.9896 | 2.9933 | **2.9667** | 2.9668 | **3.0018** | **2.9985** |
| | 15% | 3.1132 | 2.9702 | 2.9784 | 2.975 | 2.9827 | **2.9606** | 2.9606 | **3.012** | **3.0071** |
| | 20% | 3.1348 | 2.9528 | 2.9654 | 2.9602 | 2.9723 | **2.9637** | 2.9637 | **3.0398** | **3.0332** |
| | 30% | 3.1458 | 2.8906 | 2.9113 | 2.903 | 2.9238 | 2.9676 | 2.9676 | **3.0689** | **3.0593** |
| | 40% | 3.1528 | 2.7938 | 2.8149 | 2.8066 | 2.8305 | 2.9573 | 2.9574 | **3.0879** | **3.0748** |
| 120 | 5% | 3.0503 | 2.9814 | 2.9824 | 2.982 | 2.9829 | **2.964** | 2.9641 | **2.9826** | **2.9809** |
| | 10% | 3.0987 | 2.9875 | 2.9916 | 2.9899 | 2.9936 | **2.9679** | 2.9679 | **3.0062** | **3.0028** |
| | 15% | 3.0983 | 2.9611 | 2.9691 | 2.9658 | 2.9732 | **2.9523** | 2.9523 | **3.0077** | **3.0028** |
| | 20% | 3.144 | 2.9695 | 2.9822 | 2.977 | 2.9891 | **2.9811** | **2.9811** | **3.0518** | **3.0454** |
| | 30% | 3.1569 | 2.9097 | 2.93 | 2.9219 | 2.9424 | 2.9862 | 2.9862 | **3.096** | **3.0862** |
| | 40% | 3.135 | 2.7846 | 2.8046 | 2.7967 | 2.8196 | 2.9441 | 2.9442 | **3.0906** | **3.0774** |
| 200 | 5% | 3.0633 | 2.9904 | 2.9916 | 2.9911 | 2.9921 | 2.9733 | 2.9733 | **2.9919** | **2.9901** |
| | 10% | 3.0984 | 2.983 | 2.9872 | 2.9854 | 2.9893 | 2.964 | 2.964 | **2.9963** | **2.993** |
| | 15% | 3.1228 | 2.975 | 2.9832 | 2.9798 | 2.9875 | 2.9673 | 2.9673 | **3.0208** | **3.0159** |
| | 20% | 3.146 | 2.9647 | 2.9776 | 2.9724 | 2.9847 | 2.9778 | 2.9779 | **3.0499** | **3.0434** |
| | 30% | 3.1576 | 2.9065 | 2.9266 | 2.9185 | 2.9389 | 2.9836 | 2.9837 | **3.0931** | **3.0835** |
| | 40% | 3.1584 | 2.7994 | 2.8198 | 2.8117 | 2.8351 | 2.9632 | 2.9632 | **3.1019** | **3.0887** |
| 500 | 5% | 3.0737 | 2.9991 | 3.0003 | 2.9998 | 3.0009 | 2.9817 | 2.9818 | **2.996** | **2.9943** |
| | 10% | 3.105 | 2.9849 | 2.9894 | 2.9875 | 2.9916 | 2.9666 | 2.9667 | **2.9983** | **2.995** |
| | 15% | 3.1402 | 2.9834 | 2.9921 | 2.9885 | 2.9966 | 2.9768 | 2.9769 | **3.0264** | **3.0215** |
| | 20% | 3.1438 | 2.9565 | 2.9695 | 2.9642 | 2.9765 | 2.971 | 2.9711 | **3.0435** | **3.037** |
| | 30% | 3.1521 | 2.893 | 2.9134 | 2.9053 | 2.9258 | 2.9728 | 2.973 | **3.0807** | **3.071** |
| | 40% | 3.1561 | 2.7965 | 2.8167 | 2.8087 | 2.832 | 2.9617 | 2.9619 | **3.1012** | **3.088** |


**Table 4. Mean square errors for the scenario where error variance was 5; sample sizes of 20, 40, 60, 80, 100, 120, 200 and 500; and missing value percentages of 5, 10, 15, 20, 30, and 40% with SRS in missing value estimation methods.**

| Sample sizes | Missing values | Listwise delition | Missing value estimation methods | | | | | | | |
|---|---|---|---|---|---|---|---|---|---|---|
| | | | R-RQ1 | R-RQ3 | AR-CRQ1 | AR-CRQ3 | AR-MRQ1 | AR-MRQ3 | AR-MCRQ1 | AR-MCRQ3 |
| 20 | 5% | 5.07 | 5.0133 | 5.0131 | 5.0132 | 5.013 | **4.9689** | 4.9689 | 4.916 | **4.9148** |
| | 10% | 5.0714 | 4.9882 | 4.9921 | 4.9905 | 4.9941 | **4.9344** | 4.9344 | **4.8476** | **4.8451** |
| | 15% | 5.1046 | 4.9505 | 4.9632 | 4.9579 | 4.9694 | **4.9028** | 4.9028 | **4.7376** | **4.7339** |
| | 20% | 5.1802 | 4.9433 | 4.967 | 4.9572 | 4.9792 | **4.933** | **4.933** | **4.7003** | **4.6957** |
| | 30% | 5.0881 | 4.6723 | 4.718 | 4.7002 | 4.744 | 4.8004 | 4.8004 | 4.4839 | **4.477** |
| | 40% | 5.1664 | 4.4592 | 4.5265 | 4.5009 | 4.5681 | 4.8103 | 4.8103 | 4.3564 | **4.3471** |
| 40 | 5% | 5.0605 | 4.998 | 4.9991 | 4.9986 | 4.9996 | **4.9615** | **4.9615** | 4.9077 | **4.9064** |
| | 10% | 5.0876 | 4.9624 | 4.9688 | 4.9661 | 4.9719 | **4.917** | 4.917 | **4.818** | **4.8154** |
| | 15% | 5.137 | 4.9601 | 4.975 | 4.9689 | 4.9826 | **4.9311** | 4.9311 | **4.7733** | **4.7696** |
| | 20% | 5.1254 | 4.8718 | 4.8975 | 4.8871 | 4.911 | **4.8831** | **4.8831** | **4.6574** | **4.6527** |
| | 30% | 5.1812 | 4.7267 | 4.7748 | 4.7557 | 4.8019 | 4.8809 | 4.8809 | **4.5581** | **4.5513** |
| | 40% | 5.2041 | 4.4825 | 4.5474 | 4.5233 | 4.5888 | 4.8523 | 4.8523 | **4.4208** | **4.4113** |
| 60 | 5% | 5.0534 | 4.9783 | 4.9799 | 4.9792 | 4.9806 | **4.9418** | 4.9419 | **4.8798** | **4.8786** |
| | 10% | 5.0885 | 4.9505 | 4.9576 | 4.9546 | 4.9611 | **4.9086** | 4.9086 | **4.8004** | **4.798** |
| | 15% | 5.1211 | 4.9384 | 4.9535 | 4.9473 | 4.9612 | **4.9158** | 4.9158 | **4.7634** | **4.7597** |
| | 20% | 5.1182 | 4.8517 | 4.8774 | 4.8671 | 4.8911 | **4.8696** | **4.8696** | **4.6526** | **4.6479** |
| | 30% | 5.1339 | 4.6857 | 4.7322 | 4.714 | 4.7588 | 4.842 | 4.8421 | **4.5492** | **4.542** |
| | 40% | 5.1131 | 4.3994 | 4.4628 | 4.439 | 4.5032 | 4.7681 | 4.7682 | **4.3559** | **4.3463** |
| 80 | 5% | 5.0823 | 4.9989 | 5.0007 | 4.9999 | 5.0016 | **4.9629** | 4.9629 | 4.9006 | **4.8993** |
| | 10% | 5.1139 | 4.973 | 4.9805 | 4.9773 | 4.9842 | **4.9324** | 4.9324 | **4.8197** | **4.8172** |
| | 15% | 5.1185 | 4.9131 | 4.9287 | 4.9222 | 4.9366 | **4.893** | 4.893 | **4.7407** | **4.737** |
| | 20% | 5.1429 | 4.8677 | 4.8939 | 4.8833 | 4.9078 | **4.8889** | **4.8889** | **4.6705** | **4.6658** |
| | 30% | 5.1603 | 4.7025 | 4.7501 | 4.7316 | 4.7775 | 4.8663 | 4.8663 | **4.552** | **4.5451** |
| | 40% | 5.1406 | 4.4166 | 4.4802 | 4.4565 | 4.521 | 4.7922 | 4.7923 | **4.3752** | **4.3655** |
| 100 | 5% | 5.05 | 4.9767 | 4.9785 | 4.9778 | 4.9794 | **4.9444** | 4.9444 | **4.8926** | **4.8913** |
| | 10% | 5.0848 | 4.9574 | 4.9647 | 4.9617 | 4.9684 | **4.9197** | 4.9197 | **4.8149** | **4.8125** |
| | 15% | 5.1508 | 4.9521 | 4.9679 | 4.9614 | 4.9759 | **4.9336** | 4.9337 | **4.7755** | **4.772** |
| | 20% | 5.1363 | 4.8755 | 4.9006 | 4.8905 | 4.914 | **4.8976** | 4.8976 | **4.7019** | **4.6972** |
| | 30% | 5.1292 | 4.6841 | 4.7308 | 4.7126 | 4.7577 | 4.8464 | 4.8465 | **4.5425** | **4.5357** |
| | 40% | 5.1918 | 4.4588 | 4.5242 | 4.4999 | 4.566 | 4.845 | 4.845 | **4.3999** | **4.3906** |
| 120 | 5% | 5.0791 | 5.0062 | 5.0081 | 5.0073 | 5.009 | **4.9724** | 4.9724 | **4.9113** | **4.9101** |
| | 10% | 5.1243 | 4.9862 | 4.9938 | 4.9906 | 4.9976 | **4.9489** | 4.9489 | **4.8439** | **4.8414** |
| | 15% | 5.107 | 4.9138 | 4.9295 | 4.923 | 4.9375 | **4.8967** | 4.8968 | **4.7442** | **4.7407** |
| | 20% | 5.1528 | 4.8842 | 4.9097 | 4.8994 | 4.9232 | **4.9079** | **4.9079** | **4.7106** | **4.7059** |
| | 30% | 5.1519 | 4.6843 | 4.7329 | 4.7139 | 4.7609 | 4.8542 | 4.8543 | **4.535** | **4.528** |
| | 40% | 5.121 | 4.4068 | 4.4694 | 4.4462 | 4.5097 | 4.7816 | 4.7816 | **4.3696** | **4.3601** |
| 200 | 5% | 5.0898 | 5.0105 | 5.0125 | 5.0116 | 5.0134 | 4.9805 | 4.9805 | **4.9335** | **4.9322** |
| | 10% | 5.1154 | 4.9796 | 4.9873 | 4.9841 | 4.9911 | 4.9444 | 4.9444 | **4.8403** | **4.8378** |
| | 15% | 5.136 | 4.9408 | 4.9571 | 4.9504 | 4.9654 | 4.9255 | 4.9256 | **4.7621** | **4.7586** |
| | 20% | 5.1414 | 4.869 | 4.8948 | 4.8844 | 4.9085 | 4.8953 | 4.8954 | **4.6997** | **4.6949** |
| | 30% | 5.1534 | 4.6887 | 4.7362 | 4.7176 | 4.7635 | 4.8576 | 4.8577 | **4.5507** | **4.5437** |
| | 40% | 5.1211 | 4.4016 | 4.464 | 4.4408 | 4.5042 | 4.7784 | 4.7786 | **4.3747** | **4.365** |
| 500 | 5% | 5.0789 | 5.0009 | 5.0031 | 5.0022 | 5.0041 | 4.9703 | 4.9704 | **4.9151** | **4.9138** |
| | 10% | 5.1268 | 4.9898 | 4.9978 | 4.9945 | 5.0018 | 4.9562 | 4.9563 | **4.8494** | **4.8469** |
| | 15% | 5.1254 | 4.9246 | 4.9411 | 4.9343 | 4.9496 | 4.9117 | 4.9118 | **4.7487** | **4.7452** |
| | 20% | 5.1459 | 4.8709 | 4.8972 | 4.8866 | 4.9112 | 4.9004 | 4.9006 | **4.6939** | **4.6892** |
| | 30% | 5.1658 | 4.7035 | 4.751 | 4.7325 | 4.7784 | 4.8758 | 4.876 | **4.5697** | **4.5627** |
| | 40% | 5.1493 | 4.4198 | 4.4827 | 4.4594 | 4.5234 | 4.8035 | 4.8038 | **4.389** | **4.3794** |


**Table 5. Mean square errors for the scenario where error variance was 7; sample sizes of 20, 40, 60, 80, 100, 120, 200 and 500; and missing value percentages of 5, 10, 15, 20, 30, and 40% with SRS in missing value estimation methods.**

| Sample sizes | Missing values | Listwise delition | Missing value estimation methods | | | | | | | |
|---|---|---|---|---|---|---|---|---|---|---|
| | | | R-RQ1 | R-RQ3 | AR-CRQ1 | AR-CRQ3 | AR-MRQ1 | AR-MRQ3 | AR-MCRQ1 | AR-MCRQ3 |
| 20 | 5% | 7.1099 | 7.0545 | 7.0544 | 7.0544 | 7.0543 | **6.9904** | 6.9904 | **6.8634** | **6.8626** |
| | 10% | 7.0627 | 6.9451 | 6.9512 | 6.9486 | 6.9542 | **6.8639** | 6.8639 | **6.6327** | **6.6311** |
| | 15% | **6.9822** | 6.7743 | 6.7921 | 6.7845 | 6.8008 | **6.7091** | 6.7091 | **6.3703** | **6.3679** |
| | 20% | 7.0432 | **6.7587** | 6.7919 | 6.778 | 6.8088 | **6.7451** | **6.7451** | **6.2704** | **6.2675** |
| | 30% | 7.2062 | **6.5903** | 6.6618 | 6.6342 | 6.7019 | 6.7901 | 6.7901 | **6.0479** | **6.0436** |
| | 40% | 7.2022 | **6.1587** | 6.2664 | 6.2261 | 6.3316 | 6.6983 | 6.6983 | **5.7046** | **5.6986** |
| 40 | 5% | 7.0911 | 7.029 | 7.0306 | 7.03 | 7.0314 | **6.9766** | **6.9766** | **6.8514** | **6.8507** |
| | 10% | 7.0315 | 6.8913 | 6.9003 | 6.8965 | 6.9047 | **6.8273** | 6.8273 | **6.5934** | **6.5917** |
| | 15% | 7.0286 | 6.8062 | 6.8274 | 6.8187 | 6.8382 | **6.7648** | 6.7648 | **6.4116** | **6.4092** |
| | 20% | 7.1659 | **6.821** | 6.8582 | 6.8432 | 6.8777 | **6.8379** | **6.838** | **6.3466** | **6.3437** |
| | 30% | 7.2192 | **6.5727** | 6.6469 | 6.6178 | 6.6884 | 6.8076 | 6.8077 | **6.0744** | **6.0702** |
| | 40% | 7.1174 | **6.0265** | 6.1339 | 6.0951 | 6.201 | 6.609 | 6.609 | **5.6049** | **5.599** |
| 60 | 5% | 7.1319 | 7.0543 | 7.0566 | 7.0556 | 7.0577 | **7.0013** | 7.0013 | **6.8667** | **6.8659** |
| | 10% | 7.1549 | 7.0023 | 7.0124 | 7.0082 | 7.0174 | **6.9412** | 6.9412 | **6.6945** | **6.6929** |
| | 15% | 7.1351 | 6.8942 | 6.9168 | 6.9075 | 6.9282 | **6.8606** | 6.8607 | **6.4938** | **6.4916** |
| | 20% | 7.1836 | **6.8282** | 6.8673 | 6.8515 | 6.8878 | **6.8557** | **6.8558** | **6.3498** | **6.347** |
| | 30% | 7.1156 | **6.4784** | 6.5524 | 6.5234 | 6.5939 | 6.7219 | 6.722 | **6.001** | **5.9968** |
| | 40% | 7.12 | **6.0185** | 6.1273 | 6.0866 | 6.194 | 6.6098 | 6.6098 | **5.6232** | **5.6171** |
| 80 | 5% | 7.0743 | 6.9964 | 6.9988 | 6.9978 | 7 | **6.948** | **6.948** | **6.8254** | **6.8246** |
| | 10% | 7.0863 | 6.9479 | 6.9581 | 6.9539 | 6.9632 | **6.892** | 6.892 | **6.6551** | **6.6535** |
| | 15% | 7.0993 | 6.8668 | 6.8895 | 6.8801 | 6.901 | **6.8375** | 6.8375 | **6.4808** | **6.4786** |
| | 20% | 7.1073 | **6.752** | 6.7905 | 6.7749 | 6.8107 | **6.7834** | **6.7835** | **6.3031** | **6.3003** |
| | 30% | 7.1654 | **6.5152** | 6.5895 | 6.5608 | 6.6317 | 6.766 | 6.7661 | **6.0415** | **6.0372** |
| | 40% | 7.1406 | **6.042** | 6.1481 | 6.1091 | 6.2141 | 6.6311 | 6.6312 | **5.6643** | **5.6582** |
| 100 | 5% | 7.1004 | 7.0167 | 7.0193 | 7.0182 | 7.0205 | **6.9695** | 6.9695 | **6.8464** | **6.8456** |
| | 10% | 7.1808 | 7.0355 | 7.0463 | 7.0418 | 7.0517 | **6.9796** | 6.9796 | **6.7283** | **6.7269** |
| | 15% | 7.1337 | 6.8828 | 6.9067 | 6.8968 | 6.9188 | **6.8549** | 6.855 | **6.4746** | **6.4725** |
| | 20% | 7.1447 | **6.7967** | 6.8343 | 6.8191 | 6.8541 | **6.83** | 6.83 | **6.3715** | **6.3685** |
| | 30% | 7.2589 | **6.5751** | 6.6519 | 6.6222 | 6.6954 | 6.8366 | 6.8367 | **6.0833** | **6.0791** |
| | 40% | 7.1469 | **6.0595** | 6.166 | 6.1267 | 6.232 | 6.6507 | 6.6508 | **5.676** | **5.6702** |
| 120 | 5% | 7.0733 | 6.9932 | 6.9958 | 6.9947 | 6.997 | **6.9473** | 6.9474 | **6.8263** | **6.8255** |
| | 10% | 7.0984 | 6.944 | 6.9549 | 6.9504 | 6.9603 | **6.89** | 6.8901 | **6.6475** | **6.6459** |
| | 15% | 7.1241 | 6.8901 | 6.9134 | 6.9038 | 6.9253 | **6.8645** | 6.8645 | **6.4998** | **6.4976** |
| | 20% | 7.1579 | **6.808** | 6.8467 | 6.8311 | 6.8671 | **6.8442** | **6.8442** | **6.3613** | **6.3585** |
| | 30% | 7.1285 | **6.4725** | 6.5468 | 6.5179 | 6.5889 | 6.7273 | 6.7274 | **6.0123** | **6.0079** |
| | 40% | 7.1691 | **6.0732** | 6.1803 | 6.1411 | 6.247 | 6.6727 | 6.6728 | **5.6867** | **5.6809** |
| 200 | 5% | 7.0838 | 6.9999 | 7.0027 | 7.0015 | 7.004 | **6.9553** | 6.9553 | **6.8356** | **6.8348** |
| | 10% | 7.0504 | 6.8936 | 6.9049 | 6.9002 | 6.9106 | **6.8412** | 6.8413 | **6.5932** | **6.5917** |
| | 15% | **7.1243** | 6.8789 | 6.9031 | 6.8932 | 6.9155 | 6.856 | 6.8561 | **6.4773** | **6.4752** |
| | 20% | 7.121 | 6.7562 | **6.7952** | 6.7795 | 6.8158 | 6.7961 | 6.7962 | **6.3241** | **6.3211** |
| | 30% | 7.1718 | **6.5023** | 6.5777 | 6.5484 | 6.6205 | 6.7647 | 6.7648 | **6.0315** | **6.0274** |
| | 40% | 7.1896 | **6.0734** | 6.1806 | 6.1414 | 6.2476 | 6.6793 | 6.6795 | **5.6957** | **5.6897** |
| 500 | 5% | 7.1041 | 7.0201 | 7.0231 | 7.0218 | 7.0246 | **6.9759** | 6.976 | **6.8494** | **6.8486** |
| | 10% | 7.1159 | 6.9526 | 6.964 | 6.9593 | 6.9697 | **6.9037** | 6.9039 | **6.662** | **6.6604** |
| | 15% | 7.1084 | 6.8592 | 6.8833 | 6.8734 | 6.8956 | **6.8404** | 6.8406 | **6.4779** | **6.4756** |
| | 20% | 7.1465 | **6.7808** | 6.8203 | 6.8044 | 6.8413 | 6.8251 | 6.8255 | **6.343** | **6.3402** |
| | 30% | 7.135 | **6.4708** | 6.5454 | 6.5166 | 6.5879 | 6.7355 | 6.7359 | **6.0156** | **6.0113** |
| | 40% | 7.1735 | **6.0708** | 6.1775 | 6.1385 | 6.2442 | 6.679 | 6.6795 | **5.6979** | **5.692** |


**Table 6. Mean square errors for the scenario where error variance was 9; sample sizes of 20, 40, 60, 80, 100, 120, 200 and 500; and missing value percentages of 5, 10, 15, 20, 25 and 30% with SRS in missing value estimation methods.**

| Sample sizes | Missing values | Listwise delition | Missing value estimation methods | | | | | | | |
|---|---|---|---|---|---|---|---|---|---|---|
| | | | R-RQ1 | R-RQ3 | AR-CRQ1 | AR-CRQ3 | AR-MRQ1 | AR-MRQ3 | AR-MCRQ1 | AR-MCRQ3 |
| 20 | 5% | 9.1658 | 9.1155 | 9.1154 | 9.1155 | 9.1153 | **9.0266** | 9.0266 | 8.8134 | **8.8131** |
| | 10% | 9.1575 | 9.0251 | 9.0337 | 9.0301 | 9.0379 | **8.9099** | 8.9099 | **8.5144** | **8.5137** |
| | 15% | 9.2442 | 9.024 | 9.0476 | 9.0379 | 9.0595 | **8.9339** | 8.9339 | **8.3545** | **8.3536** |
| | 20% | 9.1821 | 8.8066 | 8.8512 | 8.833 | 8.8741 | **8.789** | **8.789** | **8.0242** | **8.0231** |
| | 30% | 9.0696 | 8.2922 | 8.388 | 8.3517 | 8.442 | 8.5609 | 8.5609 | 7.379 | **7.3778** |
| | 40% | 9.2087 | 7.7704 | 7.9247 | 7.8672 | 8.0169 | 8.5256 | 8.5257 | 6.9148 | **6.913** |
| 40 | 5% | 9.0983 | 9.0152 | 9.0174 | 9.0165 | 9.0185 | **8.9421** | **8.9421** | 8.7381 | **8.7377** |
| | 10% | 9.055 | 8.9144 | 8.9258 | 8.921 | 8.9314 | **8.8319** | 8.8319 | **8.4716** | **8.4708** |
| | 15% | 9.1628 | 8.8982 | 8.9267 | 8.9149 | 8.941 | **8.8432** | 8.8433 | **8.2802** | **8.2792** |
| | 20% | 9.1731 | 8.7396 | 8.79 | 8.7697 | 8.8164 | **8.7631** | **8.7631** | **7.9761** | **7.9751** |
| | 30% | 9.1341 | 8.2855 | 8.3868 | 8.3471 | 8.443 | 8.6036 | 8.6036 | **7.4512** | **7.4497** |
| | 40% | 9.064 | 7.6554 | 7.8016 | 7.7483 | 7.8909 | 8.4229 | 8.423 | **6.9127** | **6.9104** |
| 60 | 5% | 9.1347 | 9.0503 | 9.0531 | 9.0519 | 9.0545 | **8.9851** | 8.9851 | **8.7912** | **8.7908** |
| | 10% | 9.131 | 8.9562 | 8.9691 | 8.9637 | 8.9755 | **8.8789** | 8.8789 | **8.507** | **8.5063** |
| | 15% | 9.1192 | 8.8321 | 8.8627 | 8.85 | 8.8781 | **8.7865** | 8.7866 | **8.1963** | **8.1955** |
| | 20% | 9.2124 | 8.7744 | 8.826 | 8.8052 | 8.853 | **8.811** | **8.811** | **8.0275** | **8.0266** |
| | 30% | 9.1888 | 8.3568 | 8.4571 | 8.418 | 8.513 | 8.6843 | 8.6844 | **7.5453** | **7.544** |
| | 40% | 9.1625 | 7.7053 | 7.8571 | 7.8007 | 7.949 | 8.5092 | 8.5093 | **6.9678** | **6.9654** |
| 80 | 5% | 9.0678 | 8.9816 | 8.9848 | 8.9835 | 8.9863 | **8.9176** | 8.9176 | 8.7206 | **8.7203** |
| | 10% | 9.0619 | 8.8957 | 8.9095 | 8.9038 | 8.9163 | **8.82** | 8.8201 | **8.4369** | **8.4362** |
| | 15% | 9.0816 | 8.7834 | 8.8135 | 8.8011 | 8.8288 | **8.7444** | 8.7444 | **8.183** | **8.182** |
| | 20% | 9.1508 | 8.7015 | 8.7537 | 8.7326 | 8.781 | **8.7443** | **8.7444** | **7.9726** | **7.9716** |
| | 30% | 9.1192 | 8.2634 | 8.3645 | 8.3255 | 8.4213 | 8.601 | 8.6011 | **7.4572** | **7.4559** |
| | 40% | 9.0995 | 7.6533 | 7.8021 | 7.7476 | 7.8931 | 8.4547 | 8.4548 | **6.9305** | **6.9282** |
| 100 | 5% | 9.0725 | 8.9862 | 8.9895 | 8.9881 | 8.9911 | **8.925** | 8.925 | **8.7337** | **8.7333** |
| | 10% | 9.0981 | 8.9371 | 8.9507 | 8.9451 | 8.9576 | **8.8661** | 8.8661 | **8.4946** | **8.4939** |
| | 15% | 9.1319 | 8.8225 | 8.854 | 8.841 | 8.87 | **8.7856** | 8.7856 | **8.1968** | **8.196** |
| | 20% | 9.2084 | 8.7477 | 8.801 | 8.7795 | 8.8289 | **8.7951** | 8.7952 | **8.0071** | **8.0062** |
| | 30% | 9.1818 | 8.306 | 8.4098 | 8.3696 | 8.468 | 8.6557 | 8.6559 | **7.4841** | **7.4828** |
| | 40% | 9.1597 | 7.6988 | 7.8495 | 7.7946 | 7.942 | 8.5152 | 8.5154 | **6.9602** | **6.9582** |
| 120 | 5% | 9.0252 | 8.9328 | 8.9363 | 8.9348 | 8.9379 | **8.8723** | 8.8723 | **8.6807** | **8.6804** |
| | 10% | 9.0731 | 8.9116 | 8.926 | 8.92 | 8.9331 | **8.8397** | 8.8398 | **8.4517** | **8.451** |
| | 15% | 9.0631 | 8.7726 | 8.8036 | 8.7909 | 8.8195 | **8.7385** | 8.7386 | **8.1649** | **8.1641** |
| | 20% | 9.1355 | 8.69 | 8.7421 | 8.7211 | 8.7695 | **8.7388** | **8.7389** | **7.9761** | **7.9751** |
| | 30% | 9.1794 | 8.3026 | 8.4051 | 8.3653 | 8.4627 | 8.6504 | 8.6506 | **7.5096** | **7.5079** |
| | 40% | 9.1203 | 7.6519 | 7.8012 | 7.7468 | 7.893 | 8.4651 | 8.4652 | **6.9317** | **6.9295** |
| 200 | 5% | 9.0581 | 8.9736 | 8.9773 | 8.9758 | 8.9791 | 8.9148 | 8.9148 | **8.723** | **8.7227** |
| | 10% | 9.065 | 8.8903 | 8.9048 | 8.8988 | 8.9121 | 8.8228 | 8.8229 | **8.4442** | **8.4436** |
| | 15% | 9.1 | 8.8009 | 8.8329 | 8.8198 | 8.8493 | 8.7705 | 8.7707 | **8.1789** | **8.1782** |
| | 20% | 9.1051 | 8.6383 | 8.6923 | 8.6707 | 8.7208 | 8.6937 | 8.6939 | **7.9083** | **7.9072** |
| | 30% | 9.1409 | 8.2658 | 8.3682 | 8.3286 | 8.4258 | 8.6185 | 8.6187 | **7.475** | **7.4736** |
| | 40% | 9.1051 | 7.637 | 7.7856 | 7.7314 | 7.877 | 8.4522 | 8.4525 | **6.9316** | **6.9293** |
| 500 | 5% | 9.0664 | 8.9758 | 8.9797 | 8.9781 | 8.9816 | 8.9183 | 8.9185 | **8.7228** | **8.7225** |
| | 10% | 9.088 | 8.9034 | 8.9187 | 8.9123 | 8.9262 | 8.838 | 8.8382 | **8.4495** | **8.4489** |
| | 15% | 9.0942 | 8.7907 | 8.8224 | 8.8094 | 8.8386 | 8.7658 | 8.7662 | **8.1997** | **8.1988** |
| | 20% | 9.1933 | 8.736 | 8.7893 | 8.7679 | 8.8174 | 8.7958 | 8.7962 | **8.0293** | **8.0283** |
| | 30% | 9.1229 | 8.2491 | 8.3518 | 8.3123 | 8.41 | 8.6098 | 8.6104 | **7.4611** | **7.4596** |
| | 40% | 9.135 | 7.656 | 7.805 | 7.7509 | 7.897 | 8.4821 | 8.4827 | **6.9543** | **6.9519** |


**Table 7. Mean absolute percentage errors for the scenario where error variance was 1; sample sizes of 20, 40, 60, 80, 100, 120, 200, and 500; and missing value percentages of 5, 10, 15, 20, 30, and 40% with SRS in missing value estimation methods.**

| Sample sizes | Missing values | Listwise delition | Missing value estimation methods | | | | | | | |
|---|---|---|---|---|---|---|---|---|---|---|
| | | | R-RQ1 | R-RQ3 | AR-CRQ1 | AR-CRQ3 | AR-MRQ1 | AR-MRQ3 | AR-MCRQ1 | AR-MCRQ3 |
| 20 | 5% | 357.0240 | 409.9905 | 409.9852 | 409.9874 | 409.9826 | **409.4746** | 409.4746 | 408.7430 | **408.7323** |
| | 10% | 161.5125 | 145.8039 | 145.9150 | 145.8644 | 145.9650 | **144.4226** | 144.4227 | **136.7948** | **136.8606** |
| | 15% | 195.3549 | 157.7165 | 157.9755 | 157.8668 | 158.1025 | **156.6009** | 156.6011 | **146.8472** | **146.9226** |
| | 20% | 165.0662 | 151.6738 | 152.0258 | 151.8722 | 152.1969 | **151.4220** | **151.4221** | **144.4557** | **144.4908** |
| | 30% | 150.5120 | 142.5156 | 143.0980 | 142.8585 | 143.4077 | 144.0135 | 144.0136 | 137.9127 | **137.9066** |
| | 40% | 186.2471 | 149.8810 | 151.0837 | 150.6009 | 151.7644 | 155.3774 | 155.3776 | 142.6087 | **142.6521** |
| 40 | 5% | 148.1718 | 142.1702 | 142.1816 | 142.1768 | 142.1871 | **141.7056** | **141.7057** | 140.7583 | **140.7527** |
| | 10% | 332.7213 | 304.5201 | 304.6010 | 304.5674 | 304.6411 | **303.8480** | 303.8481 | **300.8522** | **300.8560** |
| | 15% | 181.5588 | 172.8844 | 173.0961 | 173.0067 | 173.2009 | **172.4254** | 172.4256 | **167.5177** | **167.5347** |
| | 20% | 208.7216 | 188.8416 | 189.2534 | 189.0736 | 189.4525 | **189.0027** | **189.0029** | **181.8044** | **181.8443** |
| | 30% | 456.8280 | 358.6434 | 365.2686 | 362.2722 | 368.3630 | 376.5335 | 376.5363 | **259.0079** | **260.5648** |
| | 40% | 990.4423 | 763.1088 | 764.3093 | 763.8936 | 765.0563 | 769.4204 | 769.4208 | **754.9609** | **755.0088** |
| 60 | 5% | 278.8807 | 273.9939 | 274.0147 | 274.0060 | 274.0248 | **273.4412** | 273.4413 | **271.5614** | **271.5667** |
| | 10% | 181.2294 | 170.4120 | 170.5020 | 170.4642 | 170.5461 | **169.8508** | 169.8511 | **167.1597** | **167.1613** |
| | 15% | 890.4421 | 980.1620 | 980.8434 | 980.5731 | 981.1931 | **978.9446** | 978.9457 | **952.7097** | **952.9737** |
| | 20% | 163.3203 | 157.5219 | 157.9053 | 157.7598 | 158.1143 | **157.7790** | **157.7794** | **150.5265** | **150.5546** |
| | 30% | 547.7462 | 449.1250 | 454.3164 | 452.3545 | 457.1381 | 465.6979 | 465.7010 | **364.3833** | **365.4964** |
| | 40% | 180.1938 | 60.5793 | 162.5951 | 161.8196 | 163.7451 | 170.2807 | 170.2816 | **144.5299** | **144.7341** |
| 80 | 5% | 216.2313 | 199.7130 | 199.7292 | 199.7222 | 199.7368 | **199.3701** | 199.3702 | 198.7978 | **198.7891** |
| | 10% | 209.7928 | 192.9521 | 193.1071 | 193.0453 | 193.1860 | **192.0357** | 192.0362 | **184.9882** | **185.0391** |
| | 15% | 183.6553 | 171.5898 | 172.0165 | 171.8482 | 172.2376 | **170.9791** | 170.9799 | **157.1203** | **157.2382** |
| | 20% | 167.7283 | 161.3135 | 161.9366 | 161.6819 | 162.2553 | **161.7816** | **161.7825** | **147.8705** | **147.9872** |
| | 30% | 184.9231 | 174.2962 | 175.5289 | 175.0668 | 176.2204 | 178.3638 | 178.3648 | **158.6837** | **158.8332** |
| | 40% | 332.4258 | 294.0354 | 296.0928 | 295.3839 | 297.3448 | 304.5341 | 304.5354 | **276.4066** | **276.6173** |
| 100 | 5% | 419.3375 | 408.0261 | 408.0737 | 408.0525 | 408.0955 | **407.0815** | 407.0819 | **402.0886** | **402.1348** |
| | 10% | 257.2928 | 226.9981 | 227.1258 | 227.0717 | 227.1880 | **226.3215** | 226.3220 | **221.6669** | **221.6932** |
| | 15% | 201.5364 | 186.6597 | 187.1070 | 186.9322 | 187.3404 | **186.0626** | 186.0637 | **171.3122** | **171.4387** |
| | 20% | 295.6438 | 278.7318 | 279.2318 | 279.0281 | 279.4889 | **279.1539** | 279.1548 | **269.3752** | **269.4381** |
| | 30% | 211.8033 | 230.5168 | 233.7384 | 232.3486 | 235.3259 | 240.1146 | 240.1178 | **185.5536** | **186.1916** |
| | 40% | 175.9687 | 151.8727 | 153.8010 | 153.1065 | 154.9501 | 161.6435 | 161.6450 | **136.4368** | **136.6201** |
| 120 | 5% | 582.2329 | 592.3044 | 592.3369 | 592.3228 | 592.3521 | **591.7209** | 591.7212 | **589.2743** | **589.2881** |
| | 10% | 510.5431 | 407.8482 | 408.0620 | 407.9718 | 408.1658 | **406.7322** | 406.7331 | **396.5975** | **396.6896** |
| | 15% | 212.4910 | 266.3517 | 266.5953 | 266.4960 | 266.7193 | **266.0657** | 266.0664 | **260.6137** | **260.6360** |
| | 20% | 234.3327 | 208.2549 | 209.4542 | 208.9768 | 210.0744 | **209.3135** | **209.3159** | **177.5880** | **177.9164** |
| | 30% | 298.6394 | 277.1593 | 278.1674 | 277.7869 | 278.7335 | 280.5361 | 280.5374 | **265.5794** | **265.6731** |
| | 40% | 229.5143 | 197.7167 | 200.3287 | 199.4316 | 201.9083 | 210.9107 | 210.9131 | **173.7585** | **174.0799** |
| 200 | 5% | 186.1120 | 179.9175 | 179.9484 | 179.9353 | 179.9632 | 179.4110 | 179.4115 | **177.3555** | **177.3640** |
| | 10% | 553.1657 | 483.2298 | 483.3648 | 483.3079 | 483.4306 | 482.5954 | 482.5964 | **477.9329** | **477.9589** |
| | 15% | 258.5249 | 187.3378 | 187.7430 | 187.5703 | 187.9404 | 186.9562 | 186.9580 | **175.6767** | **175.7732** |
| | 20% | 209.9852 | 194.7459 | 195.3659 | 195.1157 | 195.6862 | 195.3656 | 195.3676 | **181.9872** | **182.0943** |
| | 30% | 248.5371 | 232.3416 | 233.6640 | 233.1352 | 234.3713 | 236.6477 | 236.6504 | **216.4579** | **216.6256** |
| | 40% | 229.4328 | 206.4616 | 208.6076 | 207.8635 | 209.9099 | 217.5359 | 217.5392 | **188.2519** | **188.4772** |
| 500 | 5% | 214.5356 | 214.4683 | 214.5014 | 214.4876 | 214.5175 | 213.9699 | 213.9711 | **211.8940** | **211.9018** |
| | 10% | 213.5409 | 207.2012 | 207.3559 | 207.2924 | 207.4331 | 206.5123 | 206.5149 | **200.7052** | **200.7433** |
| | 15% | 222.4501 | 206.2181 | 206.7256 | 206.5210 | 206.9842 | 205.8072 | 205.8129 | **190.1129** | **190.2561** |
| | 20% | 196.8895 | 189.2922 | 189.7815 | 189.5879 | 190.0397 | 189.8393 | 189.8433 | **180.4290** | **180.4858** |
| | 30% | 348.8762 | 361.3112 | 363.2223 | 362.4833 | 364.2601 | 367.6744 | 367.6841 | **335.1211** | **335.4339** |
| | 40% | 262.2759 | 242.5020 | 246.5328 | 245.1165 | 248.9085 | 262.4728 | 262.4869 | **203.8819** | **204.4593** |


**Table 8. Mean absolute percentage errors for the scenario where error variance was 3; sample sizes of 20, 40, 60, 80, 100, 120, 200, and 500; and missing value percentages of 5, 10, 15, 20, 30, and 40% with SRS in missing value estimation methods.**

| Sample sizes | Missing values | Listwise delition | Missing value estimation methods | | | | | | | |
|---|---|---|---|---|---|---|---|---|---|---|
| | | | R-RQ1 | R-RQ3 | AR-CRQ1 | AR-CRQ3 | AR-MRQ1 | AR-MRQ3 | AR-MCRQ1 | AR-MCRQ3 |
| 20 | 5% | 396.3733 | 422.8264 | 422.821 | 422.8233 | 422.8185 | **421.9966** | 421.9966 | **419.1232** | **419.1328** |
| | 10% | 317.9099 | 307.931 | 308.1132 | 308.0258 | 308.1906 | **305.8631** | 305.8633 | **291.9799** | **292.1166** |
| | 15% | 454.3682 | 434.9818 | 435.3811 | 435.2174 | 435.5794 | **433.2143** | 433.2146 | **414.3878** | **414.5526** |
| | 20% | 254.2812 | 246.5314 | 247.3028 | 246.9751 | 247.6784 | **246.107** | **246.1073** | **223.7344** | **223.931** |
| | 30% | 278.04 | 263.731 | 265.3545 | 264.641 | 266.1389 | 267.6247 | 267.6251 | **240.4185** | **240.6636** |
| | 40% | 308.6102 | 262.9953 | 265.6628 | 264.6591 | 267.1647 | 274.707 | 274.7073 | **238.0882** | **238.3747** |
| 40 | 5% | 266.4744 | 263.3253 | 263.35 | 263.3395 | 263.3617 | **262.35** | **262.3501** | **257.7308** | **257.7634** |
| | 10% | 396.9011 | 372.1402 | 372.2718 | 372.2164 | 372.3357 | **371.0895** | 371.0898 | **364.0336** | **364.0766** |
| | 15% | 316.0263 | 306.6974 | 307.0603 | 306.9088 | 307.2391 | **305.9254** | 305.9257 | **293.7715** | **293.8603** |
| | 20% | 366.2173 | 354.3205 | 355.195 | 354.8227 | 355.622 | **354.7096** | **354.7102** | **332.5847** | **332.7857** |
| | 30% | 330.9573 | 350.9561 | 354.4466 | 352.9707 | 356.1843 | 360.8083 | 360.8098 | **298.3329** | **299.03** |
| | 40% | 2097.106 | 1859.346 | 1862.032 | 1861.061 | 1863.591 | 1872.011 | 1872.012 | **1835.512** | **1835.786** |
| 60 | 5% | 349.1543 | 347.6565 | 347.6962 | 347.6794 | 347.7153 | **346.6626** | 346.6629 | **341.4377** | **341.4786** |
| | 10% | 420.847 | 410.369 | 410.5406 | 410.4675 | 410.623 | **409.3037** | 409.3041 | **400.7923** | **400.8544** |
| | 15% | 2305.664 | 2181.875 | 2182.851 | 2182.455 | 2183.341 | **2180.143** | 2180.144 | **2140.96** | **2141.369** |
| | 20% | 278.576 | 271.2231 | 272.0026 | 271.7063 | 272.419 | **271.7572** | **271.758** | **251.4015** | **251.5567** |
| | 30% | 318.7801 | 339.4706 | 342.4024 | 341.2751 | 343.9779 | 348.6231 | 348.6249 | **294.6227** | **295.1527** |
| | 40% | 271.1867 | 248.4935 | 251.9869 | 250.711 | 253.9964 | 264.9377 | 264.9392 | **216.2249** | **216.6472** |
| 80 | 5% | 397.8818 | 391.3425 | 391.3729 | 391.36 | 391.3874 | **390.6676** | 390.6679 | **387.5581** | **387.5731** |
| | 10% | 335.6831 | 342.5455 | 342.7547 | 342.669 | 342.8587 | **341.3178** | 341.3185 | **330.4685** | **330.5557** |
| | 15% | 328.6992 | 318.0031 | 318.7793 | 318.4818 | 319.1875 | **316.8476** | 316.8491 | **286.7756** | **287.0621** |
| | 20% | 297.8024 | 280.6823 | 281.7066 | 281.2848 | 282.2214 | **281.4747** | **281.4761** | **255.5061** | **255.7462** |
| | 30% | 340.2415 | 328.7046 | 331.1411 | 330.2098 | 332.4608 | 336.4465 | 336.4485 | **292.7914** | **293.1976** |
| | 40% | 332.3621 | 331.4737 | 334.6426 | 333.5024 | 336.4863 | 346.6875 | 346.6893 | **303.063** | **303.4233** |
| 100 | 5% | 362.0147 | 342.5573 | 342.6404 | 342.6025 | 342.6775 | **340.9733** | 340.9741 | **331.1587** | **331.2629** |
| | 10% | 529.6419 | 500.5935 | 500.832 | 500.7322 | 500.9482 | **499.3082** | 499.3091 | **487.1981** | **487.3011** |
| | 15% | 355.9075 | 360.7409 | 361.7501 | 361.3167 | 362.2337 | **359.5388** | 359.5412 | **323.42** | **323.8102** |
| | 20% | 383.7087 | 381.2693 | 382.2711 | 381.8651 | 382.7811 | **382.1268** | 382.1285 | **356.8614** | **357.0861** |
| | 30% | 366.27 | 377.8006 | 382.636 | 380.5946 | 385.0477 | 392.2941 | 392.2989 | **306.6815** | **307.6715** |
| | 40% | 280.3766 | 245.8241 | 250.2469 | 248.6661 | 252.8106 | 266.9368 | 266.9399 | **203.5037** | **204.1007** |
| 120 | 5% | **509.8418** | 526.3059 | 526.3526 | 526.3327 | 526.3749 | **525.4324** | 525.4329 | **520.6026** | **520.6388** |
| | 10% | 456.7394 | 359.9311 | 360.2345 | 360.1058 | 360.3807 | **358.3531** | 358.3545 | **342.5536** | **342.7033** |
| | 15% | **587.473** | 639.975 | 640.5456 | 640.3103 | 640.8294 | **639.3123** | 639.3139 | **620.3177** | **620.4874** |
| | 20% | 358.1808 | 340.8846 | 342.6656 | 341.9535 | 343.5789 | **342.4863** | **342.4899** | **293.1374** | **293.655** |
| | 30% | 415.3867 | 422.2891 | 424.7144 | 423.8027 | 426.0435 | 430.175 | 430.1779 | **386.6414** | **387.0397** |
| | 40% | 356.1144 | 323.0185 | 329.0223 | 326.7624 | 332.369 | 350.6358 | 350.6406 | **265.3109** | **266.2019** |
| 200 | 5% | 347.63 | 343.6034 | 343.6637 | 343.6386 | 343.6931 | 342.5732 | 342.5741 | **336.4041** | **336.4548** |
| | 10% | **645.4392** | 689.0324 | 689.3093 | 689.1924 | 689.4434 | 687.7015 | 687.7035 | **674.07** | **674.1946** |
| | 15% | 535.4903 | 557.1074 | 561.4151 | 559.4717 | 563.3826 | 553.4944 | 553.5148 | **398.4252** | **400.3846** |
| | 20% | 359.2948 | 347.4576 | 348.534 | 348.1032 | 349.0875 | 348.5416 | 348.5451 | **321.3754** | **321.6214** |
| | 30% | 406.0266 | 373.3965 | 376.0463 | 375.0006 | 377.4483 | 381.8634 | 381.8686 | **335.7447** | **336.1896** |
| | 40% | 325.0766 | 304.4598 | 308.1997 | 306.8757 | 310.3892 | 322.6664 | 322.6716 | **270.0919** | **270.5525** |
| 500 | 5% | 397.1003 | 387.9585 | 388.0188 | 387.9936 | 388.0481 | 387.0228 | 387.0251 | **381.3677** | **381.4121** |
| | 10% | 422.2752 | 423.1033 | 423.3765 | 423.2655 | 423.5131 | 421.8598 | 421.8646 | **408.4577** | **408.5738** |
| | 15% | 330.8041 | 319.7075 | 320.4169 | 320.1297 | 320.7752 | 319.1339 | 319.1419 | **295.2779** | **295.5011** |
| | 20% | 386.4342 | 362.2357 | 363.2429 | 362.849 | 363.7705 | 363.3653 | 363.3734 | **338.2637** | **338.48** |
| | 30% | 531.4369 | 520.1855 | 525.1385 | 523.2082 | 527.7705 | 536.2515 | 536.2755 | **444.2679** | **445.2691** |
| | 40% | 560.9668 | 498.7476 | 511.2527 | 506.8453 | 518.4787 | 558.3848 | 558.4253 | **368.1993** | **370.3058** |


**Table 9. Mean absolute percentage errors for the scenario where error variance was 5; sample sizes of 20, 40, 60, 80, 100, 120, 200 and 500; and missing value percentages of 5, 10, 15, 20, 30 and 40% with SRS in missing value estimation methods.**

| Sample sizes | Missing values | Listwise delition | Missing value estimation methods | | | | | | | |
|---|---|---|---|---|---|---|---|---|---|---|
| | | | R-RQ1 | R-RQ3 | AR-CRQ1 | AR-CRQ3 | AR-MRQ1 | AR-MRQ3 | AR-MCRQ1 | AR-MCRQ3 |
| 20 | 5% | 750.4612 | 693.8617 | 693.8554 | 693.8581 | 693.8525 | **692.904** | 692.9041 | **689.2127** | **689.2302** |
| | 10% | 467.4288 | 452.7391 | 452.8756 | 452.8179 | 452.9412 | **450.7353** | 450.7355 | **437.8525** | **437.9655** |
| | 15% | 607.3061 | 611.7478 | 612.1518 | 611.9836 | 612.3501 | **610.017** | 610.0173 | **590.8113** | **590.9777** |
| | 20% | 382.3756 | 378.9746 | 380.1031 | 379.634 | 380.6615 | **378.3465** | **378.347** | **341.5215** | **341.8813** |
| | 30% | 362.1214 | 347.8565 | 349.9756 | 349.0768 | 351.0247 | 353.0042 | 353.0047 | **314.2307** | **314.597** |
| | 40% | 553.9524 | 553.7607 | 557.3752 | 556.0309 | 559.4029 | 569.4374 | 569.4379 | **516.8758** | **517.3331** |
| 40 | 5% | 312.4886 | 312.9199 | 312.947 | 312.9358 | 312.9602 | **311.8056** | **311.8057** | **306.1122** | **306.1558** |
| | 10% | **639.3022** | 658.7315 | 658.9114 | 658.8373 | 659.0001 | **657.2774** | 657.2777 | **645.9464** | **646.0334** |
| | 15% | 463.3882 | 454.2956 | 454.8909 | 454.6412 | 455.1819 | **453.0571** | 453.0577 | **430.0943** | **430.3065** |
| | 20% | 418.4452 | 402.5271 | 403.899 | 403.317 | 404.5676 | **403.1531** | **403.1541** | **364.9162** | **365.3096** |
| | 30% | 522.2293 | 518.1702 | 526.5711 | 522.8652 | 530.5776 | 541.146 | 541.1494 | **387.007** | **388.9696** |
| | 40% | 1267.334 | 1014.413 | 1018.478 | 1017.051 | 1020.849 | 1033.411 | 1033.413 | **973.5068** | **974.0308** |
| 60 | 5% | 480.4941 | 473.2105 | 473.261 | 473.2403 | 473.2858 | **471.8589** | 471.8593 | **464.0082** | **464.0746** |
| | 10% | 428.569 | 425.4493 | 425.6524 | 425.5673 | 425.7512 | **424.1581** | 424.1587 | **413.0386** | **413.126** |
| | 15% | 1705.542 | 1589.961 | 1590.843 | 1590.486 | 1591.286 | **1588.438** | 1588.439 | **1553.425** | **1553.777** |
| | 20% | 409.7187 | 397.9883 | 399.1266 | 398.6919 | 399.7308 | **398.7725** | **398.7737** | **366.4374** | **366.7261** |
| | 30% | 561.1126 | 467.6295 | 471.6591 | 470.1273 | 473.8369 | 480.2411 | 480.2435 | **402.7948** | **403.5847** |
| | 40% | 376.8161 | 342.9968 | 348.3318 | 346.3966 | 351.375 | 367.6689 | 367.6711 | **289.5883** | **290.3519** |
| 80 | 5% | 445.5324 | 441.547 | 441.5882 | 441.5707 | 441.6078 | **440.6295** | 440.6298 | **435.6045** | **435.6398** |
| | 10% | 552.3313 | 539.5132 | 539.7766 | 539.6675 | 539.9061 | **537.9735** | 537.9743 | **523.3188** | **523.4479** |
| | 15% | 401.4017 | 393.0985 | 394.1194 | 393.7143 | 394.6414 | **391.6475** | 391.6495 | **351.6491** | **352.0527** |
| | 20% | 408.7774 | 386.9152 | 388.5722 | 387.9032 | 389.4145 | **388.1856** | **388.1879** | **341.7282** | **342.2018** |
| | 30% | 389.7421 | 368.7795 | 371.8356 | 370.7044 | 373.5209 | 378.6026 | 378.605 | **320.6346** | **321.1863** |
| | 40% | 433.04 | 419.5373 | 424.6457 | 422.7895 | 427.5593 | 443.3487 | 443.3515 | **369.7394** | **370.4442** |
| 100 | 5% | 743.1029 | 755.3779 | 755.5042 | 755.4452 | 755.5592 | **752.9456** | 752.9468 | **736.8654** | **737.0511** |
| | 10% | 643.3278 | 656.4616 | 656.7896 | 656.6526 | 656.9494 | **654.6904** | 654.6916 | **636.5673** | **636.7396** |
| | 15% | 474.185 | 440.3931 | 441.6299 | 441.0897 | 442.2129 | **438.9609** | 438.9638 | **394.0767** | **394.578** |
| | 20% | 871.2665 | 864.3896 | 865.7652 | 865.2108 | 866.466 | **865.5703** | 865.5726 | **828.2954** | **828.6572** |
| | 30% | 589.5559 | 497.0344 | 504.7744 | 501.4655 | 508.5787 | 519.9694 | 519.977 | **380.2472** | **381.9385** |
| | 40% | 433.9685 | 364.4773 | 372.4158 | 369.665 | 377.0537 | 402.2187 | 402.224 | **281.1981** | **282.4487** |
| 120 | 5% | **1164.496** | 1181.746 | 1181.817 | 1181.786 | 1181.85 | **1180.445** | 1180.446 | **1172.303** | **1172.378** |
| | 10% | **457.1772** | 503.7146 | 504.1335 | 503.9612 | 504.3404 | **501.4991** | 501.501 | **477.6804** | **477.918** |
| | 15% | **532.175** | 462.0255 | 462.7559 | 462.4563 | 463.1202 | **461.1734** | 461.1754 | **435.1605** | **435.4066** |
| | 20% | 437.3237 | 413.6748 | 415.7798 | 414.9141 | 416.8338 | **415.5819** | **415.5861** | **357.5116** | **358.1336** |
| | 30% | 756.5897 | 716.1417 | 719.7726 | 718.4079 | 721.7522 | 727.8758 | 727.88 | **659.4141** | **660.0915** |
| | 40% | 450.8706 | 412.4233 | 417.0944 | 415.4411 | 419.8074 | 434.7024 | 434.7062 | **367.0169** | **367.6471** |
| 200 | 5% | 446.1682 | 437.6951 | 437.7571 | 437.7307 | 437.7866 | 436.6481 | 436.6491 | **430.2039** | **430.2583** |
| | 10% | **700.1205** | 749.8367 | 750.1511 | 750.0177 | 750.3024 | 748.3376 | 748.3399 | **732.3101** | **732.4598** |
| | 15% | 493.4631 | 418.4324 | 419.7434 | 419.1822 | 420.3733 | 417.2309 | 417.237 | **370.359** | **370.8765** |
| | 20% | 425.5464 | 412.6714 | 414.0686 | 413.5062 | 414.7818 | 414.0786 | 414.0832 | **377.2279** | **377.5835** |
| | 30% | 612.7861 | 593.986 | 597.3977 | 596.0568 | 599.2017 | 604.8677 | 604.8744 | **543.3552** | **543.9724** |
| | 40% | 471.3843 | 422.4269 | 428.4866 | 426.3523 | 432.0073 | 451.4099 | 451.418 | **362.1695** | **363.0486** |
| 500 | 5% | 470.9242 | 460.2495 | 460.3311 | 460.2969 | 460.3706 | 458.9775 | 458.9806 | **450.5111** | **450.5876** |
| | 10% | 459.4716 | 457.7515 | 458.1258 | 457.9734 | 458.3125 | 456.0438 | 456.0505 | **436.2439** | **436.4307** |
| | 15% | 458.9581 | 438.2558 | 439.4288 | 438.9584 | 440.0242 | 437.2954 | 437.3087 | **394.3314** | **394.7732** |
| | 20% | 565.0702 | 555.0596 | 556.4491 | 555.9068 | 557.1755 | 556.6179 | 556.6291 | **519.5542** | **519.9027** |
| | 30% | 800.0828 | 766.4581 | 771.1895 | 769.3731 | 773.7293 | 781.9255 | 781.9484 | **693.1935** | **694.13** |
| | 40% | 966.6159 | 821.1149 | 844.735 | 836.5476 | 858.4684 | 933.8673 | 933.9428 | **565.5716** | **569.7543** |


**Table 10. Mean absolute percentage errors for the scenario where error variance was 7; sample sizes of 20, 40, 60, 80, 100, 120, 200 and 500; and missing value percentages of 5, 10, 15, 20, 25 and 40% with SRS in missing value estimation methods.**

| Sample sizes | Missing values | Listwise delition | Missing value estimation methods | | | | | | | |
|---|---|---|---|---|---|---|---|---|---|---|
| | | | R-RQ1 | R-RQ3 | AR-CRQ1 | AR-CRQ3 | AR-MRQ1 | AR-MRQ3 | AR-MCRQ1 | AR-MCRQ3 |
| 20 | 5% | **726.4176** | 751.6997 | 751.6923 | 751.6955 | 751.6888 | **750.4558** | 750.4559 | **745.1667** | **745.1982** |
| | 10% | 370.051 | 381.2328 | 381.6068 | 381.4389 | 381.7766 | **376.771** | 376.7715 | **342.5826** | **342.9597** |
| | 15% | 496.688 | 507.5724 | 508.314 | 508.0007 | 508.6727 | **504.6236** | 504.6241 | **467.2244** | **467.6101** |
| | 20% | 437.6471 | 423.7297 | 424.849 | 424.3842 | 425.403 | **423.1758** | **423.1763** | **386.6536** | **387.012** |
| | 30% | 403.1041 | 383.8389 | 386.6207 | 385.4362 | 387.9914 | 390.5875 | 390.5881 | **337.1162** | **337.6714** |
| | 40% | 713.921 | 710.7703 | 715.4452 | 713.601 | 717.9474 | 730.1807 | 730.1813 | **663.0215** | **663.676** |
| 40 | 5% | 424.043 | 420.741 | 420.7763 | 420.7615 | 420.7934 | **419.3295** | **419.3297** | **411.5544** | **411.6235** |
| | 10% | **478.5885** | 460.0698 | 460.2943 | 460.2011 | 460.4041 | **458.3072** | 458.3076 | **443.431** | **443.5615** |
| | 15% | 472.3669 | 453.4062 | 454.0093 | 453.7571 | 454.3047 | **452.1407** | 452.1413 | **428.2866** | **428.507** |
| | 20% | 457.8988 | 438.3387 | 439.5158 | 439.0171 | 440.0903 | **438.8387** | **438.8396** | **406.3538** | **406.6653** |
| | 30% | 679.4666 | 706.2843 | 711.9935 | 709.5707 | 714.8127 | 722.2921 | 722.2944 | **615.6542** | **616.9101** |
| | 40% | **5027.924** | 5234.819 | 5239.465 | 5237.845 | 5242.178 | 5256.465 | 5256.466 | **5186.624** | **5187.263** |
| 60 | 5% | 533.8603 | 534.2321 | 534.3058 | 534.2741 | 534.3406 | **532.3139** | 532.3144 | **520.5379** | **520.6555** |
| | 10% | 502.5759 | 484.5606 | 484.8359 | 484.7211 | 484.9702 | **482.7985** | 482.7992 | **466.3253** | **466.4758** |
| | 15% | **2437.824** | 2519.935 | 2521.066 | 2520.622 | 2521.648 | **2517.948** | 2517.949 | **2470.512** | **2470.995** |
| | 20% | 453.0441 | 438.2988 | 439.5005 | 439.0319 | 440.1278 | **439.133** | **439.1342** | **404.9905** | **405.2996** |
| | 30% | 1018.434 | 888.7062 | 905.8432 | 899.3943 | 915.125 | 943.254 | 943.264 | **594.5933** | **598.5602** |
| | 40% | 446.7286 | 399.0657 | 404.8246 | 402.736 | 408.1035 | 425.6167 | 425.619 | **340.7522** | **341.5877** |
| 80 | 5% | 541.0856 | 533.3067 | 533.3564 | 533.3351 | 533.38 | **532.1881** | 532.1885 | **525.645** | **525.7** |
| | 10% | 435.0871 | 419.2938 | 419.6431 | 419.4966 | 419.8127 | **417.2578** | 417.2589 | **396.8294** | **397.0263** |
| | 15% | 473.7543 | 451.6715 | 452.6873 | 452.2808 | 453.2029 | **450.2224** | 450.2244 | **410.2552** | **410.6576** |
| | 20% | 454.2854 | 446.2367 | 448.31 | 447.4614 | 449.3505 | **447.828** | **447.8308** | **388.632** | **389.2623** |
| | 30% | 448.827 | 431.4693 | 434.7924 | 433.5078 | 436.5671 | 441.8726 | 441.8752 | **379.9776** | **380.586** |
| | 40% | 678.8082 | 665.4422 | 671.204 | 669.1992 | 674.5662 | 692.7515 | 692.7546 | **606.405** | **607.2376** |
| 100 | 5% | 721.18 | 702.8082 | 702.9393 | 702.88 | 702.9982 | **700.2079** | 700.2092 | **682.7908** | **682.9871** |
| | 10% | 810.4472 | 819.6088 | 820.0279 | 819.8487 | 820.2279 | **817.3991** | 817.4008 | **793.6952** | **793.9399** |
| | 15% | 491.5639 | 472.4862 | 473.6803 | 473.1799 | 474.2642 | **471.0325** | 471.0353 | **425.9613** | **426.4409** |
| | 20% | 594.4215 | 580.6282 | 582.5228 | 581.755 | 583.4824 | **582.2525** | 582.2557 | **529.0572** | **529.6034** |
| | 30% | 478.7682 | 485.601 | 489.4081 | 487.8948 | 491.3984 | 497.4113 | 497.4151 | **427.6206** | **428.3436** |
| | 40% | 456.1046 | 399.4679 | 406.9674 | 404.3423 | 411.3207 | 434.9045 | 434.9095 | **321.5435** | **322.6979** |
| 120 | 5% | **735.5547** | 720.031 | 720.1208 | 720.0822 | 720.1632 | **718.3842** | 718.3852 | **707.5568** | **707.6627** |
| | 10% | **909.8261** | 818.0973 | 818.7866 | 818.5038 | 819.1276 | **814.4664** | 814.4696 | **773.2514** | **773.6944** |
| | 15% | **576.4095** | 510.5601 | 511.533 | 511.1318 | 512.0157 | **509.4305** | 509.4332 | **473.3237** | **473.6878** |
| | 20% | 559.1264 | 527.0998 | 529.8384 | 528.7338 | 531.2296 | **529.5389** | **529.5444** | **450.4983** | **451.3667** |
| | 30% | 788.557 | 722.9402 | 727.0996 | 725.5238 | 729.3508 | 736.2992 | 736.304 | **656.9799** | **657.7865** |
| | 40% | 514.8918 | 450.7238 | 457.9915 | 455.4074 | 462.1729 | 484.8728 | 484.8786 | **376.2669** | **377.3807** |
| 200 | 5% | 560.3821 | 559.4507 | 559.5467 | 559.5068 | 559.5935 | 557.8012 | 557.8027 | **546.6495** | **546.7577** |
| | 10% | **870.8277** | 801.8949 | 802.239 | 802.0933 | 802.4049 | 800.2426 | 800.2451 | **782.1006** | **782.2731** |
| | 15% | 546.5194 | 554.0526 | 555.6623 | 554.9659 | 556.4277 | 552.5758 | 552.5833 | **494.2487** | **494.91** |
| | 20% | 487.4832 | 469.7138 | 471.208 | 470.6176 | 471.981 | 471.2288 | 471.2337 | **430.5422** | **430.9327** |
| | 30% | 599.0403 | 558.966 | 563.4472 | 561.676 | 565.7992 | 573.1585 | 573.1672 | **490.4093** | **491.2828** |
| | 40% | 529.7735 | 475.414 | 481.7525 | 479.5706 | 485.4759 | 505.9471 | 505.9555 | **410.869** | **411.797** |
| 500 | 5% | 821.9789 | 830.0827 | 830.1782 | 830.1382 | 830.2245 | 828.5893 | 828.5929 | **818.2699** | **818.3678** |
| | 10% | 549.0909 | 554.103 | 554.5535 | 554.3708 | 554.7786 | 552.0388 | 552.0469 | **527.361** | **527.6041** |
| | 15% | 524.1909 | 521.7286 | 522.9995 | 522.4837 | 523.6385 | 520.7042 | 520.7186 | **473.9867** | **474.4747** |
| | 20% | 514.5752 | 503.84 | 505.3422 | 504.7556 | 506.1266 | 505.5259 | 505.5379 | **464.7822** | **465.1709** |
| | 30% | 760.8724 | 745.1218 | 751.5162 | 749.0342 | 754.9144 | 765.8352 | 765.8659 | **644.3728** | **645.7092** |
| | 40% | 786.801 | 649.1244 | 662.591 | 657.8974 | 670.4082 | 713.4246 | 713.4679 | **506.6956** | **508.959** |


**Table 11. Mean absolute percentage errors for the scenario where error variance was 9; sample sizes of 20, 40, 60, 80, 100, 120, 200 and 500; and missing value percentages of 5, 10, 15, 20, 30 and 40% with SRS in missing value estimation methods.**

| Sample sizes | Missing values | Listwise delition | Missing value estimation methods | | | | | | | |
|---|---|---|---|---|---|---|---|---|---|---|
| | | | R-RQ1 | R-RQ3 | AR-CRQ1 | AR-CRQ3 | AR-MRQ1 | AR-MRQ3 | AR-MCRQ1 | AR-MCRQ3 |
| 20 | 5% | **603.6725** | 570.9157 | 570.9088 | 570.9118 | 570.9056 | **569.6877** | 569.6878 | **564.2018** | **564.2414** |
| | 10% | 449.4994 | 439.9817 | 440.2546 | 440.131 | 440.3776 | **436.4939** | 436.4942 | **410.841** | **411.1112** |
| | 15% | 652.8954 | 621.6508 | 622.4758 | 622.1564 | 622.9035 | **618.121** | 618.1215 | **572.6301** | **573.0768** |
| | 20% | 485.0231 | 481.6543 | 483.1272 | 482.5236 | 483.8624 | **480.9152** | **480.9158** | **430.2197** | **430.7341** |
| | 30% | 468.5767 | 443.7278 | 447.4823 | 445.7304 | 449.1745 | 452.2996 | 452.3004 | **383.966** | **384.7569** |
| | 40% | 747.7586 | 678.9428 | 684.2214 | 682.2591 | 687.1585 | 701.5145 | 701.5152 | **620.911** | **621.7104** |
| 40 | 5% | 485.6536 | 485.941 | 485.9822 | 485.9648 | 486.002 | **484.2917** | **484.2919** | **474.9915** | **475.0778** |
| | 10% | **775.1819** | 787.0131 | 787.2625 | 787.1584 | 787.384 | **785.0115** | 785.012 | **767.8901** | **768.0496** |
| | 15% | 445.2454 | 441.4317 | 442.168 | 441.861 | 442.5291 | **439.8805** | 439.8813 | **409.8568** | **410.1511** |
| | 20% | 668.0976 | 658.9483 | 660.5771 | 659.8865 | 661.3702 | **659.6675** | **659.6687** | **612.6107** | **613.1039** |
| | 30% | 712.9221 | 726.7647 | 735.0896 | 731.5045 | 739.1452 | 749.8798 | 749.8832 | **593.4361** | **595.3666** |
| | 40% | **4318.264** | 4490.4 | 4495.763 | 4493.863 | 4498.854 | 4515.066 | 4515.067 | **4433.942** | **4434.723** |
| 60 | 5% | 511.6469 | 506.399 | 506.4667 | 506.4379 | 506.499 | **504.6237** | 504.6241 | **493.7173** | **493.8234** |
| | 10% | 587.0515 | 572.5583 | 572.8415 | 572.7227 | 572.979 | **570.7519** | 570.7526 | **553.587** | **553.7449** |
| | 15% | **2672.749** | 2743.382 | 2744.689 | 2744.154 | 2745.341 | **2741.242** | 2741.244 | **2687.662** | **2688.236** |
| | 20% | 487.16 | 472.4826 | 473.8279 | 473.3066 | 474.5326 | **473.3889** | **473.3903** | **433.9908** | **434.3586** |
| | 30% | 1364.768 | 1271.076 | 1275.14 | 1273.601 | 1277.337 | 1283.719 | 1283.722 | **1204.347** | **1205.151** |
| | 40% | 490.7219 | 430.5766 | 437.3787 | 434.8645 | 441.1902 | 461.4485 | 461.4512 | **361.2466** | **362.2772** |
| 80 | 5% | 715.9471 | 719.2025 | 719.2576 | 719.234 | 719.2837 | **717.9766** | 717.977 | **710.6054** | **710.6704** |
| | 10% | 529.0046 | 516.5218 | 516.9212 | 516.7592 | 517.1207 | **514.1495** | 514.1508 | **489.6722** | **489.9098** |
| | 15% | 516.9169 | 504.4084 | 505.5928 | 505.1309 | 506.2062 | **502.6909** | 502.6932 | **454.3902** | **454.8814** |
| | 20% | 518.1969 | 497.3174 | 499.2072 | 498.4407 | 500.1627 | **498.7775** | **498.7801** | **444.7729** | **445.3325** |
| | 30% | 522.3083 | 494.3585 | 498.877 | 497.1968 | 501.3505 | 508.839 | 508.8426 | **420.1005** | **421.0098** |
| | 40% | 636.8692 | 609.9685 | 617.5 | 614.876 | 621.8816 | 645.4509 | 645.455 | **530.4506** | **531.622** |
| 100 | 5% | 670.358 | 660.1401 | 660.2901 | 660.2218 | 660.3571 | **657.1763** | 657.1777 | **637.0766** | **637.3092** |
| | 10% | 942.9645 | 956.5314 | 957.0261 | 956.8113 | 957.2589 | **953.9479** | 953.9498 | **925.7504** | **926.0513** |
| | 15% | 630.7366 | 643.3765 | 645.2784 | 644.4414 | 646.1677 | **641.1732** | 641.1777 | **569.8043** | **570.641** |
| | 20% | 785.7022 | 766.5944 | 768.6834 | 767.8261 | 769.7303 | **768.399** | 768.4025 | **709.6924** | **710.3112** |
| | 30% | 823.4941 | 784.4692 | 797.2296 | 791.7518 | 803.4665 | 822.0874 | 822.0998 | **588.0115** | **590.9292** |
| | 40% | 554.1761 | 473.2994 | 483.9635 | 480.2417 | 490.1458 | 523.5057 | 523.5127 | **359.2338** | **360.9938** |
| 120 | 5% | **846.3466** | 828.597 | 828.6833 | 828.646 | 828.7239 | **826.9885** | 826.9895 | **816.3837** | **816.4888** |
| | 10% | **899.1226** | 994.4357 | 995.0068 | 994.7668 | 995.2836 | **991.4387** | 991.4413 | **957.8678** | **958.2258** |
| | 15% | **689.8451** | 739.3852 | 740.4189 | 739.9933 | 740.9319 | **738.1888** | 738.1917 | **699.4441** | **699.8395** |
| | 20% | 568.7786 | 554.3665 | 556.4051 | 555.5826 | 557.4409 | **556.2292** | **556.2333** | **498.5878** | **499.1883** |
| | 30% | 894.5608 | 878.6701 | 882.9708 | 881.3601 | 885.3164 | 892.5432 | 892.5482 | **809.4516** | **810.2944** |
| | 40% | 586.1464 | 518.4325 | 527.8861 | 524.6473 | 533.4326 | 563.4489 | 563.4564 | **416.9958** | **418.5304** |
| 200 | 5% | 562.7229 | 560.0087 | 560.1197 | 560.0739 | 560.174 | 558.0907 | 558.0925 | **544.8119** | **544.943** |
| | 10% | **1415.147** | 1362.89 | 1363.285 | 1363.119 | 1363.475 | 1360.994 | 1360.997 | **1339.548** | **1339.759** |
| | 15% | 866.8799 | 882.3174 | 888.8843 | 885.9287 | 891.889 | 876.8041 | 876.8352 | **636.8848** | **639.9314** |
| | 20% | 575.0515 | 555.9877 | 557.76 | 557.0619 | 558.678 | 557.7898 | 557.7956 | **508.2584** | **508.7507** |
| | 30% | 845.2901 | 812.0237 | 816.0417 | 814.4815 | 818.1783 | 824.8785 | 824.8863 | **750.0429** | **750.809** |
| | 40% | 589.8235 | 532.9402 | 540.3069 | 537.7094 | 544.5651 | 567.8883 | 567.898 | **457.7727** | **458.8919** |
| 500 | 5% | 653.686 | 657.5091 | 657.6161 | 657.5712 | 657.6677 | 655.8371 | 655.8412 | **644.0459** | **644.1617** |
| | 10% | 645.0939 | 632.743 | 633.1995 | 633.0132 | 633.4265 | 630.665 | 630.6732 | **605.6098** | **605.8576** |
| | 15% | 611.0222 | 597.5108 | 598.9397 | 598.3612 | 599.6593 | 596.3544 | 596.3706 | **543.0303** | **543.5951** |
| | 20% | 694.4496 | 662.8342 | 665.3814 | 664.4174 | 666.7396 | 665.7081 | 665.7286 | **591.3296** | **592.0963** |
| | 30% | 767.5472 | 731.8858 | 737.3545 | 735.2668 | 740.2963 | 749.791 | 749.8173 | **645.4019** | **646.5143** |
| | 40% | 1164.152 | 948.3518 | 973.4455 | 964.6876 | 987.9658 | 1067.579 | 1067.659 | **677.1057** | **681.5554** |


**Table 12. Mean square errors for the scenario where error variance was 1; sample sizes of 20, 40, 60, 80, 100, 120, 200 and 500; and missing value percentages of 5, 10, 15, 20, 30 and 40% with RSS in missing value estimation methods.**

| Sample sizes | Missing values | Listwise delition | Missing value estimation methods | | | | | | | |
|---|---|---|---|---|---|---|---|---|---|---|
| | | | R-RQ1 | R-RQ3 | AR-CRQ1 | AR-CRQ3 | AR-MRQ1 | AR-MRQ3 | AR-MCRQ1 | AR-MCRQ3 |
| 20 | 5% | **1.0374** | 1.0225 | 1.0224 | 1.0225 | 1.0224 | **1.0129** | **1.0129** | **1.0757** | **1.0738** |
| | 10% | 1.0224 | 1.0012 | 1.0021 | 1.0017 | 1.0025 | **0.9908** | 0.9908 | **1.1195** | **1.1157** |
| | 15% | 1.0415 | 1.0112 | 1.0135 | 1.0125 | 1.0146 | **1.0037** | **1.0037** | **1.1947** | **1.1891** |
| | 20% | 1.0278 | 0.9918 | 0.9947 | 0.9934 | 0.9963 | **0.9911** | **0.9911** | **1.2603** | **1.2528** |
| | 30% | 1.0528 | 1.0071 | 1.0085 | 1.0077 | 1.0100 | **1.0151** | 1.0151 | **1.4203** | **1.4090** |
| | 40% | 1.0310 | 0.9788 | 0.9700 | 0.9727 | 0.9673 | **0.9773** | 0.9773 | **1.5215** | **1.5065** |
| 40 | 5% | 1.0119 | 0.9999 | 1.0002 | 1.0001 | 1.0003 | **0.9925** | **0.9925** | **1.0535** | **1.0517** |
| | 10% | **1.0184** | 1.0000 | 1.0012 | 1.0007 | 1.0019 | **0.9923** | 0.9923 | **1.1214** | **1.1176** |
| | 15% | 1.0188 | 0.9956 | 0.9976 | 0.9968 | 0.9987 | **0.9923** | **0.9923** | **1.1914** | **1.1859** |
| | 20% | 1.0122 | 0.9842 | 0.9865 | 0.9855 | 0.9880 | **0.9855** | **0.9855** | **1.2570** | **1.2496** |
| | 30% | 1.0072 | 0.9577 | 0.9581 | 0.9577 | 0.9593 | **0.9661** | 0.9661 | **1.3655** | **1.3545** |
| | 40% | **1.0100** | 0.9570 | 0.9468 | 0.9498 | 0.9434 | **0.9544** | 0.9544 | **1.4808** | **1.4661** |
| 60 | 5% | 1.0119 | 1.0031 | 1.0034 | 1.0033 | 1.0036 | **0.9963** | 0.9963 | **1.0569** | **1.0550** |
| | 10% | 1.0112 | 0.9975 | 0.9988 | 0.9983 | 0.9995 | **0.9912** | 0.9912 | **1.1193** | **1.1157** |
| | 15% | **1.0180** | 0.9971 | 0.9993 | 0.9984 | 1.0005 | **0.9944** | **0.9944** | **1.1939** | **1.1884** |
| | 20% | 1.0078 | 0.9760 | 0.9789 | 0.9777 | 0.9806 | **0.9782** | **0.9782** | **1.2443** | **1.2368** |
| | 30% | 1.0194 | 0.9715 | 0.9714 | 0.9712 | 0.9724 | **0.9794** | 0.9794 | **1.3812** | **1.3702** |
| | 40% | 1.0083 | 0.9570 | 0.9459 | 0.9493 | 0.9422 | **0.9526** | 0.9526 | **1.4815** | **1.4667** |
| 80 | 5% | 1.0168 | 1.0086 | 1.0090 | 1.0088 | 1.0092 | **1.0019** | 1.0019 | **1.0636** | **1.0617** |
| | 10% | 1.0130 | 1.0003 | 1.0016 | 1.0011 | 1.0023 | **0.9940** | 0.9940 | **1.1229** | **1.1192** |
| | 15% | 1.0159 | 0.9938 | 0.9962 | 0.9952 | 0.9975 | **0.9912** | 0.9912 | **1.1831** | **1.1776** |
| | 20% | 1.0145 | 0.9848 | 0.9873 | 0.9862 | 0.9889 | **0.9870** | **0.9870** | **1.2522** | **1.2449** |
| | 30% | 1.0179 | 0.9671 | 0.9674 | **0.9670** | 0.9685 | 0.9764 | 0.9764 | **1.3703** | **1.3593** |
| | 40% | 1.0162 | 0.9668 | **0.9552** | 0.9587 | 0.9512 | 0.9609 | 0.9609 | **1.4908** | **1.4760** |
| 100 | 5% | 1.0048 | 0.9979 | 0.9983 | 0.9981 | 0.9985 | **0.9918** | 0.9918 | **1.0539** | **1.0521** |
| | 10% | 1.0094 | 0.9975 | 0.9988 | 0.9983 | 0.9995 | **0.9918** | 0.9918 | **1.1192** | **1.1155** |
| | 15% | 1.0054 | 0.9840 | 0.9863 | 0.9854 | 0.9876 | **0.9818** | 0.9818 | **1.1756** | **1.1702** |
| | 20% | 1.0133 | **0.9813** | 0.9841 | 0.9829 | 0.9858 | **0.9839** | 0.9839 | **1.2446** | **1.2373** |
| | 30% | 1.0040 | **0.9538** | 0.9539 | 0.9535 | 0.9550 | 0.9628 | 0.9628 | **1.3546** | **1.3437** |
| | 40% | 1.0153 | 0.9667 | 0.9551 | 0.9586 | **0.9510** | 0.9606 | 0.9606 | **1.4856** | **1.4709** |
| 120 | 5% | **1.0106** | 1.0044 | 1.0048 | 1.0046 | 1.0050 | **0.9986** | 0.9986 | **1.0601** | **1.0583** |
| | 10% | **1.0069** | 0.9957 | 0.9970 | 0.9964 | 0.9976 | **0.9904** | **0.9904** | **1.1190** | **1.1154** |
| | 15% | **1.0128** | 0.9932 | 0.9955 | 0.9945 | 0.9967 | **0.9912** | 0.9912 | **1.1861** | **1.1806** |
| | 20% | 0.9996 | 0.9685 | 0.9713 | 0.9701 | 0.9730 | **0.9712** | **0.9712** | **1.2320** | **1.2248** |
| | 30% | 1.0053 | 0.9538 | 0.9539 | **0.9536** | 0.9551 | 0.9632 | 0.9632 | **1.3565** | **1.3455** |
| | 40% | 1.0100 | 0.9558 | 0.9446 | 0.9480 | **0.9408** | 0.9525 | 0.9525 | **1.4699** | **1.4553** |
| 200 | 5% | 1.0065 | 1.0020 | 1.0024 | 1.0023 | 1.0026 | **0.9968** | 0.9968 | **1.0599** | **1.0581** |
| | 10% | **1.0007** | 0.9909 | 0.9922 | 0.9917 | 0.9929 | **0.9860** | 0.9860 | **1.1150** | **1.1114** |
| | 15% | 1.0047 | 0.9854 | 0.9876 | 0.9867 | 0.9888 | **0.9836** | 0.9836 | **1.1794** | **1.1739** |
| | 20% | 0.9994 | 0.9690 | 0.9717 | 0.9705 | 0.9734 | **0.9718** | 0.9718 | **1.2344** | **1.2271** |
| | 30% | 1.0070 | 0.9537 | 0.9539 | **0.9535** | 0.9551 | 0.9638 | 0.9638 | **1.3521** | **1.3412** |
| | 40% | 1.0019 | 0.9503 | **0.9385** | 0.9420 | 0.9344 | 0.9451 | 0.9451 | **1.4670** | **1.4523** |
| 500 | 5% | 1.0005 | 0.9976 | 0.9980 | 0.9978 | 0.9982 | **0.9927** | 0.9927 | **1.0569** | **1.0550** |
| | 10% | 1.0054 | 0.9956 | 0.9969 | 0.9964 | 0.9976 | **0.9909** | 0.9909 | **1.1208** | **1.1171** |
| | 15% | 1.0013 | 0.9815 | 0.9838 | 0.9828 | 0.9851 | **0.9799** | 0.9800 | **1.1734** | **1.1680** |
| | 20% | 1.0075 | **0.9762** | 0.9789 | 0.9777 | 0.9807 | 0.9794 | 0.9794 | **1.2384** | **1.2311** |
| | 30% | 1.0014 | 0.9492 | 0.9492 | **0.9489** | 0.9503 | 0.9589 | 0.9590 | **1.3482** | **1.3372** |
| | 40% | 1.0060 | 0.9548 | 0.9429 | 0.9464 | **0.9386** | 0.9493 | 0.9493 | **1.4677** | **1.4531** |

Bold values indicate the best method from the comparison of the four methods with RSS.


**Table 13. Mean square errors for the scenario where error variance was 3; sample sizes of 20, 40, 60, 80, 100, 120, 200 and 500; and missing value percentages of 5, 10, 15, 20, 30 and 40% with RSS in missing value estimation methods.**

| Sample sizes | Missing values | Listwise delition | Missing value estimation methods | | | | | | | |
|---|---|---|---|---|---|---|---|---|---|---|
| | | | R-RQ1 | R-RQ3 | AR-CRQ1 | AR-CRQ3 | AR-MRQ1 | AR-MRQ3 | AR-MCRQ1 | AR-MCRQ3 |
| 20 | 5% | 3.0436 | 3.0152 | 3.0152 | 3.0152 | 3.0151 | **2.9864** | 2.9864 | 2.9788 | **2.9774** |
| | 10% | 3.0572 | 3.0201 | 3.0231 | 3.0218 | 3.0245 | **2.9818** | 2.9818 | 2.9639 | **2.9611** |
| | 15% | 2.9942 | 2.9315 | 2.9388 | 2.9357 | 2.9425 | **2.9052** | 2.9052 | 2.9039 | **2.8997** |
| | 20% | 3.0346 | 2.9292 | 2.9433 | 2.9374 | 2.9505 | **2.9255** | **2.9255** | 2.9158 | **2.9103** |
| | 30% | 3.0473 | **2.8243** | 2.8501 | 2.8395 | 2.8646 | 2.8986 | 2.8986 | 2.9032 | 2.8948 |
| | 40% | 3.0204 | **2.6357** | 2.6680 | 2.6554 | 2.6890 | 2.8227 | 2.8227 | 2.8150 | 2.8038 |
| 40 | 5% | 2.9957 | 2.9780 | 2.9787 | 2.9784 | 2.9791 | **2.9553** | **2.9553** | 2.9509 | **2.9495** |
| | 10% | 3.0000 | 2.9653 | 2.9693 | 2.9676 | 2.9713 | **2.9376** | 2.9376 | 2.9293 | **2.9265** |
| | 15% | 2.9948 | 2.9278 | 2.9370 | 2.9332 | 2.9417 | **2.9106** | 2.9106 | 2.8957 | **2.8916** |
| | 20% | 3.0166 | 2.9026 | 2.9179 | 2.9115 | 2.9259 | **2.9102** | **2.9102** | 2.8968 | **2.8913** |
| | 30% | 3.0269 | **2.7825** | 2.8096 | 2.7988 | 2.8253 | 2.8729 | 2.8729 | 2.8549 | 2.8466 |
| | 40% | 3.0436 | **2.6369** | 2.6690 | 2.6567 | 2.6904 | 2.8398 | 2.8398 | 2.8082 | 2.7973 |
| 60 | 5% | 3.0090 | 2.9960 | 2.9970 | 2.9966 | 2.9975 | **2.9743** | 2.9743 | 2.9652 | **2.9639** |
| | 10% | 2.9971 | 2.9635 | 2.9679 | 2.9661 | 2.9701 | **2.9380** | 2.9380 | 2.9255 | **2.9228** |
| | 15% | 3.0108 | 2.9473 | 2.9564 | 2.9526 | 2.9611 | **2.9343** | 2.9344 | 2.9322 | **2.9280** |
| | 20% | 3.0193 | 2.9011 | 2.9168 | 2.9103 | 2.9251 | **2.9125** | **2.9125** | 2.9042 | **2.8985** |
| | 30% | 3.0257 | **2.7737** | 2.8010 | 2.7903 | 2.8171 | 2.8692 | 2.8693 | 2.8456 | 2.8374 |
| | 40% | 2.9934 | **2.5822** | 2.6135 | 2.6015 | 2.6347 | 2.7878 | 2.7878 | 2.7624 | 2.7513 |
| 80 | 5% | 2.9974 | 2.9861 | 2.9872 | 2.9867 | 2.9877 | **2.9656** | 2.9656 | 2.9593 | **2.9580** |
| | 10% | 3.0075 | 2.9732 | 2.9779 | 2.9759 | 2.9802 | **2.9487** | 2.9487 | 2.9382 | **2.9353** |
| | 15% | 3.0090 | 2.9431 | 2.9528 | 2.9488 | 2.9577 | **2.9313** | 2.9313 | 2.9189 | **2.9148** |
| | 20% | 3.0009 | 2.8855 | 2.9011 | 2.8947 | 2.9094 | **2.8985** | **2.8985** | 2.8873 | **2.8818** |
| | 30% | 3.0003 | **2.7499** | 2.7770 | 2.7662 | 2.7928 | 2.8460 | 2.8461 | 2.8297 | 2.8213 |
| | 40% | 3.0102 | **2.5947** | 2.6259 | 2.6139 | 2.6471 | 2.8027 | 2.8027 | 2.7794 | 2.7682 |
| 100 | 5% | 2.9963 | 2.9850 | 2.9861 | 2.9857 | 2.9867 | **2.9646** | 2.9646 | 2.9570 | **2.9557** |
| | 10% | 3.0042 | 2.9728 | 2.9775 | 2.9755 | 2.9798 | **2.9499** | 2.9499 | 2.9389 | **2.9362** |
| | 15% | 3.0136 | 2.9448 | 2.9546 | 2.9505 | 2.9596 | **2.9338** | 2.9339 | 2.9219 | **2.9177** |
| | 20% | 3.0026 | 2.8839 | 2.8998 | 2.8933 | 2.9082 | **2.8982** | 2.8982 | 2.8799 | **2.8745** |
| | 30% | 3.0042 | **2.7524** | 2.7796 | 2.7689 | 2.7956 | 2.8503 | 2.8504 | 2.8221 | 2.8140 |
| | 40% | 3.0074 | **2.5889** | 2.6203 | 2.6083 | 2.6417 | 2.7992 | 2.7992 | 2.7687 | 2.7577 |
| 120 | 5% | 3.0008 | 2.9914 | 2.9926 | 2.9921 | 2.9932 | **2.9712** | 2.9713 | 2.9615 | **2.9602** |
| | 10% | 3.0019 | 2.9712 | 2.9757 | 2.9739 | 2.9780 | **2.9497** | 2.9497 48 | 2.9441 | **2.9414** |
| | 15% | 3.0061 | 2.9359 | 2.9459 | 2.9418 | 2.9511 | **2.9253** | 2.9254 | 2.9107 | **2.9065** |
| | 20% | 3.0059 | 2.8868 | 2.9027 | 2.8962 | 2.9112 | **2.9018** | **2.9019** | 2.8846 | **2.8792** |
| | 30% | 3.0082 | **2.7481** | 2.7761 | 2.7650 | 2.7925 | 2.8493 | 2.8494 | 2.8185 | 2.8103 |
| | 40% | 3.0012 | **2.5815** | 2.6131 | 2.6010 | 2.6345 | 2.7935 | 2.7935 | 2.7547 | 2.7438 |
| 200 | 5% | 3.0032 | 2.9932 | 2.9945 | 2.9939 | 2.9951 | 2.9734 | 2.9734 | 2.9639 | **2.9625** |
| | 10% | 3.0074 | 2.9765 | 2.9811 | 2.9792 | 2.9834 | 2.9559 | 2.9559 | 2.9511 | **2.9483** |
| | 15% | 2.9955 | 2.9258 | 2.9358 | 2.9317 | 2.9410 | 2.9166 | 2.9166 | 2.9025 | **2.8984** |
| | 20% | 2.9955 | 2.8743 | 2.8905 | 2.8839 | 2.8991 | 2.8910 | 2.8910 | 2.8730 | **2.8675** |
| | 30% | 3.0040 | **2.7464** | 2.7738 | 2.7631 | 2.7901 | 2.8478 | 2.8479 | 2.8198 | 2.8116 |
| | 40% | 3.0003 | **2.5814** | 2.6123 | 2.6005 | 2.6335 | 2.7930 | 2.7931 | 2.7612 | 2.7502 |
| 500 | 5% | 3.0074 | 2.9991 | 3.0004 | 2.9998 | 3.0011 | 2.9804 | 2.9805 | 2.9743 | **2.9730** |
| | 10% | 2.9993 | 2.9672 | 2.9722 | 2.9701 | 2.9746 | 2.9466 | 2.9467 | 2.9358 | **2.9331** |
| | 15% | 3.0036 | 2.9335 | 2.9436 | 2.9395 | 2.9489 | 2.9257 | 2.9258 | 2.9100 | **2.9059** |
| | 20% | 2.9929 | 2.8715 | 2.8876 | 2.8811 | 2.8963 | 2.8896 | 2.8898 | 2.8726 | **2.8672** |
| | 30% | 2.9982 | **2.7386** | 2.7661 | 2.7553 | 2.7824 | 2.8421 | 2.8423 | 2.8123 | 2.8041 |
| | 40% | 2.9924 | **2.5700** | 2.6011 | 2.5892 | 2.6224 | 2.7846 | 2.7848 | 2.7479 | **2.7369** |


**Table 14. Mean square errors for the scenario where error variance was 5; sample sizes of 20, 40, 60, 80, 100, 120, 200 and 500; and missing value percentages of 5, 10, 15, 20, 30 and 40% with RSS in missing value estimation methods.**

| Sample sizes | Missing values | Listwise delition | Missing value estimation methods | | | | | | | |
|---|---|---|---|---|---|---|---|---|---|---|
| | | | R-RQ1 | R-RQ3 | AR-CRQ1 | AR-CRQ3 | AR-MRQ1 | AR-MRQ3 | AR-MCRQ1 | AR-MCRQ3 |
| 20 | 5% | 5.0711 | 5.0486 | 5.0485 | 5.0485 | 5.0485 | **5.0014** | 5.0014 | 4.9253 | **4.9243** |
| | 10% | 5.0833 | 5.0433 | 5.0484 | 5.0462 | 5.0508 | **4.9785** | 4.9785 | 4.8089 | **4.8072** |
| | 15% | **2.9942** | 5.0092 | 5.0235 | 5.0174 | 5.0305 | **4.9572** | 4.9572 | 4.7228 | **4.7199** |
| | 20% | 4.9656 | 4.8056 | 4.8298 | 4.8196 | 4.8420 | **4.7985** | **4.7985** | 4.5367 | **4.5329** |
| | 30% | 4.9580 | **4.5395** | 4.5930 | 4.5711 | 4.6219 | 4.6871 | 4.6871 | 4.2551 | **4.2495** |
| | 40% | 4.9739 | **4.2483** | 4.3223 | 4.2937 | 4.3669 | 4.6243 | 4.6243 | 4.0785 | **4.0709** |
| 40 | 5% | 4.9600 | 4.9390 | 4.9401 | 4.9396 | 4.9407 | **4.9018** | **4.9018** | 4.8327 | **4.8317** |
| | 10% | 5.0206 | 4.9708 | 4.9777 | 4.9748 | 4.9811 | **4.9231** | 4.9232 | 4.7761 | **4.7743** |
| | 15% | 5.0117 | 4.9051 | 4.9212 | 4.9145 | 4.9293 | **4.8745** | 4.8745 | 4.6487 | **4.6460** |
| | 20% | 5.0357 | 4.8414 | 4.8690 | 4.8577 | 4.8833 | **4.8552** | **4.8552** | 4.5667 | **4.5630** |
| | 30% | 5.0209 | **4.5768** | 4.6309 | 4.6095 | 4.6612 | 4.7498 | 4.7499 | 4.3145 | **4.3089** |
| | 40% | 5.0459 | **4.2739** | 4.3493 | 4.3210 | 4.3960 | 4.6869 | 4.6869 | 4.0947 | **4.0874** |
| 60 | 5% | 5.0170 | 4.9975 | 4.9992 | 4.9985 | 5.0000 | **4.9597** | 4.9597 | 4.8780 | **4.8771** |
| | 10% | 5.0150 | 4.9615 | 4.9693 | 4.9660 | 4.9731 | **4.9158** | 4.9158 | 4.7593 | **4.7574** |
| | 15% | 4.9993 | 4.8850 | 4.9017 | 4.8948 | 4.9103 | **4.8606** | 4.8606 | 4.6383 | **4.6356** |
| | 20% | 4.9912 | 4.7922 | 4.8201 | 4.8087 | 4.8348 | **4.8124** | **4.8125** | 4.5317 | **4.5279** |
| | 30% | 4.9690 | **4.5187** | 4.5726 | 4.5516 | 4.6032 | 4.6993 | 4.6993 | 4.2623 | **4.2568** |
| | 40% | 4.9923 | **4.2161** | 4.2903 | 4.2627 | 4.3368 | 4.6340 | 4.6341 | 4.0528 | **4.0454** |
| 80 | 5% | 5.0250 | 5.0101 | 5.0119 | 5.0111 | 5.0128 | **4.9744** | 4.9744 | 4.9001 | **4.8991** |
| | 10% | 4.9893 | 4.9377 | 4.9454 | 4.9422 | 4.9492 | **4.8963** | 4.8964 | 4.7523 | **4.7504** |
| | 15% | 5.0016 | 4.8845 | 4.9017 | 4.8946 | 4.9105 | **4.8627** | 4.8628 | 4.6361 | **4.6334** |
| | 20% | 5.0137 | 4.8086 | 4.8375 | 4.8258 | 4.8527 | **4.8327** | **4.8327** | 4.5356 | **4.5319** |
| | 30% | 5.0374 | **4.5716** | 4.6276 | 4.6057 | 4.6593 | 4.7615 | 4.7615 | 4.3053 | **4.2997** |
| | 40% | 5.0519 | **4.2559** | 4.3317 | 4.3036 | 4.3792 | 4.6866 | 4.6867 | 4.0870 | **4.0796** |
| 100 | 5% | 4.9946 | 4.9782 | 4.9801 | 4.9793 | 4.9810 | **4.9435** | 4.9435 | 4.8675 | **4.8666** |
| | 10% | 4.9956 | 4.9441 | 4.9520 | 4.9487 | 4.9559 | **4.9042** | 4.9042 | 4.7573 | **4.7555** |
| | 15% | 4.9684 | 4.8496 | 4.8673 | 4.8600 | 4.8762 | **4.8295** | 4.8295 | 4.5968 | **4.5942** |
| | 20% | 4.9875 | 4.7809 | 4.8099 | 4.7982 | 4.8252 | **4.8070** | 4.8070 | 4.5095 | **4.5059** |
| | 30% | 5.0313 | **4.5690** | 4.6241 | 4.6027 | 4.6556 | 4.7589 | 4.7590 | 4.3063 | **4.3009** |
| | 40% | 4.9917 | **4.1962** | 4.2716 | 4.2436 | 4.3190 | 4.6278 | 4.6278 | 4.0290 | **4.0216** |
| 120 | 5% | 5.0090 | 4.9946 | 4.9965 | 4.9957 | 4.9975 | **4.9602** | 4.9602 | 4.8830 | **4.8821** |
| | 10% | 5.0158 | 4.9623 | 4.9705 | 4.9671 | 4.9745 | **4.9222** | 4.9223 | 4.7697 | **4.7679** |
| | 15% | 5.0009 | 4.8816 | 4.8995 | 4.8921 | 4.9087 | **4.8624** | 4.8625 | 4.6311 | **4.6284** |
| | 20% | 5.0228 | 4.8106 | 4.8403 | 4.8283 | 4.8560 | **4.8386** | **4.8387** | 4.5317 | **4.5281** |
| | 30% | 5.0226 | **4.5606** | 4.6157 | 4.5943 | 4.6471 | 4.7516 | 4.7516 | 4.2997 | **4.2943** |
| | 40% | 4.9806 | **4.1899** | 4.2644 | 4.2369 | 4.3114 | 4.6195 | 4.6196 | 4.0265 | **4.0192** |
| 200 | 5% | 5.0232 | 5.0094 | 5.0115 | 5.0106 | 5.0125 | 4.9767 | 4.9767 | 4.9017 | **4.9008** |
| | 10% | 4.9998 | 4.9477 | 4.9559 | 4.9525 | 4.9600 | 4.9103 | 4.9104 | 4.7636 | **4.7618** |
| | 15% | 5.0208 | 4.9009 | 4.9187 | 4.9114 | 4.9279 | 4.8842 | 4.8843 | 4.6567 | **4.6540** |
| | 20% | 5.0076 | 4.7998 | 4.8288 | 4.8171 | 4.8442 | 4.8297 | 4.8298 | 4.5419 | **4.5381** |
| | 30% | 4.9620 | **4.5005** | 4.5551 | 4.5339 | 4.5863 | 4.6929 | 4.6930 | 4.2515 | **4.2460** |
| | 40% | 4.9959 | **4.2076** | 4.2816 | 4.2542 | 4.3283 | 4.6375 | 4.6376 | 4.0465 | **4.0392** |
| 500 | 5% | 5.0087 | 4.9952 | 4.9974 | 4.9965 | 4.9985 | 4.9636 | 4.9636 | 4.8889 | **4.8880** |
| | 10% | 4.9972 | 4.9435 | 4.9520 | 4.9485 | 4.9562 | 4.9075 | 4.9077 | 4.7573 | **4.7555** |
| | 15% | 4.9951 | 4.8747 | 4.8926 | 4.8853 | 4.9018 | 4.8607 | 4.8609 | 4.6368 | **4.6341** |
| | 20% | 5.0088 | 4.7961 | 4.8257 | 4.8138 | 4.8414 | 4.8294 | 4.8296 | 4.5283 | **4.5247** |
| | 30% | 4.9935 | **4.5280** | 4.5829 | 4.5616 | 4.6143 | 4.7246 | 4.7249 | 4.2787 | **4.2732** |
| | 40% | 5.0199 | **4.2188** | 4.2936 | 4.2661 | 4.3410 | 4.6580 | 4.6584 | 4.0536 | **4.0463** |


**Table 15. Mean square errors for the scenario where error variance was 7; sample sizes of 20, 40, 60, 80, 100, 120, 200, and 500; and missing value percentages of 5, 10, 15, 20, 30 and 40% with RSS in missing value estimation methods.**

| Sample sizes | Missing values | Listwise delition | Missing value estimation methods | | | | | | | |
|---|---|---|---|---|---|---|---|---|---|---|
| | | | R-RQ1 | R-RQ3 | AR-CRQ1 | AR-CRQ3 | AR-MRQ1 | AR-MRQ3 | AR-MCRQ1 | AR-MCRQ3 |
| 20 | 5% | 7.0695 | 7.0593 | 7.0592 | 7.0592 | 7.0592 | **6.9882** | 6.9882 | 6.8233 | **6.8229** |
| | 10% | 6.9631 | 6.8924 | 6.8994 | 6.8964 | 6.9027 | **6.8002** | 6.8002 | 6.4964 | **6.4955** |
| | 15% | **7.0693** | 6.9228 | 6.9434 | 6.9347 | 6.9536 | **6.8483** | 6.8483 | 6.3740 | **6.3729** |
| | 20% | 7.0401 | 6.7909 | 6.8272 | 6.8121 | 6.8456 | **6.7796** | **6.7796** | 6.2163 | **6.2144** |
| | 30% | 6.9827 | **6.3827** | 6.4620 | 6.4297 | 6.5046 | 6.5999 | 6.5999 | 5.7546 | **5.7517** |
| | 40% | 7.0797 | **5.9908** | 6.1091 | 6.0643 | 6.1795 | 6.5724 | 6.5725 | 5.4332 | **5.4295** |
| 40 | 5% | 6.9658 | 6.9448 | 6.9465 | 6.9457 | 6.9472 | **6.8921** | **6.8921** | 6.7583 | **6.7577** |
| | 10% | 7.0406 | 6.9715 | 6.9818 | 6.9775 | 6.9868 | **6.8987** | 6.8988 | 6.5957 | **6.5947** |
| | 15% | 7.0306 | 6.8693 | 6.8927 | 6.8830 | 6.9045 | **6.8249** | 6.8249 | 6.3813 | **6.3800** |
| | 20% | 6.9718 | 6.6954 | 6.7351 | 6.7189 | 6.7556 | **6.7150** | **6.7150** | 6.1549 | **6.1530** |
| | 30% | 6.9825 | **6.3428** | 6.4238 | 6.3919 | 6.4688 | 6.5987 | 6.5987 | 5.7377 | **5.7350** |
| | 40% | 7.0371 | **5.9162** | 6.0328 | 5.9893 | 6.1035 | 6.5296 | 6.5296 | 5.4066 | **5.4028** |
| 60 | 5% | 6.9832 | 6.9655 | 6.9677 | 6.9667 | 6.9688 | **6.9142** | 6.9142 | 6.7709 | **6.7704** |
| | 10% | 7.0218 | 6.9521 | 6.9626 | 6.9582 | 6.9678 | **6.8903** | 6.8904 | 6.6076 | **6.6067** |
| | 15% | 7.0291 | 6.8696 | 6.8937 | 6.8838 | 6.9059 | **6.8344** | 6.8345 | 6.4051 | **6.4037** |
| | 20% | 6.9670 | 6.6782 | 6.7200 | 6.7030 | 6.7417 | **6.7085** | **6.7085** | 6.1321 | **6.1302** |
| | 30% | 7.0818 | **6.4360** | 6.5162 | 6.4852 | 6.5614 | 6.7010 | 6.7011 | 5.8524 | **5.8496** |
| | 40% | 7.0394 | **5.8989** | 6.0161 | 5.9730 | 6.0879 | 6.5301 | 6.5301 | 5.3841 | **5.3805** |
| 80 | 5% | 6.9899 | 6.9697 | 6.9722 | 6.9711 | 6.9734 | **6.9194** | 6.9194 | 6.7732 | **6.7728** |
| | 10% | 6.9777 | 6.9009 | 6.9123 | 6.9075 | 6.9179 | **6.8393** | 6.8393 | 6.5403 | **6.5394** |
| | 15% | 7.0737 | 6.9145 | 6.9385 | 6.9286 | 6.9506 | **6.8842** | 6.8843 | 6.4678 | **6.4664** |
| | 20% | 7.0118 | 6.7188 | 6.7608 | 6.7438 | 6.7828 | **6.7537** | **6.7538** | 6.1746 | **6.1728** |
| | 30% | 6.9853 | **6.3212** | 6.4038 | 6.3716 | 6.4501 | 6.5975 | 6.5976 | 5.7268 | **5.7241** |
| | 40% | 6.9988 | **5.8673** | 5.9827 | 5.9403 | 6.0536 | 6.4954 | 6.4955 | 5.3743 | **5.3705** |
| 100 | 5% | 7.0047 | 6.9854 | 6.9881 | 6.9870 | 6.9895 | **6.9345** | 6.9345 | 6.7804 | **6.7800** |
| | 10% | 7.0240 | 6.9497 | 6.9612 | 6.9564 | 6.9669 | **6.8905** | 6.8905 | 6.5914 | **6.5906** |
| | 15% | 7.0274 | 6.8649 | 6.8896 | 6.8795 | 6.9022 | **6.8364** | 6.8365 | 6.4047 | **6.4034** |
| | 20% | 7.0175 | 6.7189 | 6.7615 | 6.7443 | 6.7838 | **6.7571** | 6.7572 | 6.1720 | **6.1703** |
| | 30% | 7.0203 | **6.3592** | 6.4412 | 6.4094 | 6.4874 | 6.6370 | 6.6371 | 5.7685 | **5.7660** |
| | 40% | 7.0213 | **5.8620** | 5.9803 | 5.9367 | 6.0530 | 6.5077 | 6.5079 | 5.3489 | **5.3454** |
| 120 | 5% | 7.0325 | 7.0119 | 7.0147 | 7.0135 | 7.0161 | **6.9622** | 6.9623 | 6.8126 | **6.8121** |
| | 10% | 7.0585 | 6.9864 | 6.9978 | 6.9931 | 7.0035 | **6.9297** | 6.9298 | 6.6410 | **6.6401** |
| | 15% | 7.0147 | 6.8496 | 6.8746 | 6.8643 | 6.8872 | **6.8227** | 6.8228 | 6.3931 | **6.3918** |
| | 20% | 7.0363 | 6.7418 | 6.7839 | 6.7670 | 6.8061 | **6.7815** | **6.7816** | 6.2045 | **6.2028** |
| | 30% | 6.9940 | **6.3251** | 6.4079 | 6.3758 | 6.4546 | 6.6075 | 6.6076 | 5.7314 | **5.7289** |
| | 40% | 6.9648 | **5.8211** | 5.9368 | 5.8945 | 6.0082 | 6.4585 | 6.4587 | 5.3270 | **5.3232** |
| 200 | 5% | 7.0262 | 7.0055 | 7.0086 | 7.0073 | 7.0101 | 6.9568 | 6.9568 | 6.8020 | **6.8016** |
| | 10% | 6.9985 | 6.9262 | 6.9376 | 6.9328 | 6.9432 | 6.8737 | 6.8738 | 6.5958 | **6.5949** |
| | 15% | 7.0163 | 6.8484 | 6.8737 | 6.8634 | 6.8867 | 6.8248 | 6.8249 | 6.3940 | **6.3927** |
| | 20% | 6.9913 | 6.6910 | 6.7338 | 6.7165 | 6.7564 | 6.7351 | 6.7352 | 6.1578 | **6.1560** |
| | 30% | 7.0136 | **6.3414** | 6.4240 | 6.3921 | 6.4707 | 6.6278 | 6.6280 | 5.7588 | **5.7561** |
| | 40% | 7.0431 | **5.8783** | 5.9961 | 5.9530 | 6.0688 | 6.5305 | 6.5307 | 5.3722 | **5.3687** |
| 500 | 5% | 6.9748 | 6.9558 | 6.9589 | 6.9576 | 6.9605 | 6.9105 | 6.9106 | 6.7653 | **6.7648** |
| | 10% | 6.9991 | 6.9226 | 6.9348 | 6.9297 | 6.9408 | 6.8705 | 6.8707 | 6.5790 | **6.5781** |
| | 15% | 7.0152 | 6.8441 | 6.8699 | 6.8594 | 6.8831 | 6.8239 | 6.8242 | 6.3879 | **6.3866** |
| | 20% | 7.0106 | 6.7061 | 6.7493 | 6.7320 | 6.7721 | 6.7546 | 6.7550 | 6.1739 | **6.1721** |
| | 30% | 6.9618 | **6.2852** | 6.3681 | 6.3361 | 6.4151 | 6.5772 | 6.5777 | 5.7068 | **5.7041** |
| | 40% | 6.9924 | **5.8204** | 5.9382 | 5.8952 | 6.0111 | 6.4797 | 6.4802 | 5.3187 | **5.3151** |


**Table 16. Mean square errors for the scenario where error variance was 9; sample sizes of 20, 40, 60, 80, 100, 120, 200, and 500; and missing value percentages of 5, 10, 15, 20, 30 and 40% with RSS in missing value estimation methods.**

| Sample sizes | Missing values | Listwise delition | Missing value estimation methods | | | | | | | |
|---|---|---|---|---|---|---|---|---|---|---|
| | | | R-RQ1 | R-RQ3 | AR-CRQ1 | AR-CRQ3 | AR-MRQ1 | AR-MRQ3 | AR-MCRQ1 | AR-MCRQ3 |
| 20 | 5% | 9.0487 | 9.0347 | 9.0346 | 9.0347 | 9.0346 | **8.9512** | 8.9512 | 8.7425 | **8.7424** |
| | 10% | 9.1216 | 9.0419 | 9.0511 | 9.0471 | 9.0554 | **8.9231** | 8.9231 | **8.4792** | **8.4792** |
| | 15% | **9.1263** | 8.9445 | 8.9705 | 8.9596 | 8.9835 | **8.8513** | 8.8514 | **8.1871** | **8.1872** |
| | 20% | 9.1082 | 8.7787 | 8.8269 | 8.8068 | 8.8512 | **8.7644** | **8.7644** | **7.9146** | **7.9147** |
| | 30% | 9.0519 | **8.2758** | 8.3804 | 8.3380 | 8.4363 | 8.5596 | 8.5596 | 7.2954 | **7.2954** |
| | 40% | 9.0489 | **7.6315** | 7.7889 | 7.7305 | 7.8826 | 8.3975 | 8.3975 | **6.7114** | **6.7115** |
| 40 | 5% | 9.0618 | 9.0374 | 9.0397 | 9.0387 | 9.0407 | **8.9652** | **8.9652** | **8.7460** | **8.7460** |
| | 10% | 9.0585 | 8.9675 | 8.9805 | 8.9750 | 8.9868 | **8.8758** | 8.8758 | 8.4377 | **8.4376** |
| | 15% | 9.1463 | 8.9434 | 8.9732 | 8.9610 | 8.9883 | **8.8863** | 8.8863 | 8.2465 | **8.2465** |
| | 20% | 9.0675 | 8.7005 | 8.7543 | 8.7322 | 8.7820 | **8.7272** | **8.7272** | **7.8576** | **7.8577** |
| | 30% | 8.9323 | **8.1158** | 8.2214 | 8.1798 | 8.2795 | 8.4463 | 8.4463 | 7.2014 | **7.2013** |
| | 40% | 9.0316 | **7.5425** | 7.7022 | 7.6434 | 7.7986 | 8.3686 | 8.3687 | 6.6802 | **6.6799** |
| 60 | 5% | 9.1024 | 9.0762 | 9.0791 | 9.0779 | 9.0805 | **9.0099** | 9.0099 | **8.7948** | **8.7949** |
| | 10% | 9.0273 | 8.9356 | 8.9501 | 8.9440 | 8.9571 | **8.8504** | 8.8504 | **8.4028** | **8.4028** |
| | 15% | 9.1263 | 8.9162 | 8.9483 | 8.9351 | 8.9645 | **8.8688** | 8.8688 | 8.2157 | **8.2157** |
| | 20% | 9.0341 | 8.6471 | 8.7032 | 8.6803 | 8.7323 | **8.6876** | **8.6877** | **7.8075** | **7.8075** |
| | 30% | 9.0503 | **8.2133** | 8.3191 | 8.2783 | 8.3783 | 8.5603 | 8.5604 | 7.3119 | **7.3118** |
| | 40% | 8.9817 | **7.4790** | 7.6387 | 7.5800 | 7.7355 | 8.3201 | 8.3202 | 6.6286 | **6.6285** |
| 80 | 5% | 9.0018 | 8.9773 | 8.9806 | 8.9792 | 8.9822 | **8.9112** | 8.9112 | 8.6939 | **8.6938** |
| | 10% | 8.9581 | 8.8645 | 8.8788 | 8.8728 | 8.8858 | **8.7871** | 8.7871 | 8.3708 | **8.3706** |
| | 15% | 9.0817 | 8.8641 | 8.8973 | 8.8837 | 8.9142 | **8.8217** | 8.8218 | **8.1475** | **8.1477** |
| | 20% | 8.8724 | 8.4981 | 8.5522 | 8.5304 | 8.5805 | **8.5431** | **8.5432** | 7.7013 | **7.7012** |
| | 30% | 9.0138 | **8.1621** | 8.2701 | 8.2281 | 8.3304 | 8.5209 | 8.5210 | 7.2562 | **7.2561** |
| | 40% | 9.0067 | **7.4864** | 7.6472 | 7.5881 | 7.7448 | 8.3411 | 8.3412 | 6.6400 | **6.6399** |
| 100 | 5% | 8.9524 | 8.9275 | 8.9307 | 8.9293 | 8.9322 | **8.8680** | 8.8681 | 8.6711 | **8.6709** |
| | 10% | 8.9776 | 8.8862 | 8.9007 | 8.8947 | 8.9079 | **8.8116** | 8.8117 | **8.3873** | **8.3874** |
| | 15% | 8.9764 | 8.7649 | 8.7972 | 8.7839 | 8.8136 | **8.7274** | 8.7275 | **8.0858** | **8.0859** |
| | 20% | 9.0657 | 8.6792 | 8.7349 | 8.7124 | 8.7641 | **8.7292** | 8.7293 | **7.8648** | **7.8649** |
| | 30% | 8.9626 | **8.0959** | 8.2053 | 8.1630 | 8.2666 | 8.4638 | 8.4639 | **7.1761** | **7.1762** |
| | 40% | 9.0209 | **7.4968** | 7.6576 | 7.5987 | 7.7554 | 8.3561 | 8.3563 | 6.6504 | **6.6504** |
| 120 | 5% | 9.0431 | 9.0188 | 9.0224 | 9.0209 | 9.0242 | **8.9543** | 8.9544 | **8.7297** | **8.7298** |
| | 10% | 9.0522 | 8.9576 | 8.9726 | 8.9663 | 8.9800 | **8.8834** | 8.8834 | **8.4518** | **8.4518** |
| | 15% | 8.9858 | 8.7703 | 8.8032 | 8.7896 | 8.8198 | **8.7348** | 8.7349 | **8.0891** | **8.0892** |
| | 20% | 8.9380 | 8.5485 | 8.6043 | 8.5819 | 8.6337 | **8.6011** | **8.6012** | **7.7381** | **7.7382** |
| | 30% | 8.9930 | **8.1324** | 8.2407 | 8.1988 | 8.3015 | 8.4994 | 8.4995 | 7.2338 | **7.2337** |
| | 40% | 8.9832 | **7.4734** | 7.6318 | 7.5742 | 7.7286 | 8.3263 | 8.3265 | 6.6411 | **6.6410** |
| 200 | 5% | 9.0187 | 8.9942 | 8.9980 | 8.9964 | 8.9998 | 8.9340 | 8.9341 | 8.7223 | **8.7222** |
| | 10% | 9.0430 | 8.9460 | 8.9614 | 8.9550 | 8.9690 | 8.8749 | 8.8750 | 8.4466 | **8.4465** |
| | 15% | 9.0068 | 8.7894 | 8.8224 | 8.8089 | 8.8393 | 8.7582 | 8.7584 | 8.1202 | **8.1202** |
| | 20% | 9.0213 | 8.6313 | 8.6874 | 8.6648 | 8.7168 | 8.6890 | 8.6892 | **7.8329** | **7.8330** |
| | 30% | 9.0458 | **8.1704** | 8.2801 | 8.2379 | 8.3419 | 8.5477 | 8.5479 | **7.2603** | **7.2604** |
| | 40% | 8.9758 | **7.4477** | 7.6077 | 7.5494 | 7.7054 | 8.3143 | 8.3146 | **6.6105** | **6.6105** |
| 500 | 5% | 9.0099 | 8.9859 | 8.9898 | 8.9882 | 8.9918 | 8.9279 | 8.9280 | 8.7172 | **8.7172** |
| | 10% | 8.9914 | 8.8935 | 8.9092 | 8.9027 | 8.9170 | 8.8262 | 8.8264 | **8.3968** | **8.3968** |
| | 15% | 8.9688 | 8.7504 | 8.7835 | 8.7700 | 8.8004 | 8.7244 | 8.7247 | **8.0905** | **8.0905** |
| | 20% | 8.9864 | 8.5912 | 8.6479 | 8.6251 | 8.6777 | 8.6548 | 8.6553 | **7.7911** | **7.7912** |
| | 30% | 9.0008 | **8.1260** | 8.2353 | 8.1932 | 8.2969 | 8.5079 | 8.5085 | **7.2293** | **7.2294** |
| | 40% | 9.0098 | **7.4692** | 7.6297 | 7.5714 | 7.7280 | 8.3471 | 8.3478 | **6.6334** | **6.6334** |


**Table 17. Mean absolute percentage errors for the scenario where error variance was 1; sample sizes of 20, 40, 60, 80, 100, 120, 200, and 500; and missing value percentages of 5, 10, 15, 20, 30, and 40% with RSS in missing value estimation methods.**

| Sample sizes | Missing values | Listwise delition | Missing value estimation methods | | | | | | | |
|---|---|---|---|---|---|---|---|---|---|---|
| | | | R-RQ1 | R-RQ3 | AR-CRQ1 | AR-CRQ3 | AR-MRQ1 | AR-MRQ3 | AR-MCRQ1 | AR-MCRQ3 |
| 20 | 5% | 95.3747 | 94.7604 | 94.7627 | 94.7618 | 94.7638 | **94.4482** | 94.4482 | 94.7862 | **94.7666** |
| | 10% | 99.6573 | 96.8960 | 96.9349 | 96.9184 | 96.9537 | **96.5133** | 96.5133 | **97.4364** | **97.3920** |
| | 15% | **77.0905** | 77.0238 | 77.1048 | 77.0702 | 77.1448 | **76.7960** | 76.7961 | **79.1349** | **79.0612** |
| | 20% | 75.8631 | **75.0395** | 75.2156 | 75.1380 | 75.3015 | **75.0672** | **75.0673** | **76.8335** | **76.7519** |
| | 30% | 80.7803 | **78.2789** | 78.7097 | 78.5346 | 78.9515 | 79.6078 | 79.6079 | 79.0293 | **78.9406** |
| | 40% | 110.4068 | **106.7198** | 107.1859 | 107.0251 | 107.5248 | 109.8223 | 109.8224 | **109.9464** | **109.8046** |
| 40 | 5% | 111.2744 | 110.2455 | 110.2557 | 110.2512 | 110.2605 | **109.9917** | **109.9917** | **110.4391** | **110.4185** |
| | 10% | 86.3833 | 85.8688 | 85.9092 | 85.8923 | 85.9293 | **85.6247** | 85.6247 | 87.2885 | **87.2359** |
| | 15% | 92.5528 | 93.6412 | 93.8916 | 93.7866 | 94.0159 | **93.3115** | 93.3117 | **88.8422** | **88.8538** |
| | 20% | 73.2236 | **71.7732** | 71.9195 | 71.8592 | 71.9991 | **71.8785** | **71.8786** | **74.4370** | **74.3458** |
| | 30% | 93.7663 | **85.9955** | 87.0085 | 86.5975 | 87.5523 | 89.2722 | 89.2727 | **77.9898** | **78.0354** |
| | 40% | 84.6630 | **78.4458** | 78.9665 | 78.7490 | 79.3038 | 81.7218 | 81.7221 | 81.5996 | **81.4704** |
| 60 | 5% | 86.5444 | 85.8510 | 85.8616 | 85.8571 | 85.8667 | **85.6399** | 85.6400 | **86.3362** | **86.3120** |
| | 10% | 88.3480 | 89.5389 | 89.5872 | 89.5670 | 89.6111 | **89.2925** | 89.2925 | **90.5641** | **90.5172** |
| | 15% | 110.6973 | 112.6806 | 113.3106 | 113.0526 | 113.6269 | **112.0611** | 112.0619 | **92.7444** | **92.9263** |
| | 20% | 86.0313 | **85.4402** | 85.6186 | 85.5447 | 85.7139 | **85.5969** | **85.5971** | **87.5550** | **87.4670** |
| | 30% | 72.7656 | **68.9958** | 69.2683 | 69.1587 | 69.4367 | 70.0357 | 70.0359 | 73.0802 | **72.9461** |
| | 40% | 79.1220 | **73.6668** | 74.0693 | 73.8928 | 74.3378 | 76.4506 | 76.4510 | 77.9946 | **77.8465** |
| 80 | 5% | 226.0563 | 222.6887 | 222.7046 | 222.6980 | 222.7123 | **222.4059** | **222.4059** | 222.4830 | **222.4655** |
| | 10% | 169.0548 | 171.0901 | 171.1397 | 171.1187 | 171.1643 | **170.8545** | 170.8547 | 172.1611 | **172.1130** |
| | 15% | 76.3482 | 74.9303 | 75.0295 | 74.9885 | 75.0809 | **74.8255** | 74.8257 | **76.8243** | **76.7502** |
| | 20% | 81.0644 | **78.4086** | 78.7821 | 78.6291 | 78.9763 | **78.7734** | **78.7739** | 74.8995 | **74.8836** |
| | 30% | 71.2061 | **68.1303** | 68.4284 | 68.3086 | 68.6115 | 69.2638 | 69.2642 | 71.7409 | **71.6106** |
| | 40% | 97.9712 | **89.8632** | 90.6752 | 90.3651 | 91.1942 | 94.6932 | 94.6938 | 89.2844 | **89.2156** |
| 100 | 5% | 108.4177 | 107.9050 | 107.9182 | 107.9127 | 107.9246 | **107.6892** | 107.6893 | 108.2595 | **108.2365** |
| | 10% | 69.5005 | 68.9233 | 68.9777 | 68.9548 | 69.0047 | **68.6761** | 68.6762 | **69.6230** | **69.5800** |
| | 15% | 74.2229 | 72.8707 | 72.9958 | 72.9431 | 73.0593 | **72.7547** | 72.7550 | **73.8859** | **73.8233** |
| | 20% | 104.0180 | **102.9827** | 103.1763 | 103.0983 | 103.2813 | **103.1754** | 103.1758 | **104.8035** | **104.7176** |
| | 30% | 137.3405 | **134.3159** | 134.7270 | 134.5703 | 134.9775 | 135.8635 | 135.8641 | **135.9066** | **135.8021** |
| | 40% | 75.6245 | **70.7699** | 71.0220 | 70.9180 | 71.2294 | 73.0642 | 73.0647 | 76.2107 | **76.0358** |
| 120 | 5% | 117.4171 | 110.8555 | 111.0346 | 110.9610 | 111.1227 | **107.9014** | 107.9031 | **87.4125** | **87.6371** |
| | 10% | 75.8867 | 75.2628 | 75.3161 | 75.2939 | 75.3428 | **75.0275** | 75.0277 | **76.0948** | **76.0498** |
| | 15% | 89.8514 | 86.4218 | 86.5842 | 86.5176 | 86.6674 | **86.2737** | 86.2741 | **85.7168** | **85.6747** |
| | 20% | 74.7574 | **73.1423** | 73.3105 | 73.2414 | 73.4024 | **73.3142** | **73.3146** | **75.7144** | **75.6202** |
| | 30% | 79.5506 | **75.8699** | 76.2900 | 76.1173 | 76.5315 | 77.3816 | 77.3822 | 77.6837 | **77.5819** |
| | 40% | 77.2808 | **71.7693** | 72.0881 | 71.9619 | 72.3365 | 74.4156 | 74.4162 | 76.5590 | **76.3962** |
| 200 | 5% | 80.3350 | 80.1389 | 80.1524 | 80.1467 | 80.1589 | **79.9440** | 79.9442 | 80.6090 | **80.5847** |
| | 10% | 91.9970 | 91.6636 | 91.7127 | 91.6922 | 91.7373 | **91.4564** | 91.4567 | 92.8283 | **92.7799** |
| | 15% | **95.0977** | 95.5675 | 95.7074 | 95.6501 | 95.7799 | 95.4504 | 95.4510 | 95.9424 | **95.8873** |
| | 20% | 82.1227 | 80.6674 | **80.8233** | 80.7594 | 80.9089 | 80.8356 | 80.8361 | **83.5682** | **83.4690** |
| | 30% | 141.3563 | **138.8085** | 139.0878 | 138.9755 | 139.2628 | 139.9241 | 139.9249 | **142.8774** | **142.7394** |
| | 40% | 79.7663 | **74.5680** | 74.8671 | 74.7435 | 75.1004 | 77.1151 | 77.1161 | **79.6292** | **79.4607** |
| 500 | 5% | 90.2270 | 89.7014 | 89.7164 | 89.7101 | 89.7236 | **89.4990** | 89.4994 | 90.0571 | **90.0338** |
| | 10% | 77.4495 | 76.9062 | 76.9657 | 76.9409 | 76.9956 | **76.6642** | 76.6651 | **77.4929** | **77.4490** |
| | 15% | 120.0434 | 119.3034 | 119.4234 | 119.3741 | 119.4859 | **119.2154** | 119.2166 | **120.5326** | **120.4656** |
| | 20% | 77.2466 | **75.6143** | 75.8153 | 75.7340 | 75.9250 | 75.8454 | 75.8471 | **77.2497** | **77.1649** |
| | 30% | 149.4028 | **146.9771** | 147.2354 | 147.1310 | 147.3982 | 148.0304 | 148.0323 | **151.3856** | **151.2424** |
| | 40% | 115.3632 | **105.2425** | 106.1978 | 105.8445 | 106.8095 | 110.9156 | 110.9203 | **103.0001** | **102.9567** |


**Table 18. Mean absolute percentage errors for the scenario where error variance was 3; sample sizes of 20, 40, 60, 80, 100, 120, 200, and 500; and missing value percentages of 5, 10, 15, 20, 30, and 40% with RSS in missing value estimation methods.**

| Sample sizes | Missing values | Listwise delition | Missing value estimation methods | | | | | | | |
|---|---|---|---|---|---|---|---|---|---|---|
| | | | R-RQ1 | R-RQ3 | AR-CRQ1 | AR-CRQ3 | AR-MRQ1 | AR-MRQ3 | AR-MCRQ1 | AR-MCRQ3 |
| 20 | 5% | 148.7004 | 147.4176 | 147.4267 | 147.4225 | 147.4307 | **146.6588** | 146.6589 | **144.1128** | **144.12** |
| | 10% | 148.2063 | 146.6777 | 146.7414 | 146.7146 | 146.7722 | **146.0009** | 146.001 | **143.7579** | **143.7333** |
| | 15% | **131.9026** | 129.9733 | 130.1293 | 130.063 | 130.2052 | **129.5127** | 129.5128 | **126.9238** | **126.8864** |
| | 20% | 123.6774 | **120.8053** | 121.2421 | 121.0556 | 121.4557 | **120.882** | **120.8822** | **113.0267** | **113.0287** |
| | 30% | 139.765 | **132.5015** | 133.4867 | 133.1049 | 134.0208 | 135.3684 | 135.3686 | **122.4487** | **122.4526** |
| | 40% | 224.5832 | **210.2758** | 211.6033 | 211.1259 | 212.3962 | 216.7174 | 216.7176 | **203.8819** | **203.8312** |
| 40 | 5% | 216.3412 | 215.8688 | 215.8878 | 215.8797 | 215.8968 | **215.3702** | **215.3703** | **213.8529** | **213.8445** |
| | 10% | 143.3276 | 142.7209 | 142.8026 | 142.7686 | 142.8429 | **142.1898** | 142.1899 | 140.0794 | **140.0546** |
| | 15% | 149.2463 | 145.4361 | 145.8217 | 145.6613 | 146.0123 | **144.883** | 144.8833 | **133.8658** | **133.9255** |
| | 20% | 134.3183 | **131.7462** | 132.1147 | 131.9636 | 132.3024 | **131.9882** | **131.9885** | **127.0279** | **126.9879** |
| | 30% | 150.9869 | **138.8453** | 140.5751 | 139.8736 | 141.4759 | 144.1968 | 144.1975 | **118.7213** | **118.8908** |
| | 40% | 170.0963 | **156.3779** | 158.0601 | 157.4238 | 159.0274 | 164.5536 | 164.5541 | **146.9916** | **147.0083** |
| 60 | 5% | 188.0755 | 187.9218 | 187.9416 | 187.9332 | 187.9511 | **187.5072** | 187.5073 | **186.3435** | **186.3325** |
| | 10% | 216.0167 | 213.7736 | 213.8692 | 213.8296 | 213.9163 | **213.2418** | 213.2421 | **210.671** | **210.6512** |
| | 15% | 178.455 | 178.1401 | 178.9928 | 178.6437 | 179.4192 | **177.2653** | 177.2665 | **147.9714** | **148.248** |
| | 20% | 149.6323 | **146.4814** | 146.9416 | 146.7571 | 147.1795 | **146.88** | **146.8804** | **139.3871** | **139.3746** |
| | 30% | 166.8697 | **161.9126** | 162.7194 | 162.4096 | 163.168 | 164.568 | 164.5685 | 156.3299 | **156.2768** |
| | 40% | 163.3673 | **143.6991** | 146.3582 | 145.2955 | 147.8135 | 156.0269 | 156.028 | **125.2971** | **125.4925** |
| 80 | 5% | 192.3991 | 189.5935 | 189.6167 | 189.607 | 189.628 | **189.1522** | **189.1523** | 187.7387 | **187.7305** |
| | 10% | 148.9303 | 150.4853 | 150.5843 | 150.5433 | 150.6332 | **149.9741** | 149.9744 | 147.4152 | **147.3932** |
| | 15% | 130.0289 | 128.7169 | 128.9364 | 128.8461 | 129.0468 | **128.4726** | 128.473 | **124.7379** | **124.7095** |
| | 20% | 135.0307 | **132.0583** | 132.7596 | 132.4725 | 133.1151 | **132.7214** | **132.7224** | **118.4923** | **118.5653** |
| | 30% | 118.0557 | **112.0078** | 112.8233 | 112.5106 | 113.2766 | 114.7215 | 114.7222 | 106.4259 | **106.3691** |
| | 40% | 172.0073 | **154.5334** | 156.7095 | 155.9134 | 157.9804 | 165.2652 | 165.2664 | **139.9372** | **140.0389** |
| 100 | 5% | 153.285 | 153.356 | 153.3806 | 153.3702 | 153.3924 | **152.9225** | 152.9227 | 151.5672 | **151.5574** |
| | 10% | 121.301 | 120.5148 | 120.6154 | 120.5735 | 120.6648 | **120.0171** | 120.0175 | **117.4126** | **117.393** |
| | 15% | 125.0877 | 122.8623 | 123.1318 | 123.0193 | 123.2651 | **122.5969** | 122.5976 | **117.1411** | **117.1325** |
| | 20% | 184.055 | **179.833** | 180.2444 | 180.0813 | 180.4598 | **180.2335** | 180.2342 | **174.2523** | **174.222** |
| | 30% | 271.7702 | **265.8026** | 266.6883 | 266.349 | 267.1789 | 268.7607 | 268.7616 | **259.0056** | **258.9684** |
| | 40% | 130.6342 | **117.6341** | 119.1487 | 118.6001 | 120.0562 | 125.4038 | 125.4049 | 110.0513 | **110.0294** |
| 120 | 5% | 157.4864 | 151.45 | 151.5896 | 151.5322 | 151.6581 | **149.1282** | 149.1295 | **133.1545** | **133.3172** |
| | 10% | 130.2704 | 129.4296 | 129.5362 | 129.4919 | 129.5888 | **128.9172** | 128.9177 | **126.0356** | **126.0213** |
| | 15% | 165.86 | 160.5148 | 160.9261 | 160.7637 | 161.1384 | **160.1022** | 160.1033 | **148.4212** | **148.4798** |
| | 20% | 144.6316 | **142.3017** | 142.7067 | 142.543 | 142.9166 | **142.7079** | **142.7087** | **137.0833** | **137.0507** |
| | 30% | 135.7536 | **129.3199** | 130.1618 | 129.8328 | 130.6232 | 132.1191 | 132.1202 | 123.6873 | **123.6338** |
| | 40% | 127.3288 | **114.9126** | 116.331 | 115.8186 | 117.1866 | 122.2779 | 122.2792 | 108.2373 | **108.1999** |
| 200 | 5% | 144.883 | 144.4322 | 144.4575 | 144.4468 | 144.4696 | **144.0323** | 144.0327 | 142.8001 | **142.7888** |
| | 10% | 148.8196 | 148.0708 | 148.1723 | 148.1302 | 148.2225 | **147.6081** | 147.6088 | 145.1979 | **145.1765** |
| | 15% | **173.2969** | 170.5753 | 171.1511 | 170.9203 | 171.4447 | 170.0699 | 170.0725 | **152.2027** | **152.3367** |
| | 20% | 140.0196 | 137.4428 | **137.8296** | 137.6736 | 138.0303 | 137.8534 | 137.8547 | **132.9558** | **132.9127** |
| | 30% | 189.0135 | **183.2443** | 184.0676 | 183.753 | 184.5276 | 186.0512 | 186.053 | **177.6797** | **177.6249** |
| | 40% | 145.2864 | **133.1912** | 134.5667 | 134.0675 | 135.3964 | 140.3636 | 140.3658 | **127.1514** | **127.103** |
| 500 | 5% | 165.2871 | 164.8702 | 164.8965 | 164.8854 | 164.9093 | **164.4794** | 164.4803 | 163.26 | **163.2485** |
| | 10% | 153.5105 | 152.5538 | 152.6609 | 152.6166 | 152.7139 | **152.0908** | 152.0926 | **149.4424** | **149.4222** |
| | 15% | 290.5566 | 287.9215 | 288.1717 | 288.0699 | 288.2987 | **287.73** | 287.7328 | **283.1057** | **283.084** |
| | 20% | 140.6797 | **137.9772** | 138.4022 | 138.2325 | 138.624 | 138.4612 | 138.4646 | **132.4623** | **132.4306** |
| | 30% | 143.7811 | **138.2499** | 139.0236 | 138.727 | 139.4562 | 140.9199 | 140.924 | **133.6638** | **133.5954** |
| | 40% | 160.6603 | **147.5575** | 149.0514 | 148.5117 | 149.9505 | 155.3527 | 155.3585 | **140.2399** | **140.2145** |


**Table 19. Mean absolute percentage errors for the scenario where error variance was 5; sample sizes of 20, 40, 60, 80, 100, 120, 200, and 500; and missing value percentages of 5, 10, 15, 20, 30, and 40% with RSS in missing value estimation methods.**

| Sample sizes | Missing values | Listwise delition | Missing value estimation methods | | | | | | | |
|---|---|---|---|---|---|---|---|---|---|---|
| | | | R-RQ1 | R-RQ3 | AR-CRQ1 | AR-CRQ3 | AR-MRQ1 | AR-MRQ3 | AR-MCRQ1 | AR-MCRQ3 |
| 20 | 5% | 203.4985 | 203.6339 | 203.6412 | 203.6383 | 203.6448 | **202.8757** | 202.8757 | **200.0515** | **200.0527** |
| | 10% | 232.8664 | 233.0204 | 233.1128 | 233.0747 | 233.1582 | **232.0337** | 232.0338 | **226.3982** | **226.4118** |
| | 15% | **172.9467** | 171.1848 | 171.394 | 171.3073 | 171.4973 | **170.5421** | 170.5423 | **164.5503** | **164.5355** |
| | 20% | 161.2985 | **159.1848** | 159.7018 | 159.4849 | 159.9567 | **159.244** | **159.2442** | **147.859** | **147.8945** |
| | 30% | 181.5617 | **172.2612** | 173.3872 | 172.9302 | 173.9723 | 175.3947 | 175.395 | **159.5117** | **159.5387** |
| | 40% | 418.1684 | **407.0691** | 408.9114 | 408.2453 | 409.9874 | 415.691 | 415.6912 | **393.9917** | **394.0337** |
| 40 | 5% | 250.7755 | 251.1941 | 251.2146 | 251.2058 | 251.2243 | **250.6466** | **250.6467** | **248.5604** | **248.5564** |
| | 10% | 165.4405 | 163.9509 | 164.0606 | 164.0146 | 164.1139 | **163.2386** | 163.2388 | 158.7511 | **158.7482** |
| | 15% | 177.7255 | 178.142 | 178.5985 | 178.4099 | 178.8251 | **177.4586** | 177.4591 | **162.6021** | **162.6984** |
| | 20% | 170.865 | **167.9483** | 168.4887 | 168.2686 | 168.7632 | **168.3049** | **168.3053** | **157.2995** | **157.3227** |
| | 30% | 176.3063 | **166.4843** | 168.2611 | 167.5666 | 169.2101 | 172.026 | 172.0267 | **143.5701** | **143.749** |
| | 40% | 193.1219 | **175.6379** | 177.8022 | 176.9891 | 179.0332 | 185.8455 | 185.8461 | **160.4514** | **160.5423** |
| 60 | 5% | 192.0564 | 191.1502 | 191.1809 | 191.1681 | 191.1958 | **190.5018** | 190.502 | **187.5034** | **187.5107** |
| | 10% | 215.7246 | 213.4402 | 213.5543 | 213.5066 | 213.6099 | **212.7942** | 212.7945 | **208.6318** | **208.6247** |
| | 15% | 232.7915 | 230.6326 | 231.8077 | 231.3271 | 232.3945 | **229.4203** | 229.4219 | **186.4093** | **186.8393** |
| | 20% | 170.5855 | **166.4239** | 167.0154 | 166.7763 | 167.3177 | **166.9298** | **166.9304** | **154.6705** | **154.709** |
| | 30% | 182.8097 | **176.044** | 177.1218 | 176.7111 | 177.7131 | 179.5262 | 179.5269 | **164.9662** | **164.975** |
| | 40% | 177.3301 | **158.4101** | 160.9448 | 159.9731 | 162.3604 | 170.2739 | 170.275 | **139.9159** | **140.0728** |
| 80 | 5% | 463.1782 | 466.5835 | 466.6235 | 466.6069 | 466.643 | **465.8154** | **465.8156** | **461.801** | **461.8195** |
| | 10% | 292.6208 | 293.7304 | 293.8661 | 293.8094 | 293.9324 | **293.028** | 293.0284 | **287.8018** | **287.807** |
| | 15% | 170.6454 | 168.5636 | 168.8625 | 168.7398 | 169.0122 | **168.2237** | 168.2243 | **160.6225** | **160.6296** |
| | 20% | 175.6086 | **168.6279** | 169.5391 | 169.1633 | 169.9959 | **169.4835** | **169.4847** | **148.2162** | **148.3669** |
| | 30% | 165.8679 | **156.9776** | 158.1874 | 157.7253 | 158.8495 | 160.9189 | 160.9199 | **143.9081** | **143.9383** |
| | 40% | 201.0647 | **177.4169** | 180.4935 | 179.3782 | 182.2689 | 192.1717 | 192.1734 | **152.2446** | **152.4981** |
| 100 | 5% | 250.8865 | 250.8178 | 250.8485 | 250.8356 | 250.8633 | **250.2654** | 250.2656 | **247.7555** | **247.7569** |
| | 10% | 159.4151 | 158.2652 | 158.3919 | 158.3391 | 158.454 | **157.6333** | 157.6338 | **152.9668** | **152.968** |
| | 15% | 151.9349 | 150.5516 | 150.8445 | 150.7237 | 150.9907 | **150.2453** | 150.2459 | **142.9845** | **142.9888** |
| | 20% | 270.4243 | **268.3731** | 268.9347 | 268.7133 | 269.228 | **268.9192** | 268.9202 | **257.5925** | **257.613** |
| | 30% | 243.1709 | **232.1772** | 233.6273 | 233.0676 | 234.4113 | 236.8885 | 236.89 | **215.1595** | **215.2528** |
| | 40% | 177.694 | **159.4658** | 161.6839 | 160.8883 | 162.9878 | 170.3973 | 170.3989 | **143.4997** | **143.5939** |
| 120 | 5% | 215.7559 | 209.1628 | 209.298 | 209.2423 | 209.3643 | **206.9038** | 206.9051 | **191.2773** | **191.4333** |
| | 10% | 162.2093 | 161.1332 | 161.2809 | 161.2196 | 161.3534 | **160.4132** | 160.4139 | **154.6231** | **154.6364** |
| | 15% | 229.0953 | 226.5701 | 227.3629 | 227.0551 | 227.7757 | **225.7649** | 225.7671 | **197.4147** | **197.6589** |
| | 20% | 172.8289 | **169.0928** | 169.6121 | 169.404 | 169.88 | **169.6111** | **169.6121** | **159.864** | **159.8679** |
| | 30% | 199.4369 | **190.9045** | 192.0939 | 191.6382 | 192.7429 | 194.8145 | 194.8159 | **178.3088** | **178.3383** |
| | 40% | 175.7893 | **158.7317** | 160.8544 | 160.09 | 162.0986 | 169.1913 | 169.1931 | **144.061** | **144.1354** |
| 200 | 5% | 162.4653 | 162.2933 | 162.3253 | 162.3118 | 162.3407 | **161.7804** | 161.7809 | 159.4553 | **159.4551** |
| | 10% | 184.3058 | 183.6324 | 183.7619 | 183.7082 | 183.8257 | **183.0328** | 183.0338 | 178.4344 | **178.4342** |
| | 15% | **322.1418** | 318.833 | 319.2925 | 319.1073 | 319.5257 | 318.4248 | 318.4269 | **304.6485** | **304.7275** |
| | 20% | 170.9484 | 167.3597 | **167.922** | 167.6961 | 168.2115 | 167.9577 | 167.9595 | **157.0452** | **157.0634** |
| | 30% | 191.3155 | **182.3733** | 183.5299 | 183.0896 | 184.1655 | 186.2291 | 186.2314 | **170.3903** | **170.4096** |
| | 40% | 180.6005 | **164.487** | 166.4627 | 165.7535 | 167.6262 | 174.3219 | 174.3247 | **151.2512** | **151.3016** |
| 500 | 5% | 187.7353 | 187.3017 | 187.34 | 187.3239 | 187.3585 | **186.7269** | 186.7283 | **183.8233** | **183.8293** |
| | 10% | 183.5493 | 182.3278 | 182.4947 | 182.4256 | 182.577 | **181.5998** | 181.6027 | **174.9747** | **174.9977** |
| | 15% | 324.4419 | 321.636 | 321.9534 | 321.8245 | 322.114 | **321.3908** | 321.3943 | **313.4787** | **313.4879** |
| | 20% | 196.1359 | **192.4507** | 193.0014 | 192.782 | 193.2871 | 193.0766 | 193.081 | **182.5253** | **182.5387** |
| | 30% | 206.956 | **198.5238** | 199.7337 | 199.2726 | 200.3978 | 202.5976 | 202.6037 | **185.8272** | **185.8588** |
| | 40% | 201.1646 | **180.8262** | 183.3337 | 182.4421 | 184.8089 | 193.2659 | 193.2747 | **161.7957** | **161.9435** |


### 3.2. Results of a study on actual data

Mean square errors and mean absolute percentage errors of gold loss for the scenarios where error variance was 1, 3, 5, 7, and 9; a sample of 48; and missing value percentages of 5, 10, 15, 20, 30 and 40% with SRS and RSS are shown in Tables 22, 23, 24, and 25.

Table 22 shows that for all error variances and missing value percentages of gold loss, the AR-MRQ3 method achieved the minimum mean square error with SRS.

Table 23 shows that for all error variances and missing value percentages of gold loss, the AR-MCRQ1 method achieved the minimum mean absolute percentage error with SRS.

Table 24 shows that for all error variances and lower missing value percentages of gold loss with RSS, the AR-RQ1 method achieved the minimum mean square error. For all error variances and higher missing value percentages, the AR-MRQ1 and AR-MRQ3 methods achieved the same minimum mean square error.

Table 25 shows that for all error variances and missing value percentages of gold loss with RSS, the AR-MCRQ1 method achieved the minimum mean absolute percentage error. The AR-CRQ1 and AR-MCRQ3 methods achieved the minimum mean absolute percentage error for all error variances and higher missing value percentages.

Mean square errors and mean absolute percentage errors of platinum loss for the scenarios where error variances were 1, 3, 5, 7, and 9; a sample size of 12; and missing value percentages of 5, 10, 15, and 20% with SRS and RSS are shown in Tables 26, 27, 28, and 29.

Table 26 shows that, for all error variances and missing value percentages of platinum loss with SRS, the AR-MRQ3 method achieved the minimum mean square error.

Table 27 shows that, for all error variances and missing value percentages of platinum loss with SRS, the AR-MCRQ1 method achieved the minimum mean absolute percentage error.

Table 28 shows that, for all error variances and missing value percentages of platinum loss of 10, and 15% with RSS, the AR-RQ1 method achieved the minimum mean square error. For all error variances and missing value percentages of platinum loss of 20%, the AR-RQ3 method achieved the minimum mean square error. For all error variances and missing value percentage of platinum loss of 5%, the AR-MCRQ3 method achieved the minimum mean square error. Finally, the AR-MRQ3 method achieved the minimum mean square error for all error variances and higher missing value percentages.

Table 29 shows that, for all error variances and missing value percentages of platinum loss with RSS, the AR-MCRQ1 method achieved the minimum mean absolute percentage error.

Table 30 is a summary of the comparative results between the four methods. The best method, which provided the minimum MSE and MAPE for each kind of error variance and sample size, is tabulated in two columns, SRS and RSS.

AR-MRQ1 with SRS achieved the minimum mean square error for a small error variance. However, AR-MCRQ3 achieved the minimum mean square error for a large error variance. AR-MCRQ1 achieved the minimum mean absolute percentage error for all kinds of error variances.

AR-MRQ1 with RSS achieved the minimum mean square error for a small error variance. However, AR-MCRQ3 and AR-MCRQ1 achieved the minimum mean square error for a larger error variance. AR-MRQ1 achieved the minimum mean absolute percentage error for a small error variance. However, AR-MCRQ1 achieved the minimum mean absolute percentage error for a larger error variance.

Table 31, for a small error variance, the AR-MRQ1 method with SRS and RSS provided the minimum MSE. However, for middle and large error variances, AR-MCRQ3 provided the minimum MSE. Regarding the MAPE measure, for a small error variance, AR-MCRQ1

**Table 20. Mean absolute percentage errors for the scenario where error variance was 7; sample sizes of 20, 40, 60, 80, 100, 120, 200, and 500; and missing value percentages of 5, 10, 15, 20, 30, and 40% with RSS in missing value estimation methods.**

| Sample sizes | Missing values | Listwise delition | Missing value estimation methods | | | | | | | |
|---|---|---|---|---|---|---|---|---|---|---|
| | | | R-RQ1 | R-RQ3 | AR-CRQ1 | AR-CRQ3 | AR-MRQ1 | AR-MRQ3 | AR-MCRQ1 | AR-MCRQ3 |
| 20 | 5% | 228.2015 | 228.1494 | 228.1575 | 228.154 | 228.1614 | **227.2854** | 227.2855 | **223.6198** | **223.636** |
| | 10% | 223.0908 | 223.707 | 223.81 | 223.7677 | 223.8607 | **222.5859** | 222.586 | **215.6303** | **215.6536** |
| | 15% | **234.7358** | 233.1497 | 233.4115 | 233.3033 | 233.541 | **232.3515** | 232.3517 | **223.2659** | **223.2915** |
| | 20% | 192.5133 | **187.9682** | 188.624 | 188.3506 | 188.9485 | **188.0363** | **188.0366** | **171.4458** | **171.5334** |
| | 30% | 205.2531 | **194.8874** | 196.3403 | 195.7612 | 197.1004 | 198.938 | 198.9383 | **175.3498** | **175.4657** |
| | 40% | 240.3467 | **220.2805** | 222.5567 | 221.7191 | 223.8538 | 230.5951 | 230.5954 | **201.6883** | **201.8144** |
| 40 | 5% | 196.2122 | 195.2874 | 195.3122 | 195.3015 | 195.324 | **194.6167** | **194.6168** | **191.6135** | **191.6195** |
| | 10% | 222.0807 | 221.4443 | 221.5652 | 221.5145 | 221.6239 | **220.6377** | 220.6379 | **215.0701** | **215.077** |
| | 15% | 306.2351 | 297.7138 | 298.7151 | 298.3097 | 299.219 | **296.2675** | 296.2684 | **257.6727** | **258.0466** |
| | 20% | 195.8845 | **192.3611** | 192.9172 | 192.692 | 193.2001 | **192.7225** | **192.7228** | **180.4368** | **180.4716** |
| | 30% | 202.577 | **188.2687** | 190.3526 | 189.5568 | 191.4807 | 194.8509 | 194.8517 | **158.7188** | **158.9845** |
| | 40% | 224.9357 | **205.4141** | 207.9515 | 207.0064 | 209.3896 | 217.2406 | 217.2413 | **185.4819** | **185.6424** |
| 60 | 5% | 210.8323 | 209.8095 | 209.8415 | 209.8277 | 209.8566 | **209.1425** | 209.1426 | **205.8679** | **205.8791** |
| | 10% | 228.619 | 229.5535 | 229.7035 | 229.6401 | 229.776 | **228.7226** | 228.723 | **221.9457** | **221.9686** |
| | 15% | 228.9357 | 231.0276 | 231.7774 | 231.4731 | 232.1542 | **230.1854** | 230.1865 | **203.5412** | **203.7727** |
| | 20% | 201.9847 | **196.5299** | 197.2961 | 196.9914 | 197.6915 | **197.1897** | **197.1904** | **178.8487** | **178.9491** |
| | 30% | 239.0615 | **228.0592** | 229.4294 | 228.9078 | 230.1762 | 232.452 | 232.4528 | **211.1345** | **211.2211** |
| | 40% | 220.0078 | **194.3276** | 197.6428 | 196.3703 | 199.4769 | 209.5758 | 209.5771 | **167.037** | **167.3451** |
| 80 | 5% | 523.0761 | 526.462 | 526.5049 | 526.487 | 526.5257 | **525.6366** | **525.6369** | **521.0491** | **521.0743** |
| | 10% | 333.9213 | 330.858 | 331.0087 | 330.9462 | 331.0828 | **330.0621** | 330.0626 | **323.3813** | **323.4026** |
| | 15% | 204.2614 | 202.0052 | 202.3434 | 202.2045 | 202.5123 | **201.6174** | 201.618 | **192.0286** | **192.0597** |
| | 20% | 188.7285 | **184.3244** | 185.074 | 184.7688 | 185.4542 | **185.0166** | **185.0176** | **167.7243** | **167.8206** |
| | 30% | 189.5829 | **179.6569** | 181.045 | 180.5198 | 181.8045 | 184.1634 | 184.1645 | **162.6624** | **162.7469** |
| | 40% | 203.1449 | **180.5601** | 183.4352 | 182.3991 | 185.101 | 194.3648 | 194.3664 | **156.6919** | **156.9185** |
| 100 | 5% | 436.6739 | 433.5618 | 433.5977 | 433.5826 | 433.615 | **432.9114** | 432.9117 | **429.5484** | **429.559** |
| | 10% | 194.7727 | 193.8534 | 194.0193 | 193.9502 | 194.1004 | **193.0255** | 193.0262 | **185.6125** | **185.6444** |
| | 15% | 188.6218 | 186.627 | 187.0203 | 186.857 | 187.215 | **186.2265** | 186.2274 | **174.6678** | **174.7232** |
| | 20% | 365.7963 | **361.0408** | 361.7025 | 361.4396 | 362.0451 | **361.6817** | 361.6828 | **346.8159** | **346.8781** |
| | 30% | 228.324 | **215.5626** | 217.2728 | 216.6235 | 218.2045 | 221.144 | 221.1457 | **193.2026** | **193.3652** |
| | 40% | 202.1106 | **181.044** | 183.7219 | 182.7645 | 185.2817 | 194.0028 | 194.0047 | **159.3862** | **159.5721** |
| 120 | 5% | 234.3666 | 227.9744 | 228.1387 | 228.071 | 228.2192 | **225.2223** | 225.2239 | **205.754** | **205.9539** |
| | 10% | 196.7276 | 195.72 | 195.8797 | 195.8133 | 195.9581 | **194.9351** | 194.9358 | **188.1226** | **188.1463** |
| | 15% | 265.5601 | 257.7207 | 258.5367 | 258.2177 | 258.9592 | **256.8947** | 256.897 | **227.2362** | **227.4953** |
| | 20% | 222.5236 | **218.369** | 219.0118 | 218.7553 | 219.3432 | **219.0127** | **219.014** | **204.906** | **204.961** |
| | 30% | 206.4988 | **195.8267** | 197.3552 | 196.7709 | 198.1859 | 200.8182 | 200.82 | **176.8137** | **176.9302** |
| | 40% | 220.9084 | **198.6209** | 201.4449 | 200.4479 | 203.1005 | 212.3688 | 212.3711 | **175.1843** | **175.3978** |
| 200 | 5% | 202.7456 | 202.5678 | 202.6066 | 202.5903 | 202.6254 | **201.9404** | 201.941 | **198.5833** | **198.5945** |
| | 10% | 233.0575 | 231.9901 | 232.1433 | 232.0799 | 232.2188 | **231.2765** | 231.2776 | **224.9141** | **224.9333** |
| | 15% | **345.1219** | 342.4259 | 343.015 | 342.7776 | 343.3135 | 341.9037 | 341.9064 | **322.4493** | **322.5944** |
| | 20% | 221.5589 | 216.8441 | **217.5607** | 217.273 | 217.9284 | 217.6054 | 217.6077 | **201.6991** | **201.7768** |
| | 30% | 444.2255 | **433.2253** | 434.6992 | 434.1408 | 435.5053 | 438.1062 | 438.1091 | **415.1635** | **415.2635** |
| | 40% | 202.0543 | **183.0794** | 185.4587 | 184.6067 | 186.848 | 194.7104 | 194.7137 | **164.8232** | **164.9529** |
| 500 | 5% | 225.5124 | 224.9936 | 225.0379 | 225.0193 | 225.0593 | **224.3253** | 224.3269 | **220.5358** | **220.5518** |
| | 10% | 206.9888 | 205.5938 | 205.7859 | 205.7065 | 205.8806 | **204.7512** | 204.7545 | **196.3273** | **196.37** |
| | 15% | 575.1941 | 572.5294 | 572.9279 | 572.7663 | 573.1291 | **572.221** | 572.2255 | **560.6603** | **560.7124** |
| | 20% | 219.0306 | **213.8623** | 214.6288 | 214.3254 | 215.0264 | 214.7357 | 214.7418 | **197.1743** | **197.2675** |
| | 30% | 246.7447 | **236.4599** | 237.9434 | 237.3793 | 238.7536 | 241.4184 | 241.4257 | **218.4774** | **218.5783** |
| | 40% | 282.9756 | **255.2549** | 258.767 | 257.5235 | 260.8159 | 272.3538 | 272.3656 | **224.6084** | **224.9469** |


**Table 21. Mean absolute percentage errors for the scenario where error variance was 9; sample sizes of 20, 40, 60, 80, 100, 120, 200, and 500; and missing value percentages of 5, 10, 15, 20, 30, and 40% with RSS in missing value estimation methods.**

| Sample sizes | Missing values | Listwise delition | Missing value estimation methods | | | | | | | |
|---|---|---|---|---|---|---|---|---|---|---|
| | | | R-RQ1 | R-RQ3 | AR-CRQ1 | AR-CRQ3 | AR-MRQ1 | AR-MRQ3 | AR-MCRQ1 | AR-MCRQ3 |
| 20 | 5% | 308.8765 | 308.197 | 308.211 | 308.2049 | 308.2176 | **307.0716** | 307.0717 | **301.5634** | **301.6033** |
| | 10% | 282.5199 | 283.1238 | 283.2537 | 283.199 | 283.3165 | **281.7664** | 281.7666 | **272.3811** | **272.4397** |
| | 15% | **263.0108** | 260.6264 | 260.9332 | 260.8052 | 261.0838 | **259.7081** | 259.7083 | **248.0169** | **248.0747** |
| | 20% | 232.7102 | **228.1869** | 228.9492 | 228.6252 | 229.3197 | **228.2706** | **228.2709** | **207.8136** | **207.9582** |
| | 30% | 232.3853 | **221.2005** | 222.8405 | 222.1925 | 223.7028 | 225.7808 | 225.7812 | **197.391** | **197.5702** |
| | 40% | 406.5708 | **387.9422** | 390.7279 | 389.7766 | 392.3832 | 400.9869 | 400.9873 | **361.3455** | **361.5868** |
| 40 | 5% | 379.8662 | 380.452 | 380.4787 | 380.4672 | 380.4913 | **379.7216** | **379.7217** | **376.0797** | **376.0945** |
| | 10% | 248.4955 | 247.1534 | 247.307 | 247.2429 | 247.3819 | **246.1384** | 246.1387 | **237.9563** | **237.9977** |
| | 15% | 265.2678 | 257.0216 | 257.8201 | 257.4933 | 258.2182 | **255.8598** | 255.8605 | **225.72** | **225.9935** |
| | 20% | 234.4496 | **229.6977** | 230.4186 | 230.1257 | 230.7837 | **230.1827** | **230.1832** | **212.4326** | **212.5365** |
| | 30% | 267.904 | **245.9249** | 249.1849 | 247.8945 | 250.8937 | 256.0055 | 256.0068 | **197.1349** | **197.6947** |
| | 40% | 256.4665 | **230.9655** | 234.3916 | 233.1152 | 236.3205 | 246.7354 | 246.7363 | **200.6448** | **200.9831** |
| 60 | 5% | 274.0201 | 274.4118 | 274.4555 | 274.4367 | 274.4761 | **273.5131** | 273.5133 | **268.3973** | **268.4315** |
| | 10% | 297.0196 | 293.776 | 293.9538 | 293.8793 | 294.0403 | **292.779** | 292.7794 | **283.9151** | **283.9634** |
| | 15% | 246.4003 | 236.9198 | 237.5005 | 237.265 | 237.7926 | **236.2123** | 236.2132 | **216.0172** | **216.1702** |
| | 20% | 230.9762 | **225.1471** | 225.9894 | 225.6529 | 226.4216 | **225.868** | **225.8689** | **204.7622** | **204.9003** |
| | 30% | 231.9347 | **222.197** | 223.707 | 223.1347 | 224.53 | 227.0241 | 227.025 | **202.3304** | **202.4565** |
| | 40% | 251.4197 | **221.6351** | 225.5718 | 224.0549 | 227.7331 | 239.533 | 239.5346 | **187.396** | **187.8294** |
| 80 | 5% | 967.226 | 963.3419 | 963.3946 | 963.3726 | 963.4201 | **962.3332** | **962.3335** | **956.3012** | **956.3435** |
| | 10% | 294.4182 | 291.3296 | 291.501 | 291.4297 | 291.5848 | **290.4254** | 290.426 | **282.2979** | **282.3367** |
| | 15% | 219.777 | 217.1951 | 217.5889 | 217.4274 | 217.7856 | **216.741** | 216.7418 | **204.398** | **204.4622** |
| | 20% | 235.5731 | **228.5523** | 229.7202 | 229.2459 | 230.3115 | **229.654** | **229.6555** | **199.6527** | **199.9009** |
| | 30% | 214.2518 | **202.8536** | 204.4767 | 203.8616 | 205.3603 | 208.096 | 208.0973 | **181.2805** | **181.4309** |
| | 40% | 234.6107 | **211.7121** | 214.5894 | 213.5512 | 216.2473 | 225.4171 | 225.4187 | **187.4886** | **187.7172** |
| 100 | 5% | 266.2074 | 263.5141 | 263.5554 | 263.538 | 263.5753 | **262.7665** | 262.7669 | **258.5331** | **258.5553** |
| | 10% | 222.9288 | 222.0292 | 222.2086 | 222.1338 | 222.2964 | **221.1271** | 221.1278 | **212.6534** | **212.6984** |
| | 15% | 209.4874 | 207.3642 | 207.7896 | 207.6133 | 208.0004 | **206.9251** | 206.926 | **193.7235** | **193.7994** |
| | 20% | 247.7401 | **243.7068** | 244.4218 | 244.1387 | 244.7918 | **244.3955** | 244.3967 | **227.4555** | **227.5433** |
| | 30% | 337.7131 | **325.6391** | 327.4766 | 326.7791 | 328.4748 | 331.607 | 331.6088 | **300.6692** | **300.8706** |
| | 40% | 247.8758 | **219.6567** | 223.2796 | 221.9976 | 225.3865 | 237.0336 | 237.0361 | **186.9572** | **187.327** |
| 120 | 5% | 298.5901 | 302.1416 | 302.4685 | 302.3342 | 302.6291 | **296.6876** | 296.6908 | **256.314** | **256.7603** |
| | 10% | 218.5231 | 217.5994 | 217.7944 | 217.7135 | 217.8902 | **216.64** | 216.6409 | **207.2759** | **207.3301** |
| | 15% | 262.6312 | 255.2925 | 256.0624 | 255.7573 | 256.457 | **254.5206** | 254.5227 | **226.7402** | **226.9829** |
| | 20% | 221.1238 | **216.0352** | 216.7419 | 216.4616 | 217.1074 | **216.7425** | **216.7439** | **200.2304** | **200.3137** |
| | 30% | 228.8371 | **214.5518** | 216.7147 | 215.8572 | 217.8526 | 221.4415 | 221.444 | **185.3284** | **185.6018** |
| | 40% | 235.3645 | **209.4075** | 212.7492 | 211.5557 | 214.6846 | 225.4279 | 225.4306 | **180.0424** | **180.3591** |
| 200 | 5% | 231.7475 | 231.4468 | 231.4882 | 231.4707 | 231.508 | **230.7783** | 230.779 | **227.0022** | **227.0184** |
| | 10% | 293.9669 | 292.9684 | 293.1504 | 293.075 | 293.2399 | **292.1197** | 292.121 | **283.8424** | **283.8848** |
| | 15% | **339.4221** | 334.7767 | 335.5938 | 335.2666 | 336.009 | 334.0473 | 334.0511 | **305.0469** | **305.3043** |
| | 20% | 251.2568 | 246.4437 | **247.2226** | 246.9104 | 247.6223 | 247.2704 | 247.2729 | **228.9295** | **229.0375** |
| | 30% | 280.9567 | **268.8168** | 270.4407 | 269.8256 | 271.3272 | 274.1788 | 274.182 | **247.7143** | **247.861** |
| | 40% | 223.1637 | **200.3712** | 203.2759 | 202.241 | 204.9658 | 214.4283 | 214.4322 | **175.8** | **176.0366** |
| 500 | 5% | 262.8758 | 262.4849 | 262.5314 | 262.5119 | 262.5539 | **261.7805** | 261.7823 | **257.6132** | **257.6336** |
| | 10% | 239.9691 | 238.7844 | 239.0186 | 238.9218 | 239.1339 | **237.7572** | 237.7613 | **226.5883** | **226.6647** |
| | 15% | 381.6172 | 378.5317 | 379.005 | 378.8127 | 379.2434 | **378.1661** | 378.1714 | **363.3493** | **363.4415** |
| | 20% | 263.5808 | **258.5161** | 259.3197 | 259.0003 | 259.7348 | 259.4297 | 259.4361 | **240.3806** | **240.4947** |
| | 30% | 263.3312 | **251.0471** | 252.7271 | 252.0891 | 253.6423 | 256.6418 | 256.65 | **229.1672** | **229.327** |
| | 40% | 334.5533 | **297.1716** | 302.0105 | 300.2928 | 304.8091 | 320.4186 | 320.4345 | **251.6578** | **252.2517** |


**Table 22. Mean square errors of gold loss for error variances of 1, 3, 5, 7, and 9; a sample sizes of 48; and missing value percentages of 5, 10, 15, 20, 30, and 40% with SRS in missing value estimation methods.**

| Sample sizes | Missing values | Listwise delition | Missing value estimation methods | | | | | | | |
|---|---|---|---|---|---|---|---|---|---|---|
| | | | R-RQ1 | R-RQ3 | AR-CRQ1 | AR-CRQ3 | AR-MRQ1 | AR-MRQ3 | AR-MCRQ1 | AR-MCRQ3 |
| 1 | 5% | 110539.1 | 111617.2 | 111578.1 | 111609.6 | 111574.6 | 111602 | **111571.1** | 116864.3 | 115628.8 |
| | 10% | 110213.2 | 111703.1 | 111554.1 | 111672.5 | 111541.4 | 111640.9 | **111528.5** | 121832.1 | 119349.5 |
| | 15% | 110230 | 111887.5 | 111531.1 | 111808.7 | 111503.4 | 111726.5 | **111475.4** | 128070.6 | 123903.1 |
| | 20% | 110120.6 | 112524.2 | 111776.4 | 112354.5 | 111718.9 | 112172.2 | **111660.7** | 132961.2 | 127571.5 |
| | 30% | 109975.4 | 114727.7 | 112404.6 | 114183.3 | 112222.4 | 113552.9 | **112033.4** | 143917.3 | 135609 |
| | 40% | 110159 | 119747 | 114128.1 | 118429.6 | 113634.4 | 116734.2 | **113092.9** | 155642.5 | 144289.8 |
| 3 | 5% | 110535.6 | 111613.7 | 111574.6 | 111606.1 | 111571.1 | 111598.4 | **111567.5** | 116860.7 | 115625.2 |
| | 10% | 110226.5 | 111717.9 | 111568.8 | 111687.3 | 111556.1 | 111655.7 | **111543.2** | 121847.6 | 119365 |
| | 15% | 110229 | 111886.5 | 111530.1 | 111807.7 | 111502.3 | 111725.5 | **111474.3** | 128068.7 | 123901.6 |
| | 20% | 110121.2 | 112523.9 | 111776.2 | 112354.2 | 111718.7 | 112171.9 | **111660.5** | 132960.2 | 127570.5 |
| | 30% | 109964.6 | 114716.1 | 112393.1 | 114171.7 | 112210.9 | 113541.4 | **112022** | 143906.2 | 135597.6 |
| | 40% | 110163.3 | 119749.3 | 114131.1 | 118432 | 113637.4 | 116736.8 | **113096** | 155644.8 | 144291.9 |
| 5 | 5% | 110531.4 | 111608.8 | 111569.7 | 111601.2 | 111566.2 | 111593.6 | **111562.7** | 116856.1 | 115620.5 |
| | 10% | 110223.1 | 111712 | 111562.9 | 111681.4 | 111550.2 | 111649.8 | **111537.3** | 121842 | 119359.6 |
| | 15% | 110237.3 | 111895.8 | 111539.6 | 111817.1 | 111511.8 | 111734.9 | **111483.8** | 128076.4 | 123909.4 |
| | 20% | 110116.3 | 112520.3 | 111772.5 | 112350.6 | 111715 | 112168.2 | **111656.8** | 132957.8 | 127568 |
| | 30% | 109981.3 | 114731 | 112409.3 | 114186.9 | 112227.3 | 113556.9 | **112038.6** | 143917.6 | 135609.3 |
| | 40% | 110177.5 | 119762.1 | 114144.4 | 118444.9 | 113650.8 | 116749.8 | **113109.4** | 155654.4 | 144302.5 |
| 7 | 5% | 110545.1 | 111622.8 | 111583.7 | 111615.2 | 111580.2 | 111607.5 | **111576.7** | 116870 | 115634.3 |
| | 10% | 110235.6 | 111724 | 111575 | 111693.4 | 111562.3 | 111661.8 | **111549.4** | 121853.3 | 119370.4 |
| | 15% | 110224.8 | 111883 | 111526.4 | 111804 | 111498.6 | 111721.9 | **111470.6** | 128067.6 | 123900.2 |
| | 20% | 110129.1 | 112531.5 | 111783.6 | 112361.8 | 111726.1 | 112179.4 | **111667.9** | 132968.9 | 127579.2 |
| | 30% | 109972.3 | 114720.6 | 112397.6 | 114176.3 | 112215.4 | 113545.9 | **112026.4** | 143911.3 | 135602.6 |
| | 40% | 110157.6 | 119747.2 | 114128.7 | 118429.8 | 113635.1 | 116734.6 | **113093.6** | 155640.4 | 144288.4 |
| 9 | 5% | 110559.8 | 111637.8 | 111598.7 | 111630.2 | 111595.2 | 111622.6 | **111591.7** | 116883.8 | 115648.4 |
| | 10% | 110229.2 | 111719.6 | 111570.7 | 111689 | 111558 | 111657.4 | **111545.1** | 121847.3 | 119364.9 |
| | 15% | 110231.8 | 111887.7 | 111531.8 | 111809 | 111504 | 111726.9 | **111476.1** | 128069.2 | 123901.6 |
| | 20% | 110116.2 | 112517.6 | 111769.9 | 112347.9 | 111712.5 | 112165.6 | **111654.3** | 132955.1 | 127565 |
| | 30% | 109972.1 | 114726.9 | 112403.4 | 114182.5 | 112221.1 | 113552 | **112032.1** | 143915.3 | 135607.7 |
| | 40% | 110160.5 | 119746.8 | 114129.1 | 118429.6 | 113635.5 | 116734.5 | **113094.2** | 155641.8 | 144288.8 |


and AR-MRQ1 with SRS and RSS provided the minimum MAPE, respectively. However, AR-MCRQ1 with SRS and RSS provided the minimum MAPE for middle and large error variances. Finally, AR-MCRQ1 was the best method for missing value imputation in multiple regression analysis, followed by AR-MCRQ3.

Table 32, for all error variances, the sample sizes were increased, and the number of times for methods was increased with the RSS estimators providing smaller MSE than the SRS estimators. Moreover, the RSS estimators provided smaller MSE and MAPE than the SRS estimators for all error variances and sample sizes.

## 4. Discussion

This research compared the efficiency of four new adjusted missing value imputations in multiple regression analysis. The study results show that AR-MRQ1 with SRS achieved the minimum mean square error for a small error variance. However, AR-MCRQ3 achieved the minimum mean square error for a large error variance. For all error variances in mean absolute percentage error, AR-MCRQ1 achieved the minimum mean absolute percentage error. The results of this study were similar to those reported in [14,15]. In those papers, the proposed

**Table 23. Mean absolute percentage errors of gold loss for error variances of 1, 3, 5, 7, and 9; a sample sizes of 48; and missing value percentages of 5, 10, 15, 20, 30 and 40% with SRS in missing value estimation methods.**

| Sample sizes | Missing values | Listwise delition | Missing value estimation methods | | | | | | | |
|---|---|---|---|---|---|---|---|---|---|---|
| | | | R-RQ1 | R-RQ3 | AR-CRQ1 | AR-CRQ3 | AR-MRQ1 | AR-MRQ3 | AR-MCRQ1 | AR-MCRQ3 |
| 1 | 5% | 285.1246 | 285.1777 | 285.2769 | 285.1965 | 285.2861 | 285.216 | **285.2954** | **281.706** | 281.7864 |
| | 10% | 285.3053 | 283.9008 | 284.5208 | 284.0185 | 284.5801 | 284.1446 | **284.6415** | **274.7975** | 275.5009 |
| | 15% | 285.7429 | 280.8366 | 282.7458 | 281.1991 | 282.9354 | 281.6043 | **283.1359** | **263.2475** | 265.1126 |
| | 20% | 285.516 | 277.7929 | 281.2114 | 278.4422 | 281.5653 | 279.2013 | **281.9482** | **255.5784** | 258.3216 |
| | 30% | 285.1613 | 266.1675 | 274.5077 | 267.75 | 275.4565 | 269.7966 | **276.5385** | **234.0563** | 239.1611 |
| | 40% | 285.6655 | 248.2271 | 263.2748 | 251.0756 | 265.1752 | 255.2599 | **267.4938** | **210.3753** | 218.1984 |
| 3 | 5% | 285.1099 | 285.1622 | 285.2614 | 285.1811 | 285.2706 | 285.2006 | **285.2799** | **281.692** | 281.7702 |
| | 10% | 285.3448 | 283.9411 | 284.5604 | 284.0586 | 284.6196 | 284.1846 | **284.6809** | **274.8503** | 275.55 |
| | 15% | 285.7356 | 280.8257 | 282.735 | 281.188 | 282.9247 | 281.5933 | **283.1253** | **263.2339** | 265.0998 |
| | 20% | 285.5071 | 277.7861 | 281.2037 | 278.4353 | 281.5574 | 279.1942 | **281.9404** | **255.5789** | 258.3219 |
| | 30% | 285.154 | 266.1587 | 274.4994 | 267.7414 | 275.4483 | 269.7878 | **276.5304** | **234.0478** | 239.1522 |
| | 40% | 285.6614 | 248.2252 | 263.2711 | 251.0734 | 265.1712 | 255.2572 | **267.4901** | **210.3772** | 218.1984 |
| 5 | 5% | 285.0905 | 285.1424 | 285.2415 | 285.1612 | 285.2506 | 285.1807 | **285.26** | **281.6761** | 281.7555 |
| | 10% | 285.3196 | 283.913 | 284.5331 | 284.0306 | 284.5924 | 284.1569 | **284.6538** | **274.8077** | 275.5096 |
| | 15% | 285.7736 | 280.8668 | 282.7767 | 281.2295 | 282.9664 | 281.6348 | **283.1669** | **263.2731** | 265.1403 |
| | 20% | 285.5079 | 277.7852 | 281.2038 | 278.4346 | 281.5577 | 279.1935 | **281.9405** | **255.5704** | 258.3132 |
| | 30% | 285.2048 | 266.2111 | 274.5524 | 267.7935 | 275.5013 | 269.8399 | **276.5835** | **234.0905** | 239.198 |
| | 40% | 285.7268 | 248.2759 | 263.3302 | 251.1268 | 265.2314 | 255.3128 | **267.5509** | **210.4076** | 218.235 |
| 7 | 5% | 285.0567 | 285.107 | 285.2061 | 285.1258 | 285.2153 | 285.1453 | **285.2246** | **281.6403** | 281.7193 |
| | 10% | 285.3658 | 283.9562 | 284.5771 | 284.074 | 284.6364 | 284.2003 | **284.6979** | **274.8411** | 275.5445 |
| | 15% | 285.739 | 280.8343 | 282.7435 | 281.1968 | 282.9332 | 281.6021 | **283.1337** | **263.2471** | 265.1097 |
| | 20% | 285.5208 | 277.8032 | 281.2194 | 278.4524 | 281.573 | 279.2111 | **281.9556** | **255.6106** | 258.3523 |
| | 30% | 285.2083 | 266.2109 | 274.5525 | 267.7932 | 275.5015 | 269.8402 | **276.5836** | **234.0954** | 239.2002 |
| | 40% | 285.7007 | 248.2489 | 263.3029 | 251.0994 | 265.2036 | 255.2851 | **267.5225** | **210.3813** | 218.2099 |
| 9 | 5% | 285.1167 | 285.1667 | 285.2657 | 285.1855 | 285.2749 | 285.2049 | **285.2842** | **281.6925** | 281.7723 |
| | 10% | 285.3274 | 283.9242 | 284.544 | 284.0418 | 284.6033 | 284.1679 | **284.6647** | **274.8204** | 275.5217 |
| | 15% | 285.7304 | 280.8239 | 282.7339 | 281.1863 | 282.9236 | 281.5917 | **283.1241** | **263.232** | 265.0998 |
| | 20% | 285.4967 | 277.7729 | 281.1929 | 278.4226 | 281.5469 | 279.1819 | **281.93** | **255.5557** | 258.2999 |
| | 30% | 285.1467 | 266.1515 | 274.4927 | 267.7347 | 275.4417 | 269.7814 | **276.5239** | **234.0298** | 239.1381 |
| | 40% | 285.7007 | 248.2634 | 263.3119 | 251.1127 | 265.2123 | 255.2982 | **267.5305** | **210.4157** | 218.24 |


estimator was a multivariate chain ratio (MCR) and a regression estimator that used a linear combination of two auxiliary variables. The MCR and regression estimators that used the two auxiliary variables were equally highly effective, followed by the traditional multivariate ratio (MR) and regression estimators that used the two auxiliary variables. This study was also consistent with [11] in that the chain ratio (CR) estimator with SRS was more efficient than the traditional ratio estimator under certain conditions. Our results may be contrasted to [22] which reported that the regression-ratio Q1 (R-RQ1) estimator was the most efficient. This discrepancy was because the study did not include AR-CRQ1,3, AR-MRQ1,3, and AR-MCRQ1,3 estimators in the test.

AR-MRQ1 achieved the minimum mean square error for RSS and small error variance. However, AR-MCRQ3 achieved the minimum mean square error for a large error variance, followed by AR-MCRQ1. AR-MRQ1 achieved the minimum mean absolute percentage error for a small error variance. However, AR-MCRQ1 achieved the minimum mean absolute percentage error for a large error variance. To the best of the author's knowledge, RSS has not been studied before.

Moreover, the RSS estimators provided smaller mean square error and mean absolute percentage error than the SRS estimators for the same error variances, sample sizes, and missing value percentages. Therefore, the RSS estimators were more efficient than the SRS estimators.

**Table 24. Mean square error of gold loss for error variances of 1, 3, 5, 7, 9; a sample sizes of 48; and missing value percentages of 5, 10, 15, 20, 30, and 40% with RSS in missing value estimation methods.**

| Sample sizes | Missing values | Listwise delition | Missing value estimation methods | | | | | | | |
|---|---|---|---|---|---|---|---|---|---|---|
| | | | R-RQ1 | R-RQ3 | AR- CRQ1 | AR-CRQ3 | AR-MRQ1 | AR-MRQ3 | AR-MCRQ1 | AR-MCRQ3 |
| 1 | 5% | 8132.28 | **6509.063** | 6547.956 | 6516.225 | 6551.708 | 6523.778 | **6555.552** | **10515.27** | 8676.545 |
| | 10% | 11209.49 | **9193.611** | 9355.857 | 9220.557 | 9373.608 | 9250.729 | **9392.239** | **21081.36** | 17194.74 |
| | 15% | 13114.38 | 11824.79 | 11812.56 | 11804.77 | 11823.21 | **11791.35** | **11836.47** | **34381.03** | 27846.01 |
| | 20% | 13818.25 | 13271.89 | 12795.94 | 13137.57 | 12778.42 | 13006.91 | **12765.64** | **42837.51** | 34457.66 |
| | 30% | 14427.01 | 18398.93 | 15338.08 | 17662.06 | 15116.68 | 16828.53 | **14894.06** | **62723.06** | 49943.09 |
| | 40% | 14822.68 | 26974.59 | 18967.22 | 25087.09 | 18281.32 | 22688.67 | **17538.7** | **80708.14** | 63702.5 |
| 3 | 5% | 8132.056 | **6509.366** | 6548.306 | 6516.538 | 6552.063 | 6524.099 | **6555.912** | **10514.69** | 8675.995 |
| | 10% | 11210.62 | **9196.7** | 9358.917 | 9223.64 | 9376.665 | 9253.807 | **9395.293** | **21084.63** | 17198.19 |
| | 15% | 13116.64 | 11824.58 | 11812.6 | 11804.6 | 11823.28 | **11791.23** | **11836.57** | **34380.82** | 27845.39 |
| | 20% | 13818.3 | 13274.73 | 12798.71 | 13140.41 | 12781.18 | 13009.73 | **12768.38** | **42839.67** | 34460.12 |
| | 30% | 14428.71 | 18400.03 | 15339.22 | 17663.17 | 15117.82 | 16829.65 | **14895.21** | **62723.7** | 49943.86 |
| | 40% | 14824.26 | 26974.46 | 18967.38 | 25087.01 | 18281.53 | 22688.66 | **17538.97** | **80708.25** | 63702.38 |
| 5 | 5% | 8133.49 | **6511.043** | 6549.97 | 6518.212 | 6553.725 | 6525.771 | **6557.573** | **10518.57** | 8679.683 |
| | 10% | 11218 | **9199.959** | 9362.35 | 9226.931 | 9380.115 | 9257.133 | **9398.761** | **21086.45** | 17199.65 |
| | 15% | 13118.3 | 11827.29 | 11815.36 | 11807.33 | 11826.05 | **11793.97** | **11839.34** | **34382.94** | 27847.65 |
| | 20% | 13823.85 | 13277.13 | 12801.48 | 13142.87 | 12784 | 13012.28 | **12771.25** | **42840.99** | 34461.35 |
| | 30% | 14425.81 | 18398.11 | 15337.19 | 17661.23 | 15115.78 | 16827.68 | **14893.16** | **62722.83** | 49942.7 |
| | 40% | 14826.93 | 26976.23 | 18969.36 | 25088.82 | 18283.53 | 22690.52 | **17541** | **80709.47** | 63703.72 |
| 7 | 5% | 8135.977 | **6513.756** | 6552.663 | 6520.921 | 6556.417 | 6528.477 | **6560.263** | **10520.57** | 8681.81 |
| | 10% | 11217.2 | **9204.012** | 9366.461 | 9230.996 | 9384.232 | 9261.209 | **9402.883** | **21089.44** | 17202.98 |
| | 15% | 13120.25 | 11830.71 | 11818.57 | 11810.71 | 11829.23 | **11797.31** | **11842.5** | **34386.7** | 27851.76 |
| | 20% | 13826.98 | 13280.2 | 12803.98 | 13145.83 | 12786.43 | 13015.12 | **12773.61** | **42845.55** | 34465.99 |
| | 30% | 14432.89 | 18402.81 | 15342.5 | 17666.04 | 15121.16 | 16832.63 | **14898.62** | **62725.77** | 49945.84 |
| | 40% | 14820.91 | 26973.43 | 18966.22 | 25085.96 | 18280.36 | 22687.58 | **17537.77** | **80706.98** | 63701.26 |
| 9 | 5% | 8140.484 | **6514.446** | 6553.318 | 6521.604 | 6557.069 | 6529.153 | **6560.911** | **10521.31** | 8682.343 |
| | 10% | 11214.84 | **9203.122** | 9365.429 | 9230.08 | 9383.186 | 9260.264 | **9401.822** | **21088.46** | 17202.15 |
| | 15% | 13121.63 | 11831.1 | 11819.09 | 11811.12 | 11829.77 | **11797.74** | **11843.05** | **34385.98** | 27851.02 |
| | 20% | 13828.86 | 13280.39 | 12805.35 | 13146.24 | 12787.94 | 13015.77 | **12775.26** | **42842.92** | 34462.94 |
| | 30% | 14438.27 | 18407.43 | 15346.9 | 17670.62 | 15125.53 | 16837.16 | **14902.96** | **62730.56** | 49950.7 |
| | 40% | 14824.25 | 26976.96 | 18969.71 | 25089.51 | 18283.81 | 22691.13 | **17541.18** | **80708.5** | 63703.58 |


**Table 25. Mean absolute percentage error of gold loss for error variances of 1, 3, 5, 7, 9; a sample sizes of 48; and missing value percentages of 5, 10, 15, 20, 30 and 40% with RSS in missing value estimation methods.**

| Sample sizes | Missing values | Listwise delition | Missing value estimation methods | | | | | | | |
|---|---|---|---|---|---|---|---|---|---|---|
| | | | R-RQ1 | R-RQ3 | AR- CRQ1 | AR-CRQ3 | AR-MRQ1 | AR-MRQ3 | AR-MCRQ1 | AR-MCRQ3 |
| 1 | 5% | 34.4174 | **36.6773** | 36.9467 | 36.7283 | 36.9717 | 36.7812 | **36.9972** | **27.0932** | 28.2219 |
| | 10% | 43.2773 | **46.0735** | 46.8215 | 46.2151 | 46.8934 | 46.3676 | **46.9678** | **35.9682** | 37.1316 |
| | 15% | 46.8906 | 52.5821 | 53.9486 | 52.8364 | 54.0864 | **53.1254** | **54.2324** | **43.7127** | 44.6636 |
| | 20% | 48.3325 | 52.6713 | 54.4667 | 52.9971 | 54.6650 | 53.3872 | **54.8810** | **46.6301** | 47.1979 |
| | 30% | 49.2486 | 54.8575 | 56.8205 | 55.1613 | 57.0924 | 55.6140 | **57.4147** | **54.4431** | **54.1486** |
| | 40% | 49.9838 | 55.2443 | 55.8013 | **55.1992** | 56.0337 | 55.2763 | **56.3694** | **59.9973** | 58.5468 |
| 3 | 5% | 34.4396 | **36.7275** | 36.9969 | 36.7785 | 37.0219 | 36.8315 | **37.0474** | **27.1402** | 28.2693 |
| | 10% | 43.3262 | **46.1239** | 46.8722 | 46.2656 | 46.9440 | 46.4181 | **47.0185** | **36.0163** | 37.1802 |
| | 15% | 46.9147 | 52.6462 | 54.0133 | 52.9007 | 54.1511 | **53.1898** | **54.2972** | **43.7745** | 44.7256 |
| | 20% | 48.3648 | 52.7304 | 54.5269 | 53.0563 | 54.7252 | 53.4469 | **54.9411** | **46.6839** | 47.2534 |
| | 30% | 49.2656 | 54.9136 | 56.8760 | 55.2171 | 57.1482 | 55.6694 | **57.4709** | **54.5012** | **54.2067** |
| | 40% | 49.9831 | 55.2646 | 55.8177 | **55.2180** | 56.0491 | 55.2936 | **56.3830** | **60.0345** | 58.5788 |
| 5 | 5% | 34.4258 | **36.7361** | 37.0056 | 36.7872 | 37.0306 | 36.8401 | **37.0561** | **27.1479** | 28.2776 |
| | 10% | 43.3273 | **46.1545** | 46.9026 | 46.2962 | 46.9745 | 46.4487 | **47.0490** | **36.0412** | 37.2056 |
| | 15% | 46.8975 | 52.6742 | 54.0412 | 52.9289 | 54.1790 | **53.2179** | **54.3249** | **43.8047** | 44.7545 |
| | 20% | 48.3321 | 52.7786 | 54.5749 | 53.1045 | 54.7731 | 53.4947 | **54.9891** | **46.7334** | 47.3020 |
| | 30% | 49.2513 | 54.9284 | 56.8892 | 55.2315 | 57.1613 | 55.6841 | **57.4829** | **54.5209** | **54.2250** |
| | 40% | 50.0131 | 55.3143 | 55.8697 | **55.2687** | 56.1007 | 55.3460 | **56.4358** | **60.0728** | 58.6211 |
| 7 | 5% | 34.4869 | **36.8254** | 37.0948 | 36.8764 | 37.1198 | 36.9294 | **37.1452** | **27.2414** | 28.3703 |
| | 10% | 43.3570 | **46.2198** | 46.9684 | 46.3615 | 47.0403 | 46.5141 | **47.1148** | **36.1018** | 37.2668 |
| | 15% | 46.9252 | 52.7117 | 54.0794 | 52.9665 | 54.2175 | **53.2554** | **54.3636** | **43.8272** | 44.7811 |
| | 20% | 48.3800 | 52.8015 | 54.5950 | 53.1271 | 54.7929 | 53.5170 | **55.0085** | **46.7813** | 47.3469 |
| | 30% | 49.2409 | 54.9379 | 56.8992 | 55.2412 | 57.1711 | 55.6933 | **57.4936** | **54.5257** | **54.2308** |
| | 40% | 49.9735 | 55.3449 | 55.9017 | **55.2993** | 56.1331 | 55.3766 | **56.4695** | **60.1048** | 58.6520 |
| 9 | 5% | 34.4467 | **36.8363** | 37.1057 | 36.8873 | 37.1307 | 36.9402 | **37.1562** | **27.2566** | 28.3848 |
| | 10% | 43.3663 | **46.2847** | 47.0328 | 46.4264 | 47.1046 | 46.5788 | **47.1790** | **36.1767** | 37.3405 |
| | 15% | 46.9642 | 52.7755 | 54.1409 | 53.0297 | 54.2788 | **53.3184** | **54.4249** | **43.9039** | 44.8543 |
| | 20% | 48.3892 | 52.8512 | 54.6454 | 53.1767 | 54.8437 | 53.5665 | **55.0596** | **46.8185** | 47.3855 |
| | 30% | 49.3259 | 54.9726 | 56.9331 | 55.2755 | 57.2049 | 55.7263 | **57.5270** | **54.5818** | **54.2828** |
| | 40% | 50.0526 | 55.3765 | 55.9356 | **55.3315** | 56.1676 | 55.4091 | **56.5038** | **60.1177** | 58.6709 |


**Table 26. Mean square errors of platinum loss for the scenarios with error variance of 1, 3, 5, 7, and 9; a sample sizes of 48; and missing value percentages of 5, 10, 15, 20, 30, and 40% with SRS in missing value estimation methods.**

| Sample sizes | Missing values | Listwise delition | Missing value estimation methods | | | | | | | |
|---|---|---|---|---|---|---|---|---|---|---|
| | | | R-RQ1 | R-RQ3 | AR- CRQ1 | AR-CRQ3 | AR-MRQ1 | AR-MRQ3 | AR-MCRQ1 | AR-MCRQ3 |
| 1 | 5% | 7670.53 | **7569.60** | 7562.75 | 7568.27 | 7562.46 | 7566.97 | **7562.19** | **9054.11** | 8396.74 |
| | 10% | 7669.48 | **7578.04** | 7538.82 | 7569.90 | 7537.40 | 7561.80 | **7536.05** | **10455.59** | 9161.83 |
| | 15% | 7669.62 | 7681.90 | 7540.65 | 7652.41 | 7535.47 | **7621.87** | **7530.51** | **12501.24** | 10294.60 |
| | 20% | 7667.36 | 7867.43 | 7559.70 | 7803.34 | 7547.94 | 7733.80 | **7536.54** | **13833.33** | 11025.02 |
| | 30% | 7666.01 | 8759.05 | 7711.70 | 8546.41 | 7664.95 | 8289.20 | **7617.40** | **17263.53** | **12926.55** |
| | 40% | 7670.16 | 10612.20 | 8157.01 | **10125.40** | 8027.39 | 9459.66 | **7887.72** | **20870.95** | 14918.74 |
| 3 | 5% | 7673.16 | **7572.00** | 7565.17 | 7570.67 | 7564.88 | 7569.38 | **7564.61** | **9056.19** | 8398.88 |
| | 10% | 7669.61 | **7578.28** | 7538.85 | 7570.11 | 7537.42 | 7561.98 | **7536.06** | **10456.04** | 9162.64 |
| | 15% | 7672.13 | 7684.51 | 7543.23 | 7655.01 | 7538.05 | **7624.46** | **7533.09** | **12503.74** | 10297.16 |
| | 20% | 7670.08 | 7870.63 | 7562.84 | 7806.53 | 7551.07 | 7736.98 | **7539.66** | **13836.23** | 11028.13 |
| | 30% | 7669.02 | 8762.02 | 7714.56 | 8549.37 | 7667.80 | 8292.14 | **7620.24** | **17266.53** | **12929.59** |
| | 40% | 7673.71 | 10614.88 | 8159.92 | **10128.11** | 8030.31 | 9462.42 | **7890.68** | **20873.34** | 14921.25 |
| 5 | 5% | 7674.97 | **7574.03** | 7567.15 | 7572.70 | 7566.87 | 7571.40 | **7566.59** | **9058.39** | 8401.17 |
| | 10% | 7671.16 | **7579.72** | 7540.42 | 7571.56 | 7538.99 | 7563.46 | **7537.64** | **10457.51** | 9163.75 |
| | 15% | 7673.01 | 7685.79 | 7544.40 | 7656.28 | 7539.20 | **7625.71** | **7534.23** | **12505.55** | 10298.90 |
| | 20% | 7671.81 | 7872.26 | 7564.52 | 7808.17 | 7552.76 | 7738.62 | **7541.35** | **13838.01** | 11029.75 |
| | 30% | 7669.11 | 8760.65 | 7713.94 | 8548.10 | 7667.24 | 8291.00 | **7619.76** | **17264.77** | **12927.62** |
| | 40% | 7672.27 | 10614.80 | 8159.43 | **10127.96** | 8029.79 | 9462.17 | **7890.12** | **20874.47** | 14921.71 |
| 7 | 5% | 7676.67 | **7576.30** | 7569.45 | 7574.96 | 7569.16 | 7573.67 | **7568.89** | **9060.22** | 8402.99 |
| | 10% | 7672.48 | **7580.58** | 7541.43 | 7572.44 | 7540.02 | 7564.36 | **7538.68** | **10458.07** | 9164.10 |
| | 15% | 7673.02 | 7685.82 | 7544.48 | 7656.32 | 7539.29 | **7625.76** | **7534.32** | **12504.37** | 10298.20 |
| | 20% | 7670.95 | 7871.37 | 7563.53 | 7807.27 | 7551.76 | 7737.71 | **7540.34** | **13836.47** | 11028.67 |
| | 30% | 7672.18 | 8764.70 | 7717.71 | 8552.11 | 7670.98 | 8294.96 | **7623.48** | **17268.89** | **12931.85** |
| | 40% | 7675.23 | 10616.85 | 8162.22 | **10130.12** | 8032.65 | 9464.50 | **7893.06** | **20875.20** | 14923.05 |
| 9 | 5% | 7677.90 | **7576.63** | 7569.77 | 7575.30 | 7569.49 | 7574.00 | **7569.21** | **9060.94** | 8403.68 |
| | 10% | 7673.86 | **7581.91** | 7542.72 | 7573.77 | 7541.30 | 7565.68 | **7539.96** | **10458.87** | 9165.22 |
| | 15% | 7677.05 | 7689.32 | 7548.37 | 7659.88 | 7543.21 | **7629.38** | **7538.27** | **12507.88** | 10301.18 |
| | 20% | 7677.31 | 7876.64 | 7569.43 | 7812.62 | 7557.71 | 7743.17 | **7546.35** | **13841.15** | 11033.01 |
| | 30% | 7673.94 | 8765.93 | 7718.76 | 8553.32 | 7672.02 | 8296.14 | **7624.49** | **17270.03** | **12933.14** |
| | 40% | 7677.31 | 10614.87 | 8161.61 | **10128.32** | 8032.18 | 9463.00 | **7892.74** | **20871.48** | 14919.99 |


**Table 27. Mean absolute percentage errors of platinum loss for the scenarios with error variances of 1, 3, 5, 7, and 9; a sample sizes of 48; and missing value percentages of 5, 10, 15, 20, 30, and 40% with SRS in missing value estimation methods.**

| Sample sizes | Missing values | Listwise delition | Missing value estimation methods | | | | | | | |
|---|---|---|---|---|---|---|---|---|---|---|
| | | | R-RQ1 | R-RQ3 | AR- CRQ1 | AR-CRQ3 | AR-MRQ1 | AR-MRQ3 | AR-MCRQ1 | AR-MCRQ3 |
| 1 | 5% | 1845.38 | **1920.83** | 1921.92 | 1921.00 | 1921.99 | 1921.17 | **1922.05** | **1858.38** | 1874.67 |
| a | 10% | 1860.95 | **1916.41** | 1924.01 | 1917.59 | 1924.48 | 1918.85 | **1924.95** | **1793.11** | 1827.52 |
| | 15% | 1878.49 | 1898.22 | 1918.64 | 1901.38 | 1919.93 | **1904.91** | **1921.29** | **1715.01** | 1769.94 |
| | 20% | 1879.69 | 1875.52 | 1915.06 | 1881.59 | 1917.70 | 1888.77 | **1920.52** | **1637.16** | 1714.64 |
| | 30% | 1880.03 | 1775.83 | 1865.33 | 1789.46 | 1871.99 | 1807.46 | **1879.46** | **1474.10** | **1591.35** |
| | 40% | 1893.33 | 1660.80 | 1804.83 | **1682.64** | 1816.94 | 1715.29 | **1831.37** | **1354.14** | 1502.12 |
| 3 | 5% | 1845.35 | **1920.77** | 1921.86 | 1920.94 | 1921.93 | 1921.12 | **1921.99** | **1858.34** | 1874.61 |
| | 10% | 1860.37 | **1915.75** | 1923.34 | 1916.93 | 1923.80 | 1918.19 | **1924.28** | **1792.74** | 1827.06 |
| | 15% | 1880.44 | 1900.05 | 1920.47 | 1903.20 | 1921.77 | **1906.74** | **1923.12** | **1716.79** | 1771.73 |
| | 20% | 1881.02 | 1876.82 | 1916.38 | 1882.90 | 1919.02 | 1890.09 | **1921.84** | **1638.34** | 1715.87 |
| | 30% | 1881.76 | 1777.34 | 1866.96 | 1790.99 | 1873.63 | 1809.01 | **1881.11** | **1475.21** | **1592.61** |
| | 40% | 1894.12 | 1661.68 | 1805.65 | **1683.51** | 1817.76 | 1716.14 | **1832.18** | **1355.06** | 1503.04 |
| 5 | 5% | 1845.91 | **1921.58** | 1922.67 | 1921.75 | 1922.74 | 1921.92 | **1922.80** | **1859.31** | 1875.55 |
| | 10% | 1861.13 | **1916.61** | 1924.21 | 1917.79 | 1924.67 | 1919.05 | **1925.15** | **1793.35** | 1827.74 |
| | 15% | 1878.94 | 1898.78 | 1919.18 | 1901.93 | 1920.47 | **1905.47** | **1921.83** | **1715.75** | 1770.61 |
| | 20% | 1879.91 | 1875.90 | 1915.50 | 1881.98 | 1918.14 | 1889.17 | **1920.96** | **1637.16** | 1714.77 |
| | 30% | 1882.68 | 1777.99 | 1867.66 | 1791.65 | 1874.34 | 1809.68 | **1881.82** | **1475.71** | **1593.17** |
| | 40% | 1891.96 | 1660.09 | 1803.85 | **1681.89** | 1815.94 | 1714.48 | **1830.33** | **1353.96** | 1501.72 |
| 7 | 5% | 1846.99 | **1922.85** | 1923.95 | 1923.02 | 1924.02 | 1923.20 | **1924.08** | **1859.96** | 1876.37 |
| | 10% | 1861.99 | **1917.55** | 1925.15 | 1918.73 | 1925.61 | 1919.99 | **1926.09** | **1794.28** | 1828.69 |
| | 15% | 1880.43 | 1900.11 | 1920.63 | 1903.28 | 1921.94 | **1906.84** | **1923.30** | **1715.87** | 1771.10 |
| | 20% | 1877.04 | 1872.98 | 1912.40 | 1879.04 | 1915.03 | 1886.20 | **1917.84** | **1635.42** | 1712.64 |
| | 30% | 1883.56 | 1779.74 | 1869.08 | 1793.35 | 1875.74 | 1811.32 | **1883.20** | **1478.54** | **1595.58** |
| | 40% | 1894.91 | 1661.91 | 1806.06 | **1683.77** | 1818.19 | 1716.44 | **1832.62** | **1354.92** | 1503.09 |
| 9 | 5% | 1845.53 | **1921.05** | 1922.14 | 1921.22 | 1922.21 | 1921.39 | **1922.27** | **1858.69** | 1874.95 |
| | 10% | 1861.10 | **1916.38** | 1923.97 | 1917.56 | 1924.43 | 1918.82 | **1924.90** | **1793.26** | 1827.63 |
| | 15% | 1879.45 | 1899.40 | 1919.76 | 1902.55 | 1921.06 | **1906.08** | **1922.41** | **1716.61** | 1771.46 |
| | 20% | 1877.75 | 1874.18 | 1913.58 | 1880.24 | 1916.22 | 1887.39 | **1919.02** | **1636.70** | 1713.91 |
| | 30% | 1880.98 | 1776.73 | 1866.15 | 1790.35 | 1872.81 | 1808.33 | **1880.27** | **1475.27** | **1592.41** |
| | 40% | 1898.07 | 1664.39 | 1808.93 | **1686.31** | 1821.08 | 1719.07 | **1835.56** | **1356.58** | 1505.16 |


**Table 28. Mean square errors of platinum loss for the scenarios with error variance of 1, 3, 5, 7, and 9; a sample sizes of 48; and missing value percentages of 5, 10, 15, 20, 30, and 40% with RSS in missing value estimation methods.**

| Sample sizes | Missing values | Listwise delition | Missing value estimation methods | | | | | | | |
|---|---|---|---|---|---|---|---|---|---|---|
| | | | R-RQ1 | R-RQ3 | AR- CRQ1 | AR-CRQ3 | AR-MRQ1 | AR-MRQ3 | AR-MCRQ1 | AR-MCRQ3 |
| 1 | 5% | 2373.82 | **2442.95** | 2474.57 | 2447.80 | 2476.48 | 2452.90 | **2478.42** | **2969.90** | **2231.39** |
| | 10% | 3260.79 | **3059.68** | 3161.23 | 3074.23 | 3167.98 | 3090.26 | **3174.95** | **5190.85** | 3804.00 |
| | 15% | 3693.78 | **3478.05** | 3550.00 | 3483.29 | 3557.50 | **3491.61** | **3565.67** | **8092.53** | 5660.11 |
| | 20% | 3786.14 | 3742.59 | **3685.02** | 3719.13 | 3688.95 | 3698.31 | **3694.14** | **9854.84** | 6758.91 |
| | 30% | 3894.93 | 4880.93 | 4094.55 | 4710.90 | 4066.46 | 4515.63 | **4039.74** | **14005.80** | **9286.46** |
| | 40% | 3956.21 | 6918.86 | 4704.15 | **6471.52** | 4595.95 | 5882.26 | **4482.23** | **17908.44** | 11576.70 |
| 3 | 5% | 2373.80 | **2443.80** | 2475.41 | 2448.65 | 2477.32 | 2453.75 | **2479.26** | **2971.61** | **2233.14** |
| | 10% | 3263.38 | **3063.04** | 3164.54 | 3077.58 | 3171.27 | 3093.60 | **3178.25** | **5194.48** | 3807.80 |
| | 15% | 3696.23 | **3479.94** | 3551.85 | 3485.17 | 3559.35 | **3493.48** | **3567.52** | **8094.40** | 5662.01 |
| | 20% | 3789.05 | 3746.26 | **3688.72** | 3722.80 | 3692.65 | 3702.00 | **3697.84** | **9857.93** | 6762.27 |
| | 30% | 3896.49 | 4881.42 | 4095.32 | 4711.43 | 4067.26 | 4516.21 | **4040.56** | **14006.41** | **9286.82** |
| | 40% | 3959.76 | 6920.67 | 4706.60 | **6473.39** | 4598.47 | 5884.26 | **4484.85** | **17910.51** | 11578.35 |
| 5 | 5% | 2376.11 | **2445.32** | 2476.88 | 2450.17 | 2478.79 | 2455.25 | **2480.72** | **2974.28** | **2235.78** |
| | 10% | 3265.05 | **3063.84** | 3165.38 | 3078.39 | 3172.12 | 3094.41 | **3179.10** | **5195.01** | 3808.22 |
| | 15% | 3697.42 | **3481.13** | 3552.92 | 3486.34 | 3560.41 | **3494.64** | **3568.57** | **8096.06** | 5663.66 |
| | 20% | 3789.58 | 3746.13 | **3688.61** | 3722.67 | 3692.54 | 3701.87 | **3697.74** | **9858.19** | 6762.33 |
| | 30% | 3898.07 | 4883.10 | 4097.11 | 4713.12 | 4069.05 | 4517.92 | **4042.36** | **14007.75** | **9288.30** |
| | 40% | 3960.17 | 6921.84 | 4707.64 | **6474.56** | 4599.50 | 5885.41 | **4485.84** | **17911.08** | 11579.38 |
| 7 | 5% | 2379.02 | **2449.04** | 2480.67 | 2453.89 | 2482.58 | 2458.99 | **2484.53** | **2975.79** | **2237.33** |
| | 10% | 3268.44 | **3067.32** | 3168.91 | 3081.87 | 3175.66 | 3097.91 | **3182.64** | **5198.32** | 3811.59 |
| | 15% | 3698.20 | **3483.60** | 3555.33 | 3488.80 | 3562.81 | **3497.09** | **3570.97** | **8098.29** | 5666.24 |
| | 20% | 3796.36 | 3750.80 | **3694.19** | 3727.48 | 3698.19 | 3706.83 | **3703.46** | **9861.54** | 6765.16 |
| | 30% | 3900.56 | 4884.47 | 4098.96 | 4714.56 | 4070.94 | 4519.46 | **4044.29** | **14008.12** | **9288.92** |
| | 40% | 3963.45 | 6924.75 | 4710.74 | **6477.50** | 4602.61 | 5888.40 | **4488.98** | **17913.57** | 11582.08 |
| 9 | 5% | 2380.44 | **2450.02** | 2481.58 | 2454.86 | 2483.49 | 2459.95 | **2485.43** | **2976.62** | **2238.14** |
| | 10% | 3268.43 | **3068.27** | 3169.98 | 3082.85 | 3176.74 | 3098.90 | **3183.72** | **5198.24** | 3811.56 |
| | 15% | 3701.49 | **3485.34** | 3557.65 | 3490.63 | 3565.18 | **3499.01** | **3573.38** | **8098.33** | 5665.96 |
| | 20% | 3796.18 | 3750.64 | **3693.81** | 3727.28 | 3697.79 | 3706.60 | **3703.04** | **9861.21** | 6765.22 |
| | 30% | 3904.09 | 4886.82 | 4101.58 | 4716.94 | 4073.60 | 4521.87 | **4046.99** | **14011.71** | **9291.64** |
| | 40% | 3965.08 | 6926.78 | 4712.75 | **6479.54** | 4604.61 | 5890.44 | **4490.97** | **17915.13** | 11583.97 |


**Table 29. Mean absolute percentage errors of platinum loss for the scenarios with error variances of 1, 3, 5, 7, and 9; a sample sizes of 48; and missing value percentages of 5, 10, 15, 20, 30, and 40% with RSS in missing value estimation methods.**

| Sample sizes | Missing values | Listwise delition | Missing value estimation methods | | | | | | | |
|---|---|---|---|---|---|---|---|---|---|---|
| | | | R-RQ1 | R-RQ3 | AR- CRQ1 | AR-CRQ3 | AR-MRQ1 | AR-MRQ3 | AR-MCRQ1 | AR-MCRQ3 |
| 1 | 5% | 183.017 | **520.402** | 529.401 | 521.807 | 529.933 | 523.275 | **530.475** | **296.273** | **356.122** |
| | 10% | 223.352 | **592.429** | 614.515 | 595.864 | 615.880 | 599.622 | **617.292** | **332.981** | 407.068 |
| | 15% | 234.951 | **597.476** | 632.030 | 602.826 | 634.268 | **608.970** | **636.629** | **343.127** | 421.180 |
| | 20% | 237.266 | 596.707 | **642.304** | 603.718 | 645.406 | 612.201 | **648.749** | **359.848** | 438.498 |
| | 30% | 240.547 | 550.734 | 615.430 | 560.562 | 620.354 | 573.896 | **625.943** | **353.594** | **431.945** |
| | 40% | 240.453 | 445.285 | 515.798 | **455.866** | 521.941 | 472.220 | **529.335** | **306.219** | 374.997 |
| 3 | 5% | 187.325 | **524.570** | 533.571 | 525.976 | 534.104 | 527.444 | **534.645** | **300.393** | **360.255** |
| | 10% | 225.155 | **597.388** | 619.506 | 600.828 | 620.873 | 604.592 | **622.288** | **337.555** | 411.753 |
| | 15% | 236.375 | **600.725** | 635.266 | 606.072 | 637.503 | **612.214** | **639.863** | **346.481** | 424.501 |
| | 20% | 239.059 | 598.453 | **644.127** | 605.474 | 647.235 | 613.972 | **650.583** | **361.192** | 439.975 |
| | 30% | 241.799 | 553.824 | 618.577 | 563.660 | 623.506 | 577.004 | **629.100** | **356.483** | **434.912** |
| | 40% | 241.671 | 445.618 | 515.946 | **456.171** | 522.073 | 472.481 | **529.449** | **306.962** | 375.532 |
| 5 | 5% | 190.114 | **528.231** | 537.225 | 529.636 | 537.757 | 531.103 | **538.298** | **304.232** | **364.046** |
| | 10% | 226.087 | **602.624** | 624.721 | 606.061 | 626.087 | 609.821 | **627.500** | **343.038** | 417.165 |
| | 15% | 235.732 | **605.057** | 639.507 | 610.391 | 641.738 | **616.517** | **644.092** | **351.484** | 429.295 |
| | 20% | 240.317 | 603.248 | **648.883** | 610.264 | 651.988 | 618.755 | **655.334** | **366.197** | 444.907 |
| | 30% | 243.332 | 557.000 | 621.785 | 566.843 | 626.716 | 580.195 | **632.313** | **359.584** | **438.042** |
| | 40% | 242.974 | 449.679 | 520.144 | **460.254** | 526.283 | 476.595 | **533.673** | **310.741** | 379.446 |
| 7 | 5% | 196.337 | **532.991** | 541.989 | 534.396 | 542.521 | 535.864 | **543.063** | **308.878** | **368.722** |
| | 10% | 231.359 | **604.975** | 627.098 | 608.415 | 628.465 | 612.180 | **629.880** | **345.077** | 419.293 |
| | 15% | 238.882 | **607.415** | 641.932 | 612.759 | 644.168 | **618.897** | **646.526** | **353.323** | 431.296 |
| | 20% | 243.216 | 605.512 | **651.190** | 612.535 | 654.298 | 621.034 | **657.647** | **368.226** | 447.014 |
| | 30% | 245.866 | 557.648 | 622.258 | 567.463 | 627.177 | 580.777 | **632.758** | **360.786** | **439.020** |
| | 40% | 244.057 | 453.691 | 524.394 | **464.300** | 530.553 | 480.693 | **537.967** | **314.331** | 383.248 |
| 9 | 5% | 201.096 | **537.112** | 546.105 | 538.516 | 546.638 | 539.984 | **547.179** | **313.109** | **372.923** |
| | 10% | 234.887 | **606.881** | 628.984 | 610.318 | 630.350 | 614.080 | **631.764** | **347.217** | 421.366 |
| | 15% | 244.796 | **611.062** | 645.651 | 616.417 | 647.891 | **622.567** | **650.254** | **356.446** | 434.579 |
| | 20% | 243.653 | 609.860 | **655.501** | 616.877 | 658.607 | 625.370 | **661.953** | **372.774** | 451.495 |
| | 30% | 248.577 | 561.162 | 625.799 | 570.983 | 630.721 | 584.303 | **636.305** | **364.196** | **442.473** |
| | 40% | 250.278 | 455.222 | 526.100 | **465.859** | 532.272 | 482.294 | **539.704** | **315.507** | 384.585 |


**Table 30. The methods provided the minimum mean square error and mean absolute percentage error with SRS and RSS for each error variance and sample size condition.**

| Error variances | Sample sizes | SRS | | RSS | |
|---|---|---|---|---|---|
| | | MSE | MAPE | MSE | MAPE |
| 1 | Small | AR-MRQ1 | AR-MCRQ1 | AR-MRQ1 | AR-MRQ1 |
| | Middle | AR-MRQ1 | AR-MCRQ1 | AR-MRQ1 | AR-MRQ1 |
| | Large | AR-MRQ1 | AR-MCRQ1 | AR-MRQ1 | AR-RQ1 |
| | Huge | AR-MRQ1 | AR-MCRQ1 | AR-MRQ1 | AR-MRQ1 |
| 3 | Small | AR-MRQ1 | AR-MCRQ1 | AR-MCRQ3 | AR-MCRQ3 |
| | Middle | AR-MRQ1 | AR-MCRQ1 | AR-MCRQ3 | AR-MCRQ3 |
| | Large | AR-MRQ1 | AR-MCRQ1 | AR-MCRQ3 | AR-MCRQ3 |
| | Huge | AR-MRQ1 | AR-MCRQ1 | AR-MCRQ3 | AR-MCRQ3 |
| 5 | Small | AR-MCRQ3 | AR-MCRQ1 | AR-MCRQ3 | AR-MCRQ1 |
| | Middle | AR-MCRQ3 | AR-MCRQ1 | AR-MCRQ3 | AR-MCRQ1 |
| | Large | AR-MCRQ3 | AR-MCRQ1 | AR-MCRQ3 | AR-MCRQ1 |
| | Huge | AR-MCRQ3 | AR-MCRQ1 | AR-MCRQ3 | AR-MCRQ1 |
| 7 | Small | AR-MCRQ3 | AR-MCRQ1 | AR-MCRQ3 | AR-MCRQ1 |
| | Middle | AR-MCRQ3 | AR-MCRQ1 | AR-MCRQ3 | AR-MCRQ1 |
| | Large | AR-MCRQ3 | AR-MCRQ1 | AR-MCRQ3 | AR-MCRQ1 |
| | Huge | AR-MCRQ3 | AR-MCRQ1 | AR-MCRQ3 | AR-MCRQ1 |
| 9 | Small | AR-MCRQ3 | AR-MCRQ1 | AR-MCRQ3 | AR-MCRQ1 |
| | Middle | AR-MCRQ3 | AR-MCRQ1 | AR-MCRQ3 | AR-MCRQ1 |
| | Large | AR-MCRQ3 | AR-MCRQ1 | AR-MCRQ1 | AR-MCRQ1 |
| | Huge | AR-MCRQ3 | AR-MCRQ1 | AR-MCRQ1 | AR-MCRQ1 |


**Table 31. The methods provided the minimum mean square error and mean absolute percentage error with SRS and RSS for each error variance condition.**

| Error variances | SRS | | RSS | |
|---|---|---|---|---|
| | MSE | MAPE | MSE | MAPE |
| Small | AR-MRQ1 | AR-MCRQ1 | AR-MRQ1 | AR-MRQ1 |
| Middle | AR-MCRQ3 | AR-MCRQ1 | AR-MCRQ3 | AR-MCRQ1 |
| Large | AR-MCRQ3 | AR-MCRQ1 | AR-MCRQ1, AR-MCRQ3 | AR-MCRQ1 |


Our results agree well with those reported by [3–5,13]: the RSS estimators were more efficient than the SRS estimators for the same quartile, coefficient of correlation, and sample size.

## 5. Conclusion

This study compared the efficiencies of four new adjusted missing value imputation methods in multiple regression analysis. The four estimation methods were the following: a regression-ratio quartile1,3 (R-RQ1,3) imputation of Al-Omari, Jemain, and Ibrahim; an adjusted regression-chain ratio quartile1,3 (AR-CRQ1,3) imputation of Kadilar and Cinji; an adjusted regression-multivariate ratio quatile1,3 (AR-MRQ1,3) imputation of Feng, Ni, and Zou; and an adjusted regression-multivariate chain ratio quartile1,3 (AR-MCRQ1,3) imputation of Lu for each simple random sampling (SRS) and rank set sampling (RSS). The measures for comparing the performance were mean square error (MSE) and mean absolute percentage error (MAPE).

**Table 32. The number of times for methods provided the smaller mean square error and mean absolute percentage error with SRS and RSS for each error variance condition.**

| Error variances | Sample sizes | Scenarios in total | Smaller MSE | | Smaller MAPE | |
|---|---|---|---|---|---|---|
| | | | SRS | RSS | SRS | RSS |
| 1 | Small | 96 | 11 | **85** | 0 | **96** |
| | Middle | 96 | 1 | **95** | 0 | **96** |
| | Large | 96 | 2 | **94** | 0 | **96** |
| | Huge | 96 | 0 | **96** | 0 | **96** |
| 3 | Small | 96 | 20 | **76** | 0 | **96** |
| | Middle | 96 | 0 | **96** | 0 | **96** |
| | Large | 96 | 5 | **91** | 0 | **96** |
| | Huge | 96 | 10 | **86** | 0 | **96** |
| 5 | Small | 96 | 26 | **70** | 0 | **96** |
| | Middle | 96 | 15 | **81** | 0 | **96** |
| | Large | 96 | 4 | **92** | 0 | **96** |
| | Huge | 96 | 0 | **96** | 0 | **96** |
| 7 | Small | 96 | 32 | **64** | 0 | **96** |
| | Middle | 96 | 8 | **88** | 0 | **96** |
| | Large | 96 | 12 | **84** | 0 | **96** |
| | Huge | 96 | 14 | **82** | 8 | **88** |
| 9 | Small | 96 | 26 | **70** | 0 | **96** |
| | Middle | 96 | 22 | **74** | 0 | **96** |
| | Large | 96 | 17 | **79** | 0 | **96** |
| | Huge | 96 | 20 | **76** | 0 | **96** |

Bold values indicate the better SRS or RSS estimators when comparing the smaller MSE and MAPE. Each sample size provided two values, the proportions of missing values provided six values, and missing value estimation methods provided eight values. There were 2x6x8 = 96 scenarios in total for each sampling.


## Future recommendations

5.1 The random forest imputation may be a suitable method for estimating missing values imputations in multiple regression analysis [21]. A comparative study should be conducted between this imputation method and the author's method.

5.2 Many kinds of variables can have missing values. We may consider missing value imputation for the dependent variable or that for both the independent and dependent variables [30]. An attempt at imputation of missing values for the dependent variable and for both the independent and dependent variables, although it will be extensive work, should be considered.

## Supporting information

**S1 Code. The data used to support this study were simulated from a normal distribution using the R studio program.**
(ZIP)

**S2 Code. The data used to support this study were simulated from a normal distribution using the R studio program.**
(ZIP)

**Actual Data. The data used for gold and platinum loss by customer groups.**
(PDF)

## Acknowledgments

The funders had no role in study design, data collection and analysis, publication decisions, or manuscript preparation.

## Author contributions

**Conceptualization:** Saichon Sinsomboonthong.

**Formal analysis:** Saichon Sinsomboonthong.

**Funding acquisition:** Saichon Sinsomboonthong.

**Investigation:** Juthaphorn Sinsomboonthong, Saichon Sinsomboonthong.

**Methodology:** Juthaphorn Sinsomboonthong, Saichon Sinsomboonthong.

**Resources:** Saichon Sinsomboonthong.

**Software:** Juthaphorn Sinsomboonthong, Saichon Sinsomboonthong.

**Validation:** Saichon Sinsomboonthong.

**Writing – original draft:** Saichon Sinsomboonthong.

**Writing – review & editing:** Saichon Sinsomboonthong.

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
