## [Decision Letter · Decision Letter 0]

3 May 2024

PONE-D-24-09564New Adjusted Missing Value Imputation in Multiple Regression with Simple Random Sampling and Rank Set Sampling MethodsPLOS ONE

Dear Dr. Sinsomboonthong,

Thank you for submitting your manuscript to PLOS ONE. After careful consideration, we feel that it has merit but does not fully meet PLOS ONE’s publication criteria as it currently stands. Therefore, we invite you to submit a revised version of the manuscript that addresses the points raised during the review process. Both reviewers were very critical about your manuscript. Revising your manuscript won't guarantee publication in PLOS ONE eventually. If you choose to revise your manuscript, please do so taking into account the reviewers' comments very carefully.

We look forward to receiving your revised manuscript.

Kind regards,

Robin Haunschild

Academic Editor

PLOS ONE

“School of Science, King Mongkut’s Institute of Technology Ladkrabang (KMITL) for consideration of funding this research project [grant number 2567-02-5-001]”

4. Please update your submission to use the PLOS LaTeX template. The template and more information on our requirements for LaTeX submissions can be found at http://journals.plos.org/plosone/s/latex.

“The authors would like to thank the committees of the Department of Statistics and other

Departments, School of Science, King Mongkut’s Institute of Technology Ladkrabang (KMITL)

for consideration of funding this research project [grant number 2567-02-5-001].”

“School of Science, King Mongkut’s Institute of Technology Ladkrabang (KMITL) for consideration of funding this research project [grant number 2567-02-5-001]”

Reviewers' comments:

Reviewer's Responses to Questions

**Comments to the Author**

1. Is the manuscript technically sound, and do the data support the conclusions?

Reviewer #1: No

Reviewer #2: Partly

2. Has the statistical analysis been performed appropriately and rigorously? 

Reviewer #1: N/A

Reviewer #2: No

3. Have the authors made all data underlying the findings in their manuscript fully available?

Reviewer #1: No

Reviewer #2: Yes

4. Is the manuscript presented in an intelligible fashion and written in standard English?

Reviewer #1: No

Reviewer #2: Yes

5. Review Comments to the Author

Reviewer #1: The manuscript investigates some imputation methods in multiple regression analysis for missing values. In essence, this work is only considered as the research report rather than technique article. I cannot understand the specfic contribution of this work. Meanwhile, the writing and organization is poor. Therefore, I cannot recommend this paper for publication in PLOS ONE.

Reviewer #2: The manuscript focusses on the efficiency of four imputation methods: regression-ratio quartile1,3 (R-RQ1,3) imputation of Al-Omari, Jemain and Ibrahim; adjusted regression-chain ratio quartile1,3 (AR-CRQ1,3) imputation of Kadilar and Cinji; adjusted regression-multivariate ratio quatile1,3 (ARMRQ1,3) imputation of Feng, Ni, and Zou; and adjusted regression-multivariate chain ratio quartile1,3 (AR-MCRQ1,3) imputation of Lu for each simple random sampling (SRS) and rank set sampling (RSS). Simulation studies were conducted to investigate the performance of different methods using mean square error and mean absolute error.

The manuscript addresses an interesting and important topic of missing imputation and is written in a clear and concise manner, but a major revision is needed, which should address the following concerns:

- Literature review: The literature review is very selective, sorted by year of publication, and mainly includes older literature related to the approaches proposed by the authors. Important statistical literature on missing imputation in regression analysis (e.g. Song & Guo, 2024; Thongsri & Samart, 2022, see references) is not included, nor is included important basic literature to missing imputation, e.g. by Rubin or Little (see references). The question of alternatives to multiple imputation, e.g. maximum likelihood, is not discussed at all (e.g. Chen & Ibrahim, 2013). The work of Al-Omari, Jemain and Ibrahim does only refer to SRS, but is not related to missing imputation. To contribute to the current literature on missing imputation in linear regression would also require a review of the entire literature on the subject. Last but not least, the four proposed methods of missing imputation are new. They are mentioned in the introduction, but the more detailed explanations only follow in the methods section. Readers who are not familiar with SRS have no chance of understanding the idea of these new methods. It would be helpful to explain the basic idea of these procedures right in the beginning.

- New methods of missing imputation: As these are new approaches developed by the authors themselves ("The authors propose three new methods"), actually a comparison with common missing imputation methods in linear regression, such as stochastic regression imputation, predictive mean matching imputation or random forest imputation (e.g., Thongsri & Samart, 2022) may be necessary.

- Assumptions: Three assumptions are discussed in the context of missing imputation, which also form the basis of later simulations: "missing completely at random (MCAR)", "missing at random (MAR)" and "missing not at random (MNAR)" (Rubin, 1987; van Buuren, 2018). Only the MAR assumption is briefly mentioned in the introduction and in the simulation study, but not elaborated on. The missing imputation mechanism and the assumptions should also be addressed in much more detail. What is the case, if the data are MNAR?

- MAR: MAR is assumed in the simulation study (p. 8). Roughly speaking, if a missing value of any variable in the dataset depends on observed values of other variables in the dataset, then it is defined as MAR. I wonder how MAR is realized. Since X1 is complete, the data in X2 and its missingness should related to X1, i.e. the higher X1, the more missing values in X2. In the simulation, however, X1 and X2 are simulated independently of each other and the positions of the missing values are randomized. This is more in favor of MCAR. These issues would need further clarification in the revision. The variance inflation factor (VIF), which also takes into account the sample size and the root mean square error, would be a more appropriate measure of multiple collinearity than the Pearson correlation.

- SRS estimator: The ratio estimator introduced on page 10 needs further explanation, especially in the context of missing values and missing value imputation. The presentation of Eq. 1 is somewhat confusing. In my view less confusing is the definition by Lu (2013): u_YSRS = mean_y_srs*(u_x/mean_xsrs), where u_x is replaced by mean_x_in (Eq. 5). For me it is not quite clear what is meant by “mean_x_in is the perfect sample mean of independent variable X2”. Common taxonomies in missing value statistics with a missing value variable R, with R=1 (complete) and R=0 (missing) may be helpful in the statistical derivations.

- Conclusions: It is relatively difficult to draw practical implications from the simulation studies, as certain methods of multiple imputation are better or worse depending on the level of error variance and the measures (mean absolute percentage error, mean square error), comparable to the heterogeneous results of Jomprapan (2012). Furthermore, I wonder what a researcher should do when he or she has a linear regression with more than 2 predictors. The practical implications should be made clear. In the statistical literature missing imputation approaches are discussed for the dependent variables, for the covariates and for both variables (e.g., Shao, 2013). It is often the case in applications that there are missing values in both the dependent variable and the covariates. The proposed approaches, which focus on the covariates, therefore appear to be of only limited use.

References

Chen Q., Ibrahim J.G. (2013). A note on the relationships between multiple imputation, maximum likelihood and fully bayesian methods for missing responses in linear regression models. Statistics and its Interface, 6(3), 315-324.

Hasan H., Ahmad S., Osman B.M., Sapri S., Othman N. (2017). A comparison of model-based imputation methods for handling missing predictor values in a linear regression model: A simulation study. AIP Conference Proceedings, 1870, art. no. 060003

Little R.J. (2021). Missing data assumptions. Annual Review of Statistics and Its Application, 8, pp. 89 – 107.

Little R.J. (1992). Regression with missing X’s: a review. J. Am. Stat. Assoc. 87:1227–3

Rubin D.B. (1987). Multiple Imputation for Nonresponse in Surveys. New York: Wiley

Shao, J. (2013). Estimation and imputation in linear regression with missing values in both response and covariate. Statistics and its Interface, 6 (3), pp. 361 - 368

Song, L., Guo, G. (2024). Full Information Multiple Imputation for Linear Regression Model with Missing Response Variable. Journal of Applied Mathematics, 54 (1), pp. 77 - 81

Thongsri T., Samart K. (2022). Development of Imputation Methods for Missing Data in Multiple Linear Regression Analysis. Lobachevskii Journal of Mathematics, 43 (11), pp. 3390 – 3399

Von Buuren, S. (2018). Flexible Imputation of Missing data. New York: Chapman & Hall.

Yang X., Belin T.R., Boscardin W.J. (2005). Imputation and variable selection in linear regression models with missing covariates. Biometrics, 61 (2), pp. 498 - 506

6. PLOS authors have the option to publish the peer review history of their article (what does this mean?). If published, this will include your full peer review and any attached files.

Reviewer #1: No

Reviewer #2: No

---

## [Author Response · Author response to Decision Letter 1]

5 Jul 2024

I have respond comments of reviewers and editor in file of title "Response to Reviewers".

---

## [Decision Letter · Decision Letter 1]

7 Aug 2024

PONE-D-24-09564R1New adjusted missing value imputation in multiple regression with simple random sampling and rank set sampling methodsPLOS ONE

Dear Dr. Sinsomboonthong,

Thank you for submitting your manuscript to PLOS ONE. After careful consideration, we feel that it has merit but does not fully meet PLOS ONE’s publication criteria as it currently stands. Therefore, we invite you to submit a revised version of the manuscript that addresses the points raised during the review process.

We look forward to receiving your revised manuscript.

Kind regards,

Robin Haunschild

Academic Editor

PLOS ONE

Reviewers' comments:

Reviewer's Responses to Questions

**Comments to the Author**

1. If the authors have adequately addressed your comments raised in a previous round of review and you feel that this manuscript is now acceptable for publication, you may indicate that here to bypass the “Comments to the Author” section, enter your conflict of interest statement in the “Confidential to Editor” section, and submit your "Accept" recommendation.

Reviewer #1: All comments have been addressed

Reviewer #2: All comments have been addressed

2. Is the manuscript technically sound, and do the data support the conclusions?

Reviewer #1: Yes

Reviewer #2: Partly

3. Has the statistical analysis been performed appropriately and rigorously? 

Reviewer #1: Yes

Reviewer #2: No

4. Have the authors made all data underlying the findings in their manuscript fully available?

Reviewer #1: Yes

Reviewer #2: Yes

5. Is the manuscript presented in an intelligible fashion and written in standard English?

Reviewer #1: Yes

Reviewer #2: Yes

6. Review Comments to the Author

**Reviewer #1: **(No Response)

**Reviewer #2:** The revised manuscript focusses on the efficiency of four imputation methods: regression-ratio quartile1,3 (R-RQ1,3) imputation of Al-Omari, Jemain and Ibrahim; adjusted regression-chain ratio quartile1,3 (AR-CRQ1,3) imputation of Kadilar and Cinji; adjusted regression-multivariate ratio quatile1,3 (ARMRQ1,3) imputation of Feng, Ni, and Zou; and adjusted regression-multivariate chain ratio quartile1,3 (AR-MCRQ1,3) imputation of Lu for each simple random sampling (SRS) and rank set sampling (RSS). Simulation studies were conducted to investigate the performance of different methods using mean square error and mean absolute error.

I really appreciate the improvements the authors have made to the manuscript (e.g. discussion of MAR and MCAR). Thank you very much! Despite these improvements and the importance of the topic of missing imputation in linear regression, I cannot recommend the revised manuscript for publication in PlosOne for the following reasons:

- MCAR: A strong limitation of the results is the assumption of MCAR in the simulation study. Listwise deletion or maximum likelihood estimation would be equally effective. The advantage of missing imputation over complete case analysis is that it avoids the reduction in sample size and diminishes the confidence intervals of the parameters, as mentioned in the manuscript. However, the simulation study would need to include at least one MAR condition. It would also need to include a control condition with listwise deletion. Overall, MCAR seems to be rather unrealistic for practical applications.

- MAR: On page 6 we read: "Although it is easy to show that full-case analysis is unbiased and efficient when the missing at random (MAR) assumption is made, the aforementioned methods are still commonly used in practice for this setting". Strictly speaking, this statement only applies in the case of MCAR or in the case of missing values only for the outcome variable. Under the assumption of MAR, complete case analysis without missing imputation or full information maximum likelihood of missing values in the covariates might lead to bias.

- Generalizability of the results: The results of the simulation study ultimately relate to a regression analysis with 2 covariates and MCAR. This is a very limited scope of the approach.

- Sample size (Table 1). It makes little sense to include very small sample sizes of 20 or 40 with 2 covariates as a condition in the simulation. Higher sample sizes of 1000 and more and higher proportions of missing values of 40-50% would also be of interest. Since MCAR is assumed, the question is whether a proportion of 5-10% missing values really significantly increases the power of the statistical tests, especially for larger sample sizes above 100. I fear that the differences between the methods will diminish with larger sample sizes and smaller proportions of missing values.

- Alternatives: Alternatives to the proposed approaches are discussed but not considered in the analysis (e.g., FIML, complete case analysis), the results remain idiosyncratic. In statistics as well, it is important not only to replicate the results of other authors' simulation studies but also to compare the results with the results of other studies with the sample design of one's own study.

7. PLOS authors have the option to publish the peer review history of their article (what does this mean?). If published, this will include your full peer review and any attached files.

Reviewer #1: No

Reviewer #2: No

---

## [Author Response · Author response to Decision Letter 2]

29 Aug 2024

The authors have made all the second revision in accordance with the recommendations of the reviewers.

---

## [Decision Letter · Decision Letter 2]

28 Nov 2024

PONE-D-24-09564R2New adjusted missing value imputation in multiple regression with simple random sampling and rank set sampling methodsPLOS ONE

Dear Dr. Sinsomboonthong,

Thank you for submitting your manuscript to PLOS ONE. After careful consideration, we feel that it has merit but does not fully meet PLOS ONE’s publication criteria as it currently stands. Therefore, we invite you to submit a revised version of the manuscript that addresses the points raised during the review process.

We look forward to receiving your revised manuscript.

Kind regards,

Robin Haunschild

Academic Editor

PLOS ONE

Journal Requirements:

Reviewers' comments:

Reviewer's Responses to Questions

**Comments to the Author**

1. If the authors have adequately addressed your comments raised in a previous round of review and you feel that this manuscript is now acceptable for publication, you may indicate that here to bypass the “Comments to the Author” section, enter your conflict of interest statement in the “Confidential to Editor” section, and submit your "Accept" recommendation.

Reviewer #3: All comments have been addressed

Reviewer #4: All comments have been addressed

2. Is the manuscript technically sound, and do the data support the conclusions?

Reviewer #3: Yes

Reviewer #4: Yes

3. Has the statistical analysis been performed appropriately and rigorously? 

Reviewer #3: Yes

Reviewer #4: Yes

4. Have the authors made all data underlying the findings in their manuscript fully available?

Reviewer #3: Yes

Reviewer #4: Yes

5. Is the manuscript presented in an intelligible fashion and written in standard English?

Reviewer #3: Yes

Reviewer #4: No

6. Review Comments to the Author

Reviewer #3: Thank you for the opportunity to review your paper on New Adjusted Missing Value Imputation in Multiple Regression with Simple Random Sampling and Rank Set Sampling Methods. This paper makes a noteworthy contribution to statistical methodologies for handling missing data in regression analysis, an area of significant importance for researchers facing incomplete datasets. I am pleased to recommend this paper for publication, as it presents a valuable contribution to the field of statistical analysis in handling missing data within regression models. Congratulations on this well-executed and impactful research!

Reviewer #4: The authors are required to revise all in-text citations to ensure that they are consistent throughout the paper and adhere to the PLOS ONE format. Additionally, the quality of the English writing must be improved by having a native speaker proofread this work.

7. PLOS authors have the option to publish the peer review history of their article (what does this mean?). If published, this will include your full peer review and any attached files.

Reviewer #3: No

Reviewer #4: No

---

## [Editor Report · Decision Letter 3]

16 Dec 2024

New adjusted missing value imputation in multiple regression with simple random sampling and rank set sampling methods

PONE-D-24-09564R3

Dear Dr. Sinsomboonthong,

We’re pleased to inform you that your manuscript has been judged scientifically suitable for publication and will be formally accepted for publication once it meets all outstanding technical requirements.

Kind regards,

Robin Haunschild

Academic Editor

PLOS ONE
---

## [Editor Report · Acceptance letter]

PONE-D-24-09564R3

PLOS ONE

Dear Dr. Sinsomboonthong,

I'm pleased to inform you that your manuscript has been deemed suitable for publication in PLOS ONE. Congratulations! Your manuscript is now being handed over to our production team.

Kind regards,

on behalf of

Dr. Robin Haunschild

Academic Editor

PLOS ONE